# Formation processes, fire use, and patterns of human occupation across the Middle Palaeolithic (MIS 5a-5b) of Gruta da Oliveira (Almonda karst system, Torres Novas, Portugal)

Diego E. Angelucci[1,2], Mariana Nabais[2,3], João Zilhão[2]*

1 Dipartimento di Lettere e Filosofia, Università degli Studi di Trento, Trento, Italy, 2 UNIARQ, Centro de Arqueologia da Universidade de Lisboa, Lisboa, Portugal, 3 Institut Català de Paleoecologia Humana i Evolució Social, Tarragona, Spain

* joao.zilhao@campus.ul.pt

**Data Availability Statement:** All relevant data are within the paper.

## Abstract

Gruta da Oliveira features a *c.* 13 m-thick infilling that includes a *c.* 6.5 m-thick archaeological deposit (the "Middle Palaeolithic sequence" complex), which Bayesian modelling of available dating results places in MIS 5a (layers 7–14) and MIS 5b (layers 15–25), *c.* 71,000–93,000 years ago. The accumulation primarily consists of sediment washed in from the slope through gravitational processes and surface dynamics. The coarse fraction derives from weathering of the cave's limestone bedrock. Tectonic activity and structural instability caused the erosional retreat of the scarp face, explaining the large, roof-collapsed rock masses found through the stratification. The changes in deposition and diagenesis observed across the archaeological sequence are minor and primarily controlled by local factors and the impact of humans and other biological agents. Pulses of stadial accumulation—reflected in the composition of the assemblages of hunted ungulates, mostly open-country and rocky terrain taxa (rhino, horse, ibex)—alternate with interstadial hiatuses—during which carbonate crusts and flowstone formed. Humans were active at the cave throughout, but occupation was intermittent, which allowed for limited usage by carnivores when people visited less frequently. During the accumulation of layers 15–25 (*c.* 85,000–93,000 years ago), the carnivore guild was dominated by wolf and lion, while brown bear and lynx predominate in layers 7–14 (*c.* 71,000–78,000 years ago). In the excavated areas, conditions for residential use were optimal during the accumulation of layers 20–22 (*c.* 90,000–92,000 years ago) and 14 (*c.* 76,000–78,000 years ago), which yielded dense, hearth-focused scatters of stone tools and burnt bones. The latter are ubiquitous, adding to the growing body of evidence that Middle Palaeolithic Neandertals used fire in regular, consistent manner. The patterns of site usage revealed at Gruta da Oliveira are no different from those observed 50,000 years later in comparable early Upper Palaeolithic and Solutrean cave sites of central Portugal.

**Funding:** Funding for this work was provided by FCT (Fundação para a Ciência e Tecnologia; Portugal) grants UIDB/00698/2020 and UIDP/00698/2020. The funders had no role in study design, data collection and analysis, decision to publish, or preparation of the manuscript.

**Competing interests:** The authors have declared that no competing interests exist.

# 1. Introduction

## 1.1. Background

The last decades' drawn-out debates on the place occupied by Neandertals in the human story have largely been settled by widespread acknowledgment that they made a significant contribution to the genomes of the Upper Palaeolithic peoples of Eurasia, one that is still readily apparent in the continent's present-day populations [1–4]. However, the extent to which Neandertals were cognitively and behaviourally equivalent—whether in capabilities, achievements, or both—to the early anatomically modern human peoples of sub-Saharan Africa remains contentious. For instance, claims that cave art in Borneo and Sulawesi date back beyond 40–45 ka (thousands of years) ago tend to be readily accepted under the assumption that they must have been made by anatomically modern people migrating out of Africa—even though no actual fossil remains of such people exist in the region that are of comparable antiquity [5, 6]; yet, the age in excess of 65 ka obtained for Spanish cave art, implying Neandertal authorship, has been questioned on technical grounds—even though exactly the same dating technique (U-series) and experimental approach (sequential sampling across microlayers of calcite overlying the art) were used [7–14].

If the contradictions entailed in the stance of Neandertal art sceptics are glaring and bespeak of paradigmatic bias [15], the fact nevertheless remains that opposing views of Neandertal culture do involve relevant issues of empirical substance. Among others, examples thereof are whether articulated Neandertal skeletons imply burial, whether the charcoal and burnt bone present in occupation floors imply controlled use of fire, or whether the spatial patterning of archaeological remains implies living arrangements akin to those observed among extant hunter-gatherers [16–25]. These are key issues that have one thing in common: the archaeological evidence that arguments depend upon is highly sensitive to the impact of site formation processes and, therefore, requires critical examination under the taphonomy lens. The limited re-excavation of classic sites and the sampling of their extant profiles for analysis with new methods or new techniques can provide new and useful insights; clearly, however, the best way to move forward is through the detection and exploration of pristine sites using modern techniques of excavation and recording, thereby making it possible to overcome the issues of ambiguity or incompleteness that often hinder attempts to bring earlier work to bear on newer questions.

It was in the context of this understanding of how to advance our knowledge of the Middle and Upper Pleistocene archaeology of western Iberia that the Almonda karst research project was initiated, in 1987. The rationale was that the importance of a water outlet—the karst spring of the Almonda river—must have attracted human settlement since remote times and so that Palaeolithic sites, albeit as-yet unknown, must exist in the limestone scarp rising behind and above. This prediction was verified by systematic speleo-archaeological survey, which identified a number of localities at different elevations of an interconnected staircase of large passages, some of which have since been open for regular archaeological excavation [26–28]. Among the latter, the Middle Palaeolithic site of Gruta da Oliveira has been shown to be of particular relevance for issues of Neandertal evolution and Neandertal culture.

## 1.2. The site

The lower Tagus flows across a Tertiary basin separated from Portuguese Estremadura's Central Limestone Massif by a *c.* 40 km-long, NE-SW-oriented, up to 100 m-high escarpment, regionally known as the *Arrife*. This major landscape feature represents the surface evidence (exposed over time as a result of the Pleistocene incision of the basin's fluvial network and

attendant erosional dynamics) of a reverse fault generated as the siliciclastic deposits of the Lower Tagus Basin were overthrust by the Massif's Mesozoic sedimentary rocks [29, 30] (Fig 1).

A number of resurgences exist along this tectonic contact; the spring of River Almonda, a *c.* 20 km-long tributary of the Tagus, is one of the most important in terms of water discharge. Two underground streams converge at this outlet: the North River drains the Vale da Serra synclinal, and the West River drains the Mira-Minde polje. The bedrock consists of Middle Jurassic limestone, including white to light grey oolitic, microcrystalline, and massive limestone, with subordinate marly limestone, calcirudite, and calcilutite. Upslope, beyond the escarpment's crest, there are outcrops of the synclinal's late Tertiary sedimentary cover, which is composed of sand, conglomerate, and clay.

The karst network associated with the Almonda spring is extensive (*c.* 12 km have already been mapped) and developed as a multiphase maze cave system ([32]:233). The several collapsed entrances identified by the system's speleo-archaeological survey mostly correspond to fossil resurgences where water ceased to circulate because of the incision of the hydrological basin, causing the migration of the karst's vadose and phreatic zones to progressively lower elevations. Humans and animals used those underground spaces as they became dry and available for occupation, leading to the formation of an outstanding archaeological and paleoenvironmental record spanning the last half a million years of the Quaternary.

At the base of the scarp face, just above the spring, the Galeria da Cisterna features brecciated remnants of Upper Palaeolithic age and a major Early Neolithic burial context, as well as evidence of continued funerary usage through the Bronze Age, the Iron Age, and the Roman Era [26, 33–41]. A few metres higher-up, the Lapa dos Coelhos was occupied during the Magdalenian and the Solutrean [42–46].

At the top of the scarp face, three sites are known. Gruta da Aroeira is a *c.* 400 ka-old Acheulean context rich in stone tools and animal remains that also yielded the Aroeira 3 human cranium and evidence of in situ use of fire [47–62]. The others are Gruta do Pinheiro and Gruta do Aderno, both of Middle (or early Upper) Pleistocene age; except for the annual reports submitted to the heritage authorities, they remain unpublished.

Gruta da Oliveira is located half-way up the scarp face. It contains an archaeological stratigraphy comprising 19 units (layers 7–25) that U-series and luminescence methods have dated to the second half of Marine Isotope Stage (MIS) 5, within the *c.* 70–110 ka interval. Based on Bayesian modelling of the dating results, the site's chronostratigraphic framework can be summarised as follows (Fig 2):

- layers 8–14 likely date to GS (Greenland Stadial) 20 (72.3–74.1 ka ago) and GS 21 (76.4–77.8 ka ago), with the carbonate incrustation seen across the site at the interface between layers 12 and 13 (see below) possibly reflecting the intervening, short GI (Greenland Interstadial) 20 interstadial;

- layers 15–19 likely date to GS 22 (85.1–87.6 ka ago);

- layers 20–22 likely date to GS 23, a very short quasi-stadial, and the cooler end of GI 23 (90.1 to approximately 92.0 ka ago);

- layers 23–25 likely date to the earlier, warmer parts of GI 23 (>92 ka ago).

The discovery of the Gruta da Oliveira took place in 1989 and its excavation, initiated the following year, carried on until 2012. The rich wood charcoal and animal bone assemblages retrieved are associated with human remains and abundant, characteristically Middle

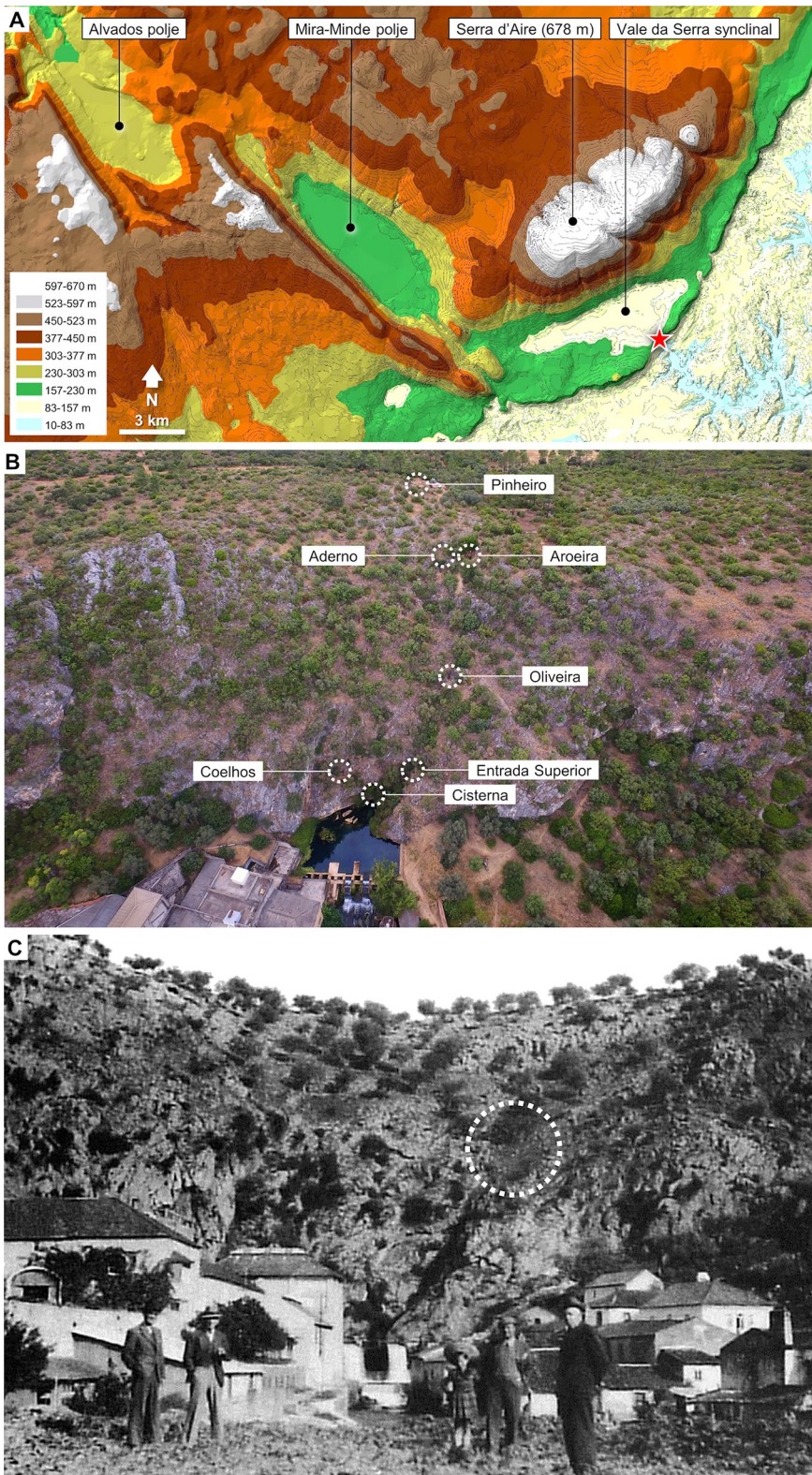

**Fig 1. Location. A.** Gruta da Oliveira in its geomorphological setting: the red star indicates the Almonda spring. **B.** Archaeological sites of the Almonda karst system discovered and archaeologically excavated since 1988. **C.** 1940s image of the Almonda scarp face, showing where the receding slope exposed the Gruta da Oliveira's infilling, since concealed by vegetation growth (dotted circle); after reference [31].

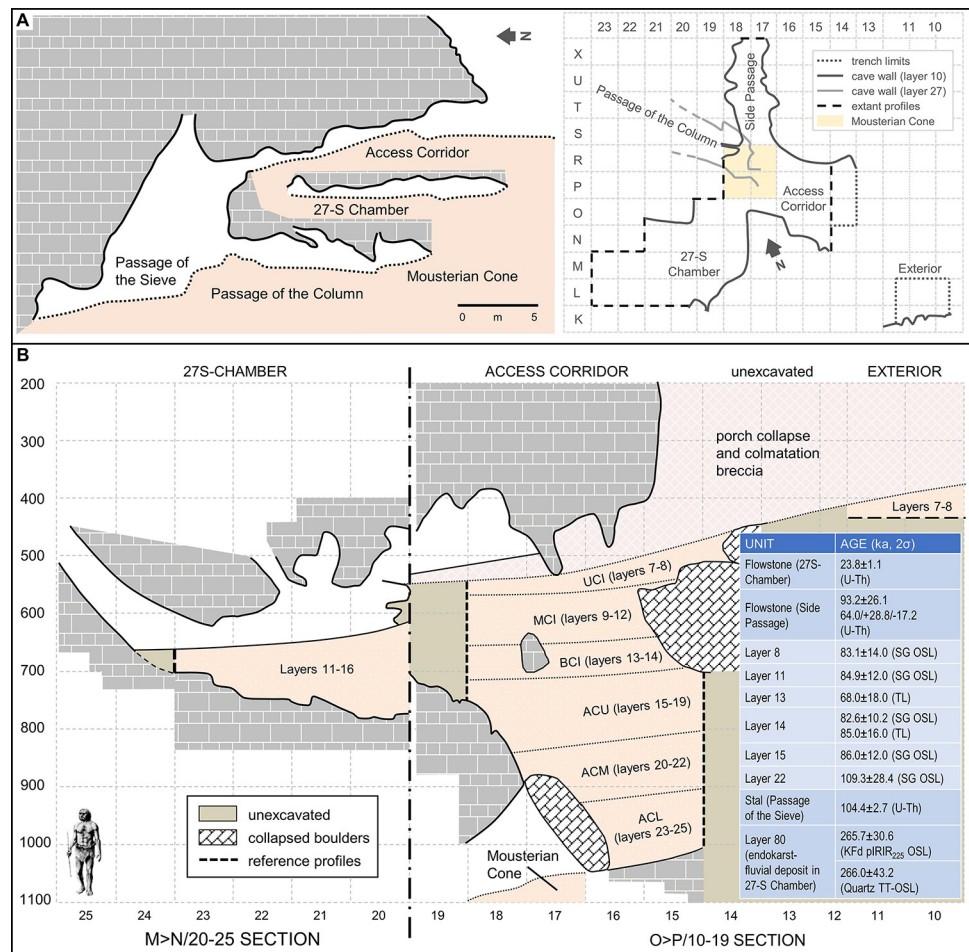

**Fig 2. Stratigraphy and plan. A.** Schematic N-S section of the Almonda scarp face at the elevation of the Gruta da Oliveira, showing the different passages and the site's excavation grid (site plan outline drawn at the base of layer 10). **B.** Schematic stratigraphic profile (elevations in cm below datum) and summary of luminescence and U-Th results for the archaeological sequence and the speleothems that constrain its age (2σ uncertainties); the offset between the two sections is an artefact of their being 2 m apart along the x-axis of the excavation grid. The acronyms refer to the ensemble framework (see text, section 3.4.1.): UCI, Upper Cave Interior; MCI, Middle Cave Interior; LCI, Lower Cave Interior; ACU, Access Corridor Upper; ACM, Access Corridor Middle; ACL, Access Corridor Lower. After reference [63].

Palaeolithic stone tools (Fig 3). Previous studies of the site include a geoarchaeological analysis of the sedimentary infill as known until 2008, revealing the good preservation of the archaeological contexts found therein [64], and a taphonomic analysis of the stone tools' spatial distributions, revealing the stratigraphic integrity of the lower part of the succession [65]. The provenience of raw materials, the technology and typology of stone tools, the taphonomy and composition of the faunal assemblages (including micromammals), and the Neandertal human remains have also been published [65–72].

Here, we describe the stratigraphic units recognised post-2008 and provide a complete, overall view of site formation under a geoarchaeological perspective. We combine this evidence with the micromorphological analysis of in situ fire features identified during the 2004 and 2010 field seasons to demonstrate a regular, continued use of fire across the time span represented by the archaeological deposit. We further show that our findings are consistent with the occupation patterns revealed by the spatial distribution of proxies for the on-site activity of

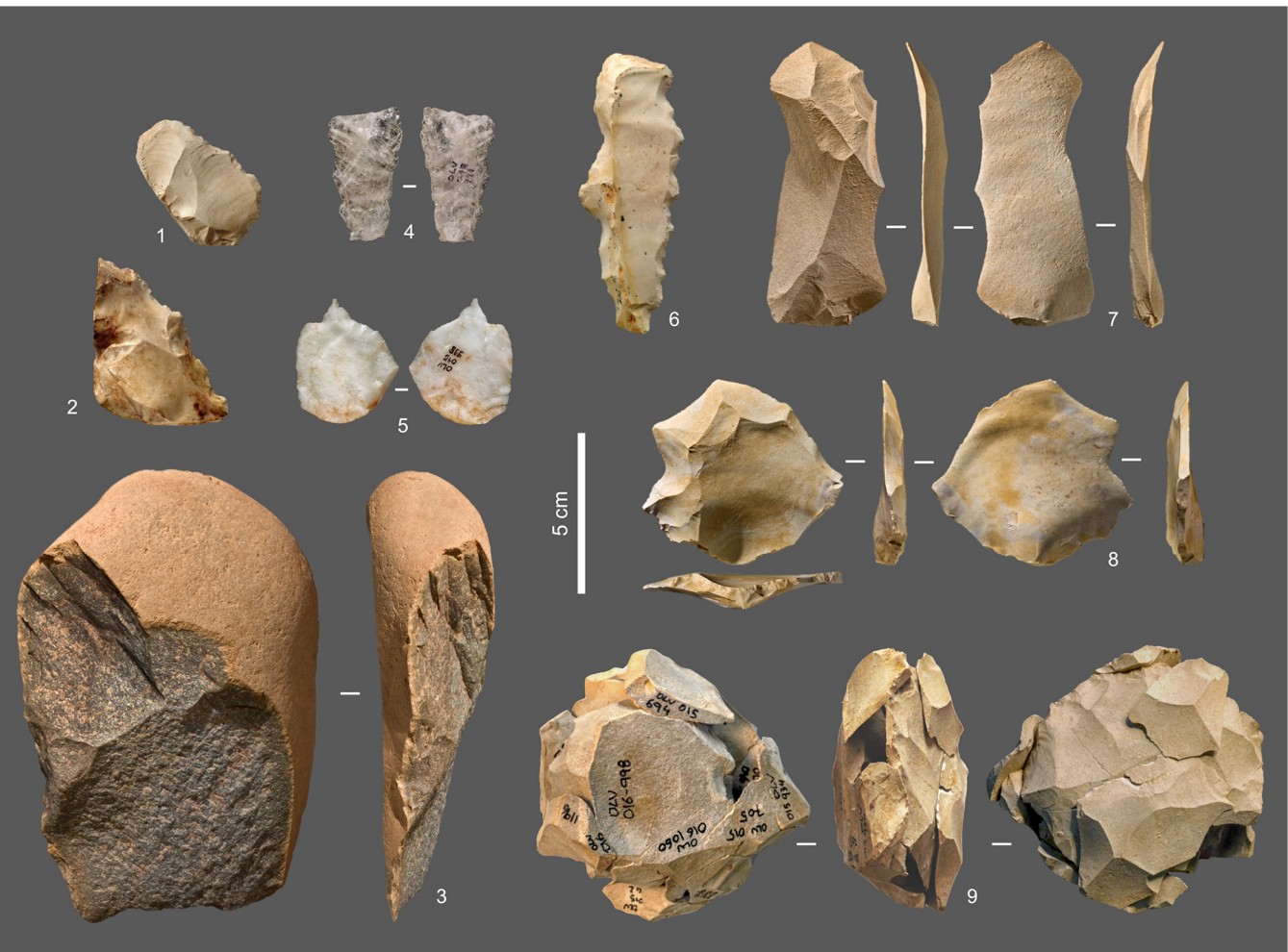

**Fig 3. Stone tools. 1–2.** Denticulated sidescrapers; **3.** Cleaver; **4.** Laminar flake; **5.** Perforator; **6.** Denticulate; **7.** Laminar Levallois flake; **8.** Levallois flake; **9.** Refitted Levallois core (**1–2., 6–9.** Chert; **3.** Quartzite; **4–5.** Quartz).

carnivores (coprolites) and humans (burnt bones and stone tools), and we assess whether such patterns' change through time relates to environmental change at the local (i.e., at the site itself), the regional, and the global scales.

## 2. Materials and methods

As detailed elsewhere [64, 65], the site was divided in one square meter grid units (Fig 2A). Finds were piece-plotted against the grid and an arbitrary elevation datum set at 117.267 m asl (above modern sea level). The minimal unit used in the excavation was the spit (*nível artificial*) —a slice of deposit cut to constrain within tighter limits the vertical position of the non-piece-plotted or sieve-collected finds. Spit thickness varied between a maximum of 20 cm, for beds primarily made of large éboulis, and a minimum of 5 cm, when dealing with finer sediments with potentially good preservation of spatial distributions. The basal surface of spits followed the general dip of the deposit as observed at a given elevation and always respected strati-graphic boundaries. Spits were numbered sequentially within each grid unit, from 1 to *n*, pre-ceded by the letter A (e.g., spit A1, spit A70, etc.).

Individual stratigraphic units were designated as layers, which are the equivalent of the Geoarchaeological Field Unit (GFU) as theoretically defined in the context of the excavation of the Lagar Velho rock-shelter (Leiria, Portugal) [73]: three-dimensional bodies composed of sediments, of either natural or cultural origin, that (a) feature characteristics differentiating them from surrounding ones and suggesting change in the dynamics of the accumulation (e.g., base-of-boulder planes defining the cave floor extant at the time of roof-collapse events, flowstone-covered or calcite-incrusted surfaces, increases in the clay content of the matrix and associated colour changes), and (b) can be followed across significant horizontal extents. As some thusly defined units span the three main areas of the cave interior (Fig 2A), the different sequences excavated in each area could be laterally correlated and integrated into a single stratigraphic succession scheme.

To describe the different stratigraphic units, field observations were combined with extensive examination of all extant stratigraphic profiles, which considered sedimentary, soil/diagenetic, and archaeological features. The units of stratification defined during fieldwork were retained, and further subdivided when, in profile view, significant vertical variation was apparent.

Twenty undisturbed samples were collected for micromorphological observation during the field seasons carried out between 2001 and 2012—thanks to good sediment cohesion, mostly by simple extraction, without the use of Kubiëna tins. Large-sized thin sections were prepared at the laboratory *Servizi per la Geologia* (Piombino, Italy), trough impregnation (using a mixture of resin, styrene, and hardener), curing, cutting into cm-thick slabs and eventual preparation of 25 μm thin sections of either 95 × 55 mm or 55 × 45 mm. Ten have been previously described [64], the other ten are presented here (Table 1).

Thin sections were observed under the polarizing microscopes of the *Laboratorio B. Bagolini* (University of Trento, Italy) and of the AMBI-LAB (University of La Laguna, Spain) at magnifications comprised between 20× and 1000×, using plane-polarised light (PPL), crossed-polarised light (XPL), and oblique incident light (OIL), the latter for observation in standard lighting conditions and for primary fluorescence. Fluorescence observation used two distinct wideband filter combinations: ultraviolet and blue (with excitation filters between 330–335 and 420–480 nm, respectively, and suppression filters at 420 and 520 nm, respectively). Images were captured with a digital camera for polarising microscopy. Description follows the guidelines proposed in references [74–76], with integrations for anthropogenic features based on references [77–79].

**Table 1. List of thin sections analysed in this study.**

| # | Label | Year of collection | Provenance | Unit | Remarks |
|---|-------|-------------------|------------|------|---------|
| 1 | **OLV8-1** | 2008 | cross-section NOP15-E | **16** | |
| 2 | **OLV8-2** | 2008 | cross-section NOP15-E | **16** | lower part of unit 16 |
| 3 | **OLV8-3** | 2008 | cross-section NOP15-E | **17** | upper boundary of unit 17, strongly bioturbated |
| 4 | **OLV8-4** | 2008 | excavation surface, grid square O15 | **18** | crust on top of unit 18, partly phosphatised |
| 5 | **OLV9-1** | 2009 | cross-section NOP15-E | **19** | |
| 6 | **OLV1201** | 2012 | excavation surface, grid square R19 | **27** | includes crusts on top of unit 27 |
| 7 | **OLV1202** | 2012 | cross-section NOP15-E | **24** | strongly cemented |
| 8 | **OLV1203** | 2012 | cross-section NOP15-E | **22 (up)** | strongly cemented; sub-units 22/1 & 22/2 (upper part) |
| 9 | **OLV1204** | 2012 | cross-section NOP15-E | **22 (low)** | strongly cemented; sub-units 22/2 (lower part), 22/3 & 22/4 (top) |
| 10 | **OLV1205** | 2012 | cross-section NOP15-E | **20** | strongly cemented |

Undisturbed samples were collected from the Access Corridor except for OLV1201, which comes from the Passage of the Column. All thin sections measure 9.5 cm x 5.5 cm and were prepared at the laboratory *Servizi per la Geologia* (see text).

All necessary permits were obtained for the described study, which complied with all relevant regulations. The animal and human bone remains from the Gruta da Oliveira are deposited in *Museu Nacional de Arqueologia* (Lisbon, Portugal), where they can be accessed by request to its director. The stone tools are housed at *Faculdade de Letras da Universidade de Lisboa* (Lisbon, Portugal), where they can be accessed by request to the director of UNIARQ (*Centro de Arqueologia da Universidade de Lisboa*). The individuals seen in figures relating to the reported research are volunteers who gave fully informed consent for the photographs to be taken and used for the illustration of the excavation work in an academic context, including research and publication.

## 3. Results

### 3.1. Cave morphology

The collapsed porch corresponds to the Exterior area of the site (grid units K-M/9-12). Here, the base of the *c.* 2.5 m-thick brecciated éboulis sealing the archaeological deposit lies at *c.* 113.5 m asl, *c.* 40 m above the extant spring. In 1991, that deposit was tested to a depth of *c.* 50 cm. In the following year, realising that, in this location, at the very edge of the scarp face, removal of the overburden faced unsolvable logistical problems, the decision was made to restrict the investigation of the site to the cave's interior. An access trench was cleared and engineered for safety, and the excavation of the under-roof deposit comprised between the Exterior and the Passage of the Sieve (*Galeria do Crivo*) was initiated, gradually revealing the complex morphology of the infilled passages (Fig 2).

The Passage of the Sieve provided the discovery route via which the site was first identified; it develops at *c.* 95 m asl, connecting the cave porch with the deeper karst along a major N-S fault that, as one approaches the scarp face, branches into a number of interconnected passages formed at different elevations of a system of orthogonal joints. The narrowness of most rendered further exploration impossible, but three—the Passage of the Column (*Galeria da Coluna*), the 27-S Chamber (*Sala 27 de Setembro*), and the Side Passage (*Divertículo Lateral*)— provided speleological approaches to the inner rim of a sedimentary talus extending across the Access Corridor (*Corredor de Acesso*) and blocking further progression.

The Passage of the Column opens *c.* 10 m above the floor of the Passage of the Sieve and develops at *c.* 105 m asl. At the time of discovery, the Passage of the Column ended in an accumulation of sediment with apex at *c.* 107 m asl—the Mousterian Cone (*Cone Moustierense*; subsequently excavated as layers 26–27 of the site's stratigraphic succession). This deposit abutted a roof formed by the underside of a large, parallelepipedal boulder that, even though obstructing communication with the Access Corridor above, was insufficient to fully block the gravitation of sediment and finds via fissures or via the voids left between sediment and cave walls (Fig 4A, 4B). Eventually, stone tool refitting confirmed that the Mousterian Cone's artefacts and animal bone did derive from the overlying Access Corridor [65], thereby corroborating the hourglass formation model originally put forth [26, 27].

The excavation of the Access Corridor trench (grid units N-R/15-18) exposed smooth, vertical or subvertical, parallel cave walls bounding a straight, E-W-oriented, *c.* 4 m-long stretch of the former outlet's terminal meander (Fig 4C). Here, support for the sedimentary infill was variously provided: along the grid's O-P lines, by the large rock mass and associated éboulis that obstructed communication with the Passage of the Column; along the grid's N line, by a bedrock floor lying at about 107.5 m asl; in the grid's 17–19 rows, by an exteriorly sloping, karren-like rock surface extending down from the floor of the adjacent 27-S Chamber.

The 27-S Chamber (grid units L-O/19-24) opens onto the SW corner of the Access Corridor and corresponds to the enlargement of a stratification joint; it forms a *c.* 4×3 m space with

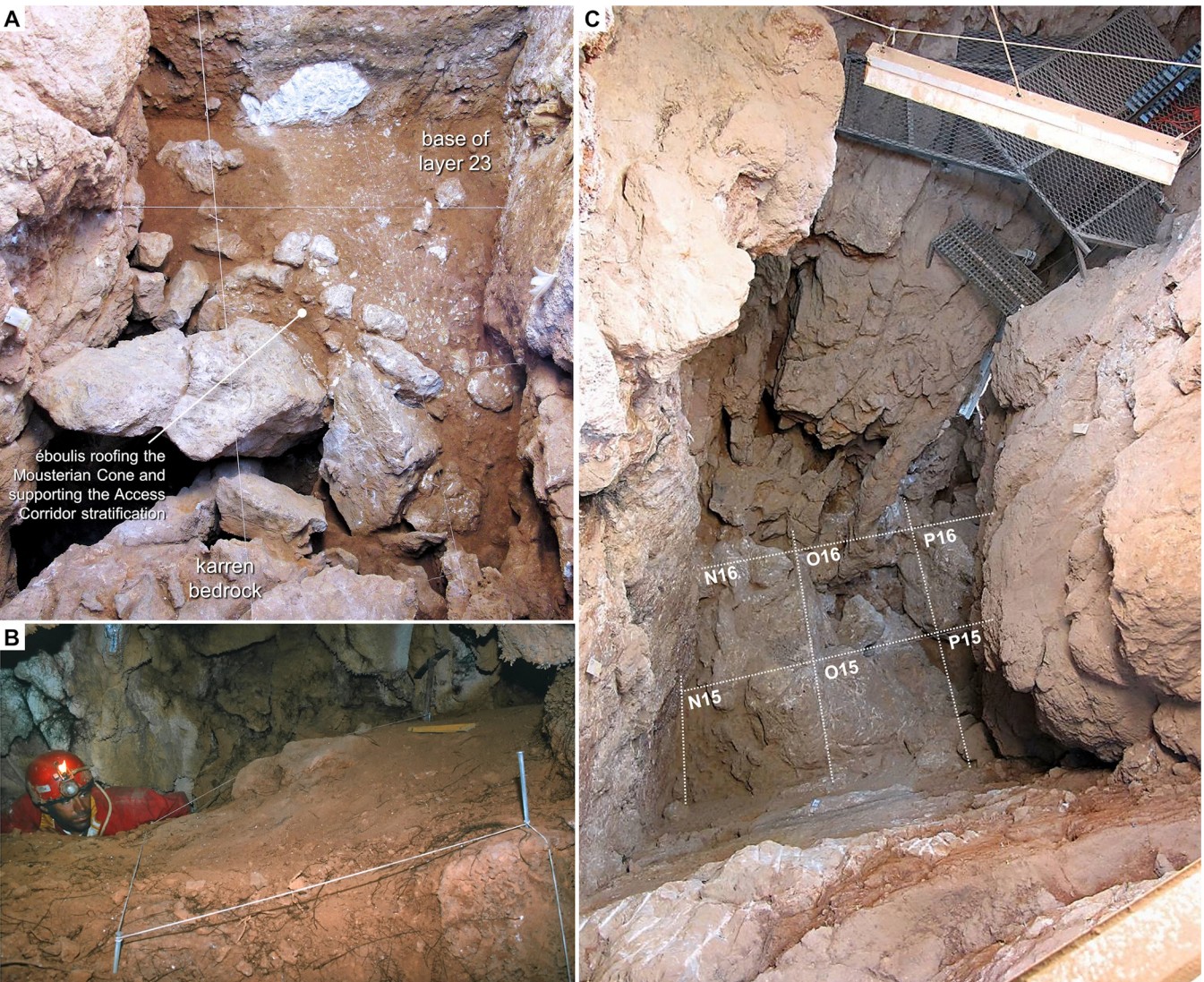

**Fig 4. Access Corridor. A.** The basal, infilled éboulis (layers 23–25) supporting the overlying stratification, during excavation (2011). **B.** The Mousterian Cone at the time of discovery (1989; after reference [31]). **C.** Zenithal view of the bedrock in the Access Corridor area at the end of the excavation (2012), exposing the connection with the Passage of the Column (in grid unit P17 and beyond).

an irregular roof at *c.* 112.5 m asl and a karren-like floor at *c.* 110 m asl (Fig 5). At a slightly higher elevation (*c.* 111 m asl), a ninety-degree step in the opposite, NW corner of the Access Corridor leads to the Side Passage (grid units S-X/17-19) (Fig 5). This narrow and low, *c.* 2 m-high diverticulum features a chimney rising to *c.* 115 m asl (i.e., to no more than one or two metres below the surface) and communicates with the Passage of the Sieve through an opening that leans over the latter's floor from a height of *c.* 20 m.

## 3.2. Stratigraphy and field characteristics of the infilling

Three main stratigraphic complexes have been defined [64]: (a) the collapse sealing the cave (layers 1 to 6), which extends across the Exterior and Access Corridor areas (Fig 2); (b) the "Middle Palaeolithic sequence" (layers 7 to 25), formed of fine, mostly reddish-brown sediment that rather regularly alternates with flowstones and carbonate crusts (Figs 6–8 illustrate

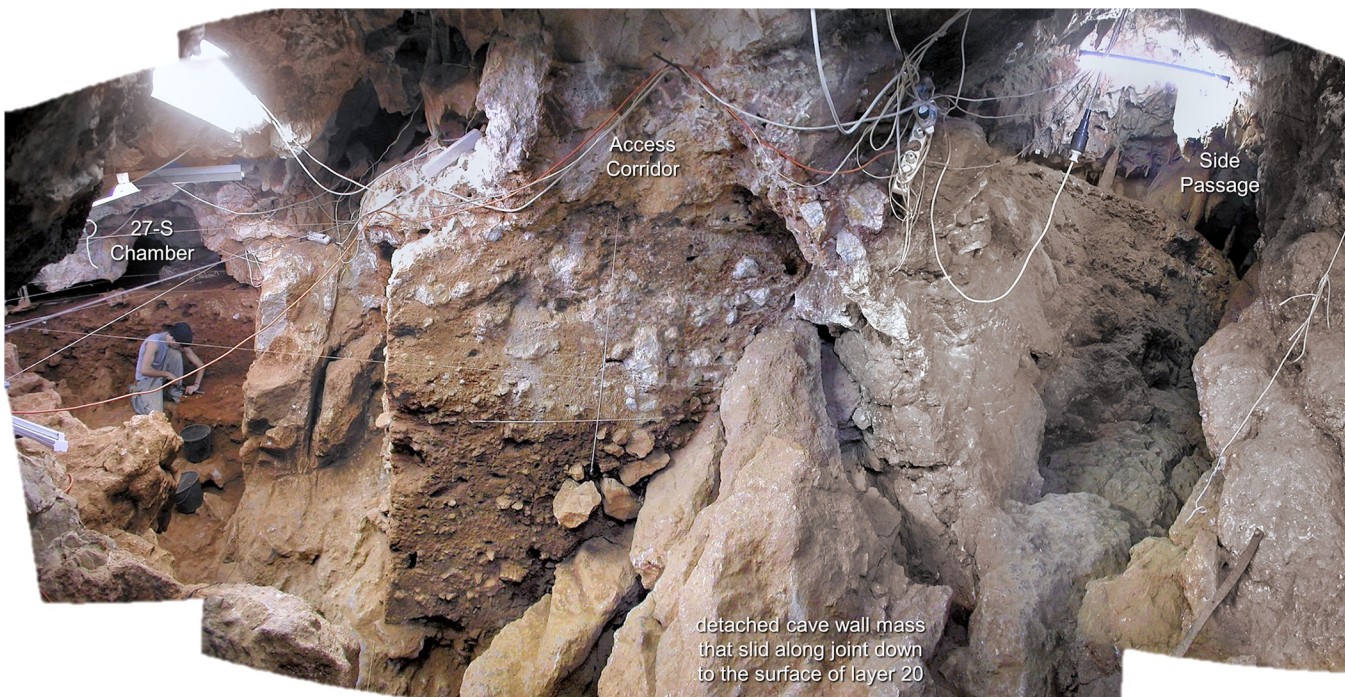

**Fig 5. Cave Interior.** The 27-S Chamber and the Side Passage as seen from the Access Corridor during the 2006 field season.

the reference profiles spanning this complex in the upper part of the Access Corridor, and, down to bedrock, in the Side Passage and the 27-S Chamber); and (c) the remnant of fluvial sediment (layer 80) preserved in the karren-like depressions of the 27-S Chamber (Fig 7).

The "Middle Palaeolithic sequence" complex features two major discontinuities. At the interface between layers 14 and 15, a clear, linear boundary is associated with carbonate incrustation (as observed during excavation in rows 17–19) and flowstone formation (in rows 14–15, under a major roof-collapsed boulder; Fig 2). At the interface between layers 19 and 20, a similar boundary was found in association with (a) major rock fall, including the detachment of a huge chunk of cave wall that slid along a joint down to the then-extant cave floor, i.e., down to the surface of layer 20 (Figs 5, 8), and (b) extensive carbonate incrustation along the grid's N line (Figs 9, 10). Bayesian modelling of available dating results suggests that the flowstone and carbonate incrustations capping layers 7, 15, and 20 correspond to local manifestations of GI (Greenland Interstadial) 19, 21, and 22, respectively, placing layers 7–14 in MIS 5a (*c.* 71–85 ka) and layers 15–25 in MIS 5b (*c.* 85–93 ka) [63].

At the time of reference [64]'s report, the excavation of the Access Corridor had reached the base of layer 19; it was not until the four seasons of fieldwork carried out between 2009 and 2012 that the underlying units, layers 20–25, were recognised, and that open-space connection with the Mousterian Cone and the Passage of the Column was re-established. This fieldwork confirmed that (a) unlike layers 11–14, which are broadly horizontal across most of their excavated extent, layers 15–25 dip NW, and (b) like layers 15–19, layers 20–25 are made up of sediment washed in from the slope. The archaeological content of layers 15–25 is also primarily made up of material syn-depositionally derived from activity areas located outwards, in the then-extant cave porch, with two exceptions, layers 15 and 20–22: during their formation, the latter harboured in situ human activity in, respectively, the 27-S Chamber and the Access Corridor, as corroborated by stone tool refitting [65]. Layers 21–22 featured well preserved

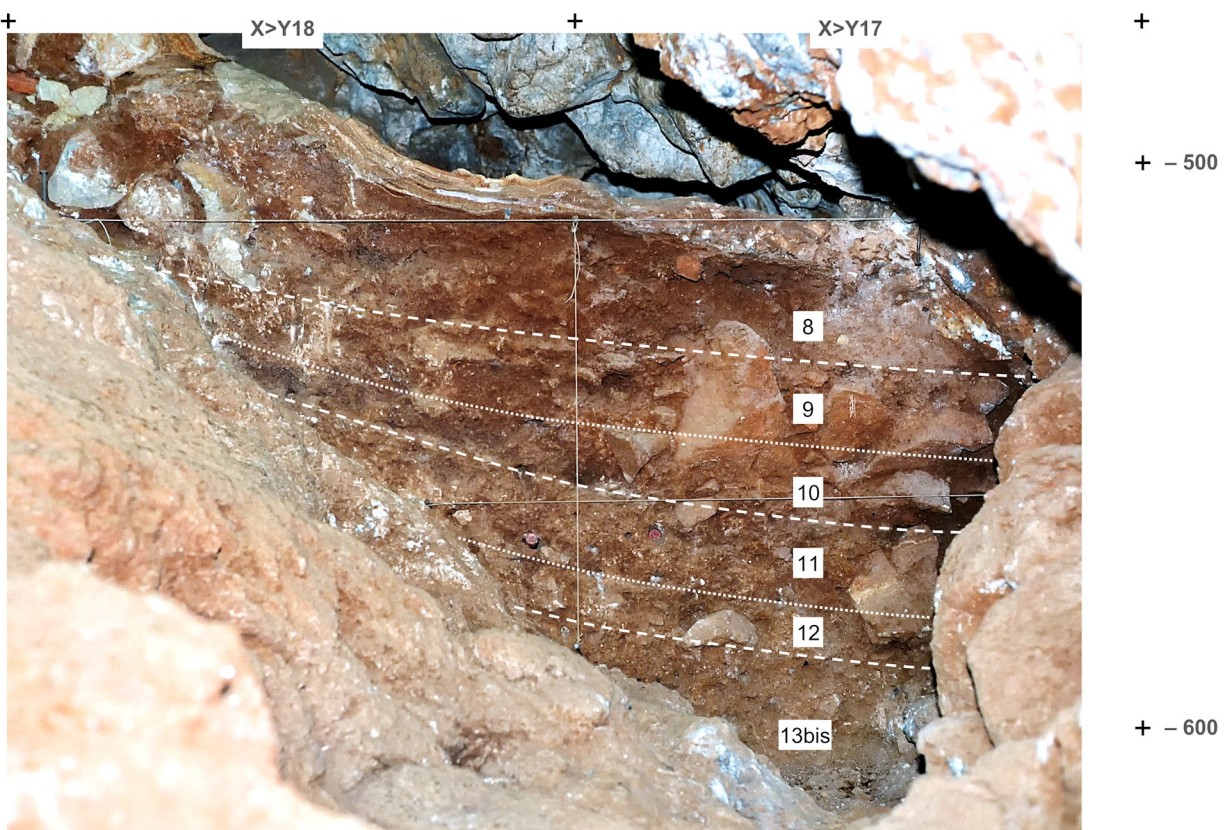

**Fig 6. Upper Cave Interior stratigraphy.** Orthorectified photo of the Side Passage profile (X>Y17-18). Elevations are in cm below datum.

hearths, one fully excavated, in layer 21 (Fig 11), and two superimposed ones, cut by the N-P/15>14 profile, in layer 22 (Figs 9, 10; see below).

Local post-depositional disturbance, in particular along the walls and near large boulders, is ubiquitous. However, as shown by stone tool refitting [65], the impact of such processes varies significantly across layers 15–25. Overall, the degree of stratigraphic integrity is high, especially with regards to layers 15, 21, and 22; however, a not insignificant incidence of vertical, mostly downward displacements was observed in layers 16–19, 20, and 23–25.

Table 2 provides descriptions of the units recognised across the succession in its entirety, from the colmatation éboulis to the basal alluvium. By comparison to the units that reference [64] reported on, layers 20–25 are not much different: mainly silty, with colours ranging from reddish brown to light reddish brown, often massive, and including variable quantities of limestone fragments detached from the walls and roof of the cave (usually more abundant near the walls). Separation between the different units is often provided by flowstone or carbonate crusts, which become powderier and more weathered from top to bottom. Diagenetic features related to secondary phosphate accumulation and to the chemical reaction between calcium carbonate and phosphate also become more pronounced as one moves downward—to such an extent that naked-eye detection was possible during the excavation itself, not just under the microscope (see below).

Layer 25 is a thick accumulation of angular, randomly arranged limestone boulders, locally cemented by calcium carbonate; this éboulis fills the complex volumetry defined by the cave walls and the deeply crevassed, karren-like relief of the Access Corridor's bedrock. Layer 26 is

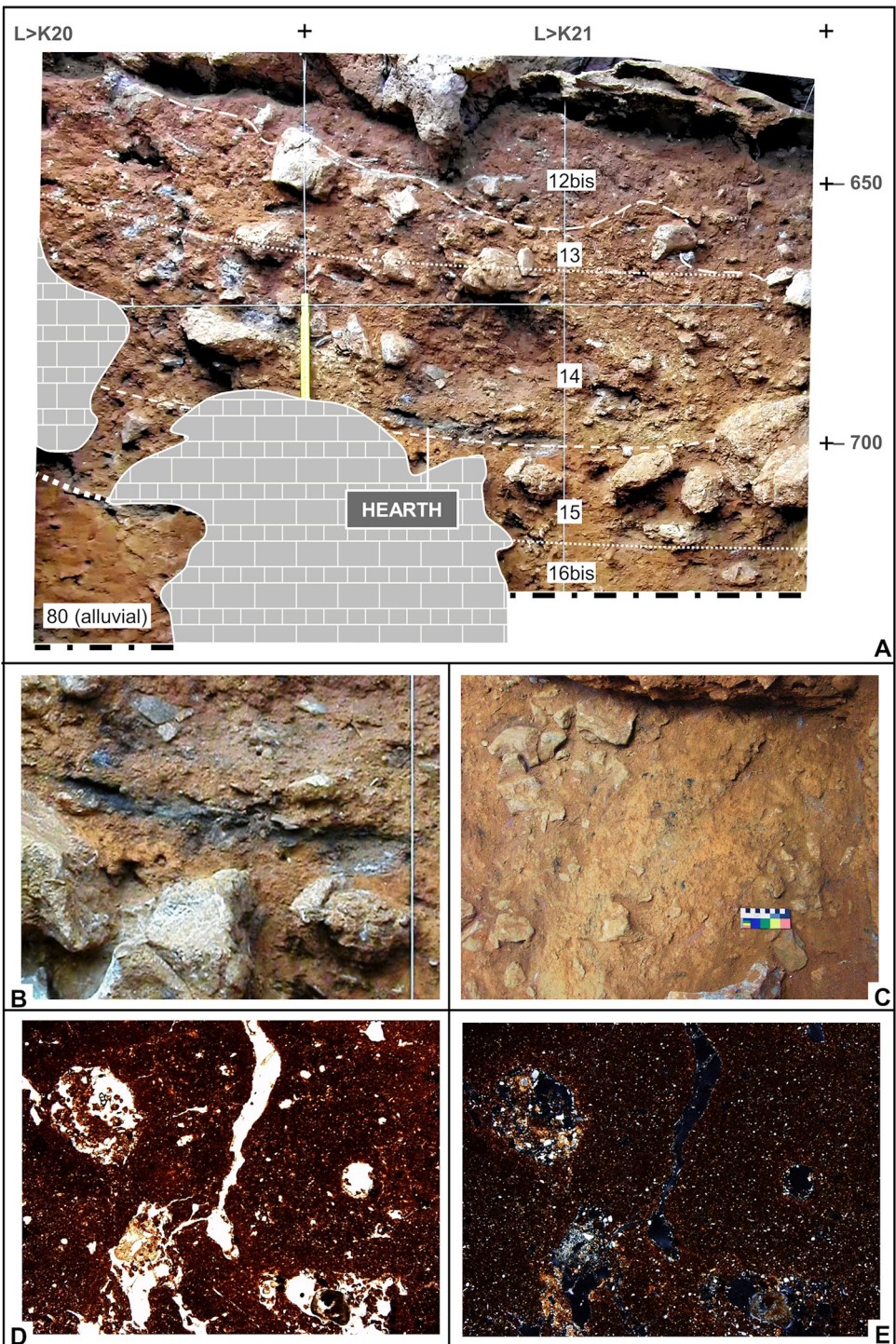

**Fig 7. Basal and Middle Cave Interior stratigraphy. A.** Orthorectified photo of the 27-S Chamber profile (L>K20-21); elevations are in cm below datum. **B-C.** The hearth feature at the base of layer 14 in close-up, cut-by-profile view (**B**), and the immediately underlying burnt, brick-coloured surface, in zenithal view (**C**). **D-E.** Micrographs from the sediment reddened by the hearth, showing the groundmass under PPL (plain polarized light; **D**) and XPL (cross polarized light; **E**). Note the red colour of the fine material (PPL) and the undifferentiated b-fabric (XPL), with interference colour masked by amorphous iron oxide produced by thermal impact (the porosity and pedofeatures are post-depositional and relate to biological activity). Both micrographs are taken from thin section OLV0608 ([64]); width of frame = 8 mm.

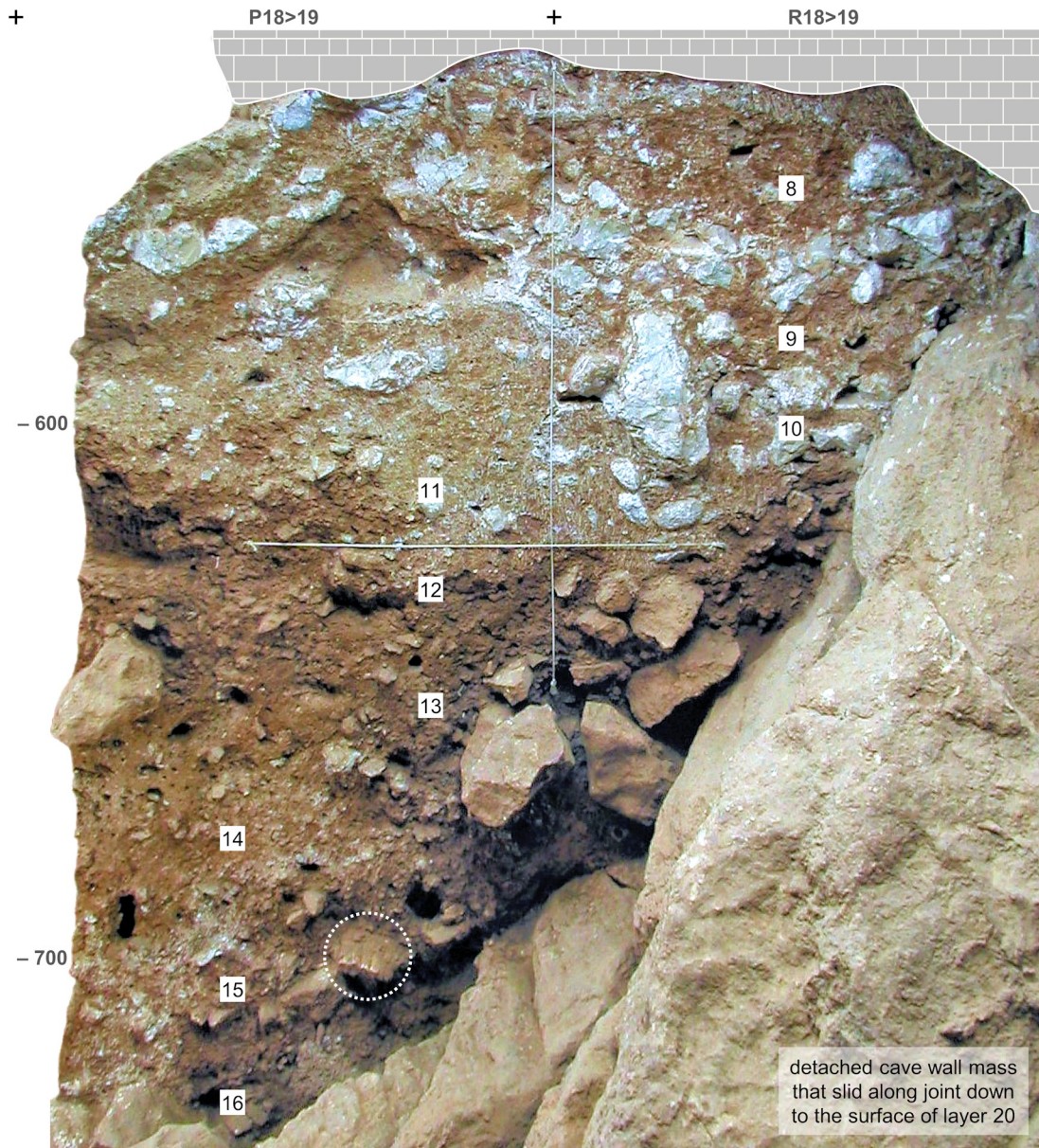

**Fig 8. Cave Interior stratigraphy at the back end of the Access Corridor.** Orthorectified photo of stratigraphic profile P-R18>19 (in the baulk occupying the centre of Fig 5) The dotted circle indicates a rhino mandible outcropping from the profile at the top of layer 15. For detailed interpretation, see ([64]: Fig 9, Table 2). Elevations are in cm below datum.

the Mousterian Cone and layer 27 is the deposit found beneath, forming the Passage of the Column's sediment floor (Fig 12). Both the matrix and the archaeological content of layers 26–27 correspond to infiltrations from the overlying sedimentary column filling the Access Corridor up, as inferred at the time of discovery (and since corroborated by the U-series dating of speleothems formed atop of layer 26 and 27 to, respectively, the recent Holocene and the Last Glacial Maximum; [63]). This post-depositional process suffices to explain the stratigraphic paradox (i.e., the fact that the Passage of the Column's deposit represents a more recent accumulation than the Access Corridor's, despite the latter's lying directly above).

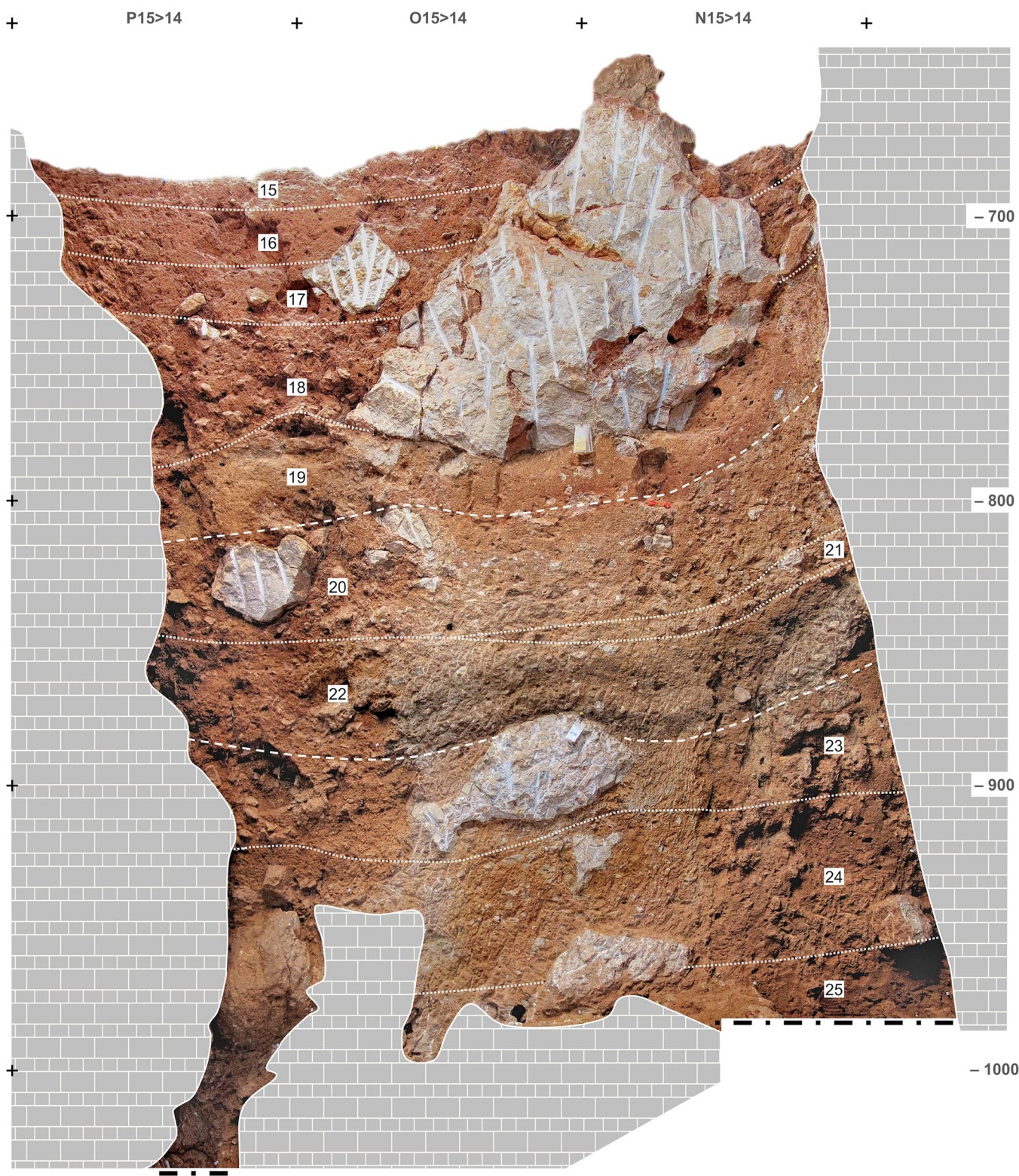

**Fig 9. Access Corridor stratigraphy.** Orthorectified photo of stratigraphic profile N-P15>14. Elevations are in cm below datum.

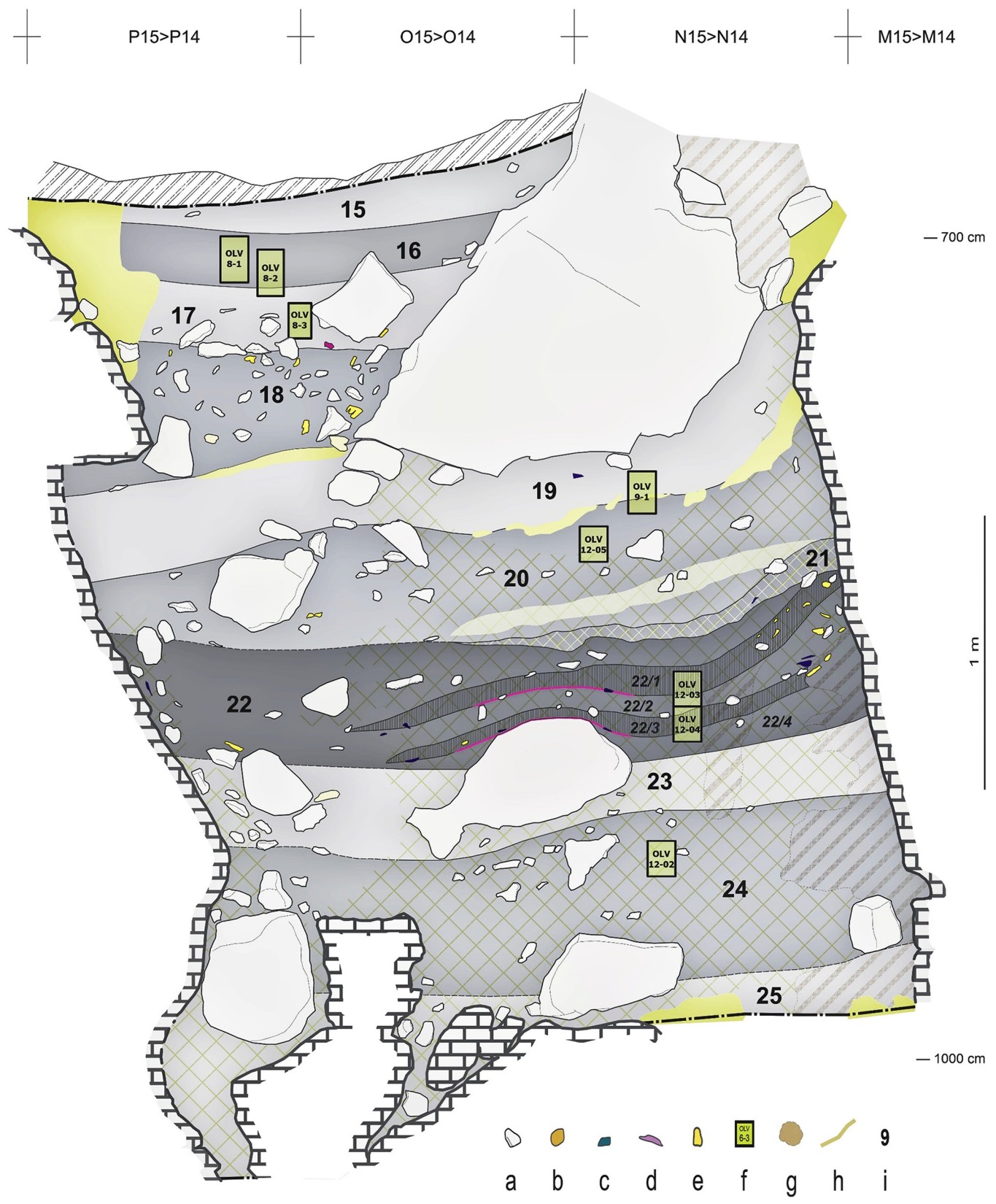

**Fig 10. Access Corridor stratigraphy.** Interpretative drawing of stratigraphic profile N-P15>14. Key: **a.** stones; **b.** coprolites; **c.** lithic artefacts, chert **d.** lithic artefacts, quartzite; **e.** bones **f.** undisturbed samples for soil micromorphology; **g.** burrows; **h.** secondary accumulation of calcium carbonate; **i.** unit designations. Elevations are in cm below datum.

The existence of the void partially filled by the Mousterian Cone and the hourglass formation process at work in this part of the cave through the accumulation of layers 15–25 led to the development of a sort of localised, weakly developed, covered sinkhole: coupled with the deposit's cohesive mechanical behaviour and moisture content, the loss of mass caused by suffosion (see, e.g., reference [80]) led to the deformation of the sedimentary column above, which sudden movements related to seismic events or to the collapse of massive boulders may also have contributed to. Hence, the sinking of the basal parts of the stratification in line P of the grid, along the north wall of the passage (Figs 9, 10).

The fact that no refits have so far been found linking layers 26–27 with layer 14 or any other layer further up in the "Middle Palaeolithic sequence" complex suggests that, (a) the Mousterian Cone finds derive entirely from layers 15–25, and (b) by the time the accumulation of layer 15 came to an end, overall stabilisation of the sedimentary body had been achieved. Thus, where the basal part of the Access Corridor succession is concerned, subsequent modifications would seem to have been limited to the precipitation of calcite from rainwater seeping along cave walls and roof fissures, leading to the development of flowstone and stalagmite atop the stabilised surfaces of the Mousterian Cone and the Passage of the Column, coupled with limited bioturbation by roots and rootlets.

Locally, layer 27 rests on a discontinuous thin layer of homogeneous silt, squeezed between this unit and the limestone bedrock. This silt film was named "layer 70" and may correlate, stratigraphically, to "layer 80", the alluvial sediment dated to within the 220–310 ka interval that fills depressions in the karren-line bedrock of the 27-S Chamber (Fig 7) [63]. Under the microscope, reworked fragments of such layer 80-like sediment were indeed observed within the groundmass of layer 27 (see below).

### 3.3. Archaeological micromorphology

**3.3.1. The Access Corridor infilling.** The thin sections from layers 20–27 display micromorphological characteristics that are quite similar to those reported in reference [64], except for the stronger development of pedofeatures in connection with the secondary carbonate and phosphate accumulation observed in basal layers 22–25. Only the sample from the Passage of the Column (Table 1) looks somewhat distinct. Some micromorphological characteristics (coarse components, microstructure, pedofeatures) are recurrent and can be described for all units together.

Under the microscope, coarse components are ubiquitous and cluster into the four groups described in reference [64]: SIL (Siliciclastic silt and sand fraction), LST (Limestone fragments), CRB (Other carbonate components), and ABC (Anthropogenic and biogenic components) (Table 3). The SIL group includes non-carbonate elements ranging from very fine silt to coarse sand (with occasional larger, ≤3–4 mm grains): their shape is variable (subangular grains are dominant), and their composition comprises monocrystalline quartz (dominant), feldspars (common to scarce; see Figs. 11A, 12H, and 13G of reference [64]), polycrystalline quartz (scarce), fine-grained chert grains (occasional) as well as pyroxenes, amphiboles and micas (rare). The CRB group includes calcite crystals (Fig 13A, 13B) and reworked fragments of speleothems or carbonate crusts. The LST category includes different types of limestone of diverse shape and size (mostly micritic limestone, see e.g., Fig 14C, with other types occurring occasionally), sometimes large (from 1 mm to several cm; Fig 14). The components of the

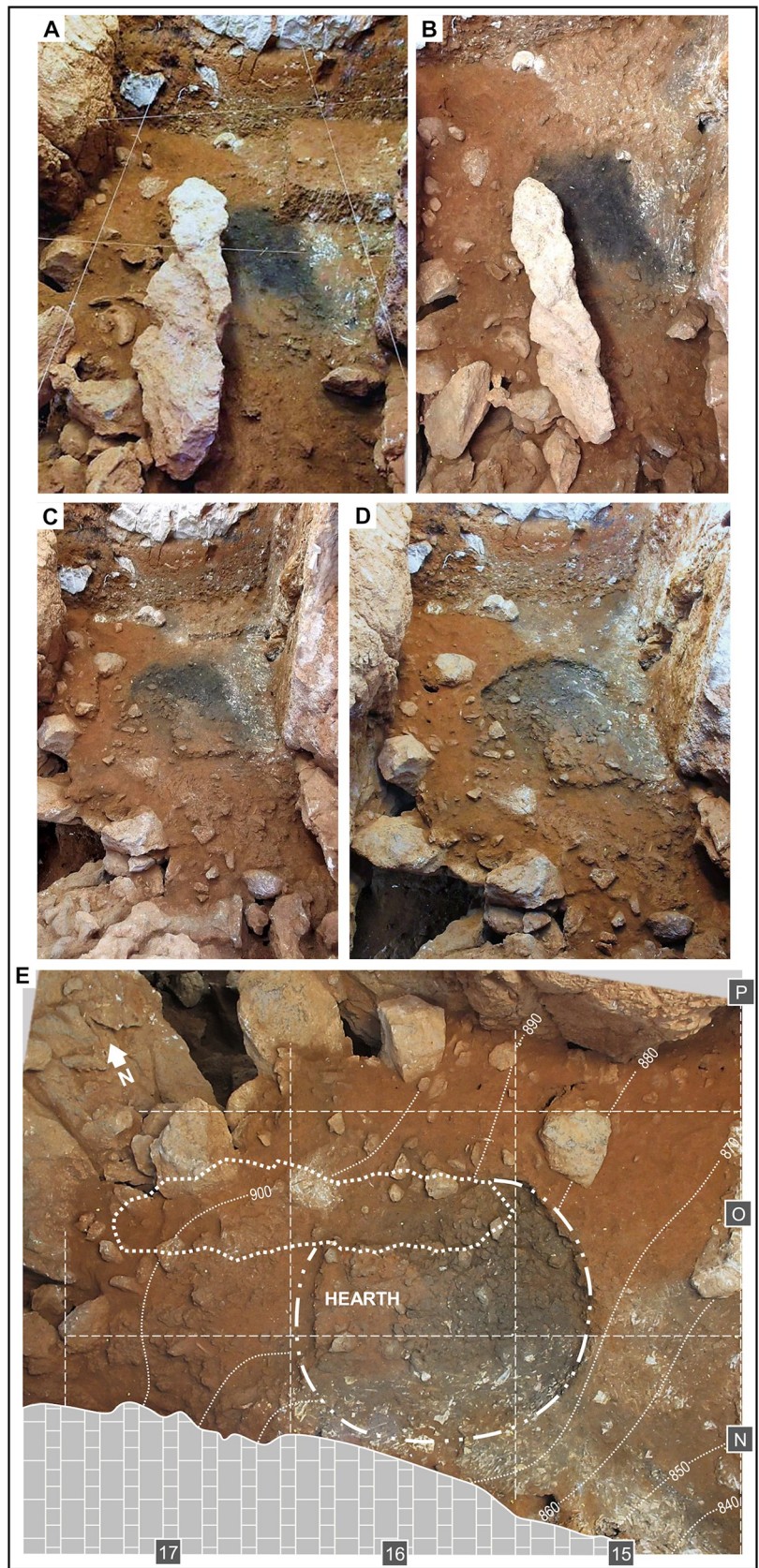

**Fig 11. The layer 21 hearth. A.** At the base of spit A66, a dark stain outcrops; in grid unit N15, in the upper right corner, spit A66 remained unexcavated at this stage. **B.** The contact between layers 20 and 21 (interface between spits A66 and A67) is now fully exposed; note the carbonate cementation of the surface in grid units N/15-16, along the wall, in marked contrast and significant discontinuity with overlying layer 20 (compare with panel **A**). **C.** During the excavation of spit A67, after removal of the large keel-like, roof-fallen boulder, the limits of the feature in row 15 had been determined, and its infill had been cut along the separation between grid units O16 and O15. **D-E.** Base of the A67 spit across the Access Corridor trench in oblique (**D**) and zenithal, orthorectified (**E**) views (in the zenithal view, the dotted line marks the outline of the overlying fallen boulder previously removed); the feature's infill has now been excavated, revealing a combustion basin cut into the cave floor; complete removal of the accumulation of collapsed boulders in grid unit P17 fully opened the connection with the Mousterian Cone below; between hearth and wall, layer 21 was shown to consist of a hard, carbonate-impregnated, phosphate-weathered breccia rich in animal bone remains. Elevations are in cm below datum.

ABC group occur throughout and, in some units (e.g., layers 12, 13, 14, and 22), can be particularly common. This group includes lithic artefacts (Fig 15) [81], bones and bone fragments of variable size, often displaying thermal alteration at maximum heating temperatures in the range of 500–600˚C ([82] (Fig 15B), and fragments of charred material. On average, the archaeological remains are well preserved and show no relevant weathering features.

Some components fall in none of the groups. There are fragments of phosphate composition that may have belonged to excrements and coprolites of the ABC group (e.g., Fig 12C-12E of reference [64]), and fragments of phosphatised limestone and of detached reaction rims (Fig 14C, 14D) [84], the latter more common towards the base. Occasionally, one also sees fragments of reworked soil material from outside the cave ('pedorelicts', *sensu* reference [85]; Fig 13E, 13F), as well as fragments of reworked fine alluvial sediment akin to layer 80 (Fig 16). All samples contain sand-sized crystals of gypsum showing traces of weathering (Fig 13C, 13D), but they are rare (most thin sections contain fewer than ten such crystals). The fine material is mostly speckled and usually exhibits a reddish-brown colour (Table 3).

Microstructure is poorly developed. Apedal microstructures are prevalent, with some samples occasionally showing poorly developed microgranular or subangular blocky aggregation. Porosity is on average low and with prevalence of channels, chambers, and vughs (Table 3). Concerning the groundmass, the coarse/fine (c/f) limit was fixed at 2 μm; the c/f ratio and related distribution patterns vary, and the b-fabric mainly ranges from undifferentiated to crystallitic (Table 4). Pedofeatures are also similar across the succession (Table 4). In the Access Corridor, the accumulation of secondary carbonate is widespread and, as noted in reference [64], implies the presence of carbonate-rich water seeping from the entrance and shafts above. This process explains both the cementation of layers 20–25 and the diversity of pedofeatures (Table 5), which mostly relate to accumulation in a vadose setting [86, 87] and include micrite hypocoatings (Figs 14A, 16A), micrite coatings—occasionally microsparite or even NFC (needle-fibre calcite) coatings—and micrite, sometimes microsparite, nodules (Fig 16B).

The secondary accumulation of phosphate, noted in all units, is common, and becomes more so towards the base of the succession, where it is also better expressed and can be easily recognised, whether by eye, with a "black-light" lamp, or based on the degraded, powdery consistence of the deposit's calcium carbonate component (limestone fragments and speleothems, in particular calcareous crusts), caused by carbonate-with-phosphate chemical reactions. At Gruta da Oliveira, the phosphate derives from biogenic inputs remobilised by infiltration water, and appears as nodules (see Fig 13E, 13F of reference [64]), sometimes as coatings or infillings (a few) and, especially, as the kinds of hypocoatings known as "reaction rims" (Fig 14; [84]). The development of the latter can be so intense as to concern the entire fragment (Fig 14D) and eventually lead to detachment of the rim under the impact of post-depositional disturbance processes (Fig 14C). Rims and nodules are particularly common in layers 22, 23, and 24, but detached rim fragments are found in the groundmass of almost all units (Table 4).

**Table 2. Field characteristics of the succession excavated in the Access Corridor and Passage of the Column.**

| Unit | Main characteristics |
|---|---|
| | **ACCESS CORRIDOR** |
| **1–6** | surface horizons and boulders sealing cave entrance |
| **k/7** | well-cemented carb. flowstone with laminar structure |
| **7** | calcareous breccia with sandy loam matrix |
| **8** | clayey silty loam; 4YR4/4; common limestone frs. with surface patina; poor to moderate carb. cementation and common carb. nodules; lb sharp, wavy |
| **k/9** | two superposed carb. crusts with laminar structure |
| **9-up** | silty loam; 5YR4/3 with 6YR4/2 mottles; common limestone frs. with surface patina, scarce calcareous mm-sized fraction, occasional calcite crystals; moderate carb. cementation; lb sharp wavy |
| **9-low** | clast-supported calcareous breccia with silty loam matrix; 5YR4/4; strong carb. cementation; lb clear |
| **k/10** | disc. carb. cementation, at places degraded carb. crust |
| **10** | clayey silty loam; 5YR4/4; few limestone frs., calcareous mm-sized fraction and calcite crystals; common coprolites and digested bones; moderate carb. cementation; lb clear |
| **k/11** | well-dev. continuous carb. flowstone |
| **11** | clayey silty loam; 6YR4/4; few to common limestone frs. (including frost slabs), scarce mm-sized fraction and speleothem frs.; patchy carb. cementation and few carb. pendants; few calcified roots; lb clear, with large limestone frs. |
| **k/12** | disc., moderately cemented carb. crust, degraded at places |
| **12** | silty loam; 6YR4/4; few to common limestone frs. (including frost slabs), scarce mm-sized calcareous fraction and speleothem frs.; carb. pendants and few calcified roots; lb clear, with limestone frs. |
| **k/13** | carb. crust |
| **13** | silty loam; 6YR4/4; common limestone and speleothem frs., scarce mm-sized fraction formed of limestone frs. and bone frs.; platy elements dip SW; patchy carb. cementation, few carb. pendants and calcified roots; darker areas (6YR3/3) with organic matter, ash, charcoal, and phosphate; lb clear |
| **k/14** | carb. crust |
| **14** | silty loam; 6YR4/4; very few limestone frs.; poorly dev. granular structure; disc. carb. cementation, few calcified roots; lb clear linear |
| **14-base** | silty loam; 5YR4/4; common limestone frs. and speleothem frs.; lb clear linear |
| *(no name)* | accumulation of boulders |
| **k/15** | degraded carb. crust |
| **15** | mm-thick sets dipping SW of clayey silty loam (on average); 6YR4/4; common limestone frs. (including frost slabs); degraded carb. crust between sets; open-work among larger stones; lb clear |
| **k/16** | thin cont. carb. crust, at places with laminar structure |
| **16** | silt; 5YR 4/5; few small limestone frs.; moderate carb. cementation; lb clear, linear |
| **17** | silty sand; 6YR 4.5/5; common limestone frs. and few speleothem frs.; weak lamination; lb clear, linear |
| **k/18** | well-cemented carb. crust |
| **18** | silty loam; 6YR 4.5/6; common limestone and speleothem frs.; weak lamination; lb sharp, erosive |
| *(no name)* | accumulation of boulders |
| **k/19** | slightly phosphatised flowstone with laminar structure |
| **19** | sandy silt; 6YR 4.5/6; scarce limestone frs. and speleothem frs.; weak carb. cementation (on top); dips inwards; lb clear, linear |
| **k/20** | phosphatised carb. crust |
| *(no name)* | accumulation of boulders |
| **20** | silty loam; 7.5YR5/4; few to common heterometric limestone frs., increasing downwards, with thin carb. and phosphate coating; massive; strongly cemented by carb. at its base; dips inwards (to Wc); lb clear |
| **k/21** | carb. crust |
| **21** | silty loam; 6YR5/4; few to common limestone frs.; strongly cemented by micrite; common lithics and bones; lb clear |

(*Continued*)

**Table 2.** (Continued)

| Unit | Main characteristics |
|------|----------------------|
| **22** | set of layers with dominant anthropic inputs and well-visible thermoalteration features (see text for details) |
| **23** | silty loam; 6YR5/6, slightly mottled (yellow or olive mottles); few limestone frs., fine (2 mm to 2 cm), often coated by phosphate, showing horizontal orientation pattern on top of unit; moderately to strongly cemented by carb.; lb clear, poorly distinct |
| **k/24** | phosphatised carb. crust |
| **24** | silty loam; 6YR5/7, mottled (yellow or brownish red mottles); common limestone frs., fine (2 mm to 2 cm), often coated by phosphate; occasional fine rounded non-carbonate granules (*cfr* fluvial remnant, unit 80); lb clear, poorly distinct |
| **25** | accumulation of dm- to m-sized angular limestone boulders; spaces are filled with sediment similar to unit 24, often enriched in phosphate, bridged by carb. cement (sometimes sparite) or empty; boulders are densely packed; where cemented, the unit appears as a concrete-like breccia |
| | **PASSAGE OF THE COLUMN** |
| **26** | clayey silt; 6YR5/7; common to frequent heterometric limestone frs., mostly angular, unweathered (loc. almost clast-supported, particularly at its base) and occasional small rounded non-carbonate granules (*cfr* fluvial remnant, unit 80 –see text); firm; moderate to strong carb. cementation; well-dev. layered flowstone on top (max. thickness 2–3 cm); abundant archaeological remains; lb abrupt to bedrock |
| **27** | clayey silt to (locally) silty clay; 4YR4/5; common heterometric (few mm to some dm) limestone frs., mostly subangular, unweathered, with some (clay?) coating but no phosphate coating, occasional fine rounded non-carbonate granules (*cfr* unit 80 –see text); massive, locally strongly cemented by carb.; sealed by layered flowstone on top (total thickness *ca.* 20 cm); common lithic artefacts, rare bones and charcoal frs.; lb abrupt to bedrock |
| **70** | thin intercalation of homogeneous silt without stones, 4YR4/6, loc. found between layer 27 and bedrock (*cfr* silt intercalations of unit 80 –see text) |

Key: carb.—calcium carbonate; cont.: continuous; dev.: developed; disc.: discontinuous; fr(s).: fragment(s); lb: lower boundary; loc.: locally.

Other pedofeatures are also worthy of notice: clay coatings (sometimes impure or dusty clay) occur in the upper part of the Access Corridor's succession, between layers 11 and 17; biogenic features such as infillings or passage features are widespread throughout and mostly relate to biological activity taking place during or soon after the accumulation of the sediment; pedofeatures related to the precipitation of Fe-Mn oxide (Fig 16C, 16D), both intrusive and impregnative, are present in the upper part of the succession (layers 11 to 18) and in layer 20. Finally, the occurrence of weakly preserved parallel, inclined lamination was detected in layers 16, 17, and 18 (see Fig 13H of reference [64]) (Table 4).

**3.3.2. The Access Corridor hearths.** Layer 21 is a hearth context excavated as a 10 cm-thick spit, A67. It consists of a large elliptical fire feature, almost 1.5 m in diameter, and associated finds, and first appeared as an extensive black stain outcropping under a large, keel-like, roof-collapsed boulder whose fall damaged the hearth's north-eastern edge. A massive, heavily cemented accumulation of bones extended across the space between the feature and the wall of the cave, in grid units N/15-16. The excavation of the feature's infilling—a dense mass of fine, blackish sediment rich in charcoal and burnt bone—revealed neat boundaries, defining a flat-based basin that, in grid units N-O/15, was cut into underlying layer 22 (Fig 11).

The floatation of the feature's infilling yielded extremely small and much altered charcoal fragments, among which only *Pinus sylvestris* and *Juniperus* sp. could be identified [88]. Quartzite thermoclasts, heated chert items, and a total of 263 burnt bone fragments were recovered in both the infill and alongside, in the surrounding surfaces. Most (84%) of the latter were black, grey, or white, indicating combustion at high temperatures, and a few were large enough to have been piece-plotted and taxonomically identified (to red deer, tortoise, rabbit,

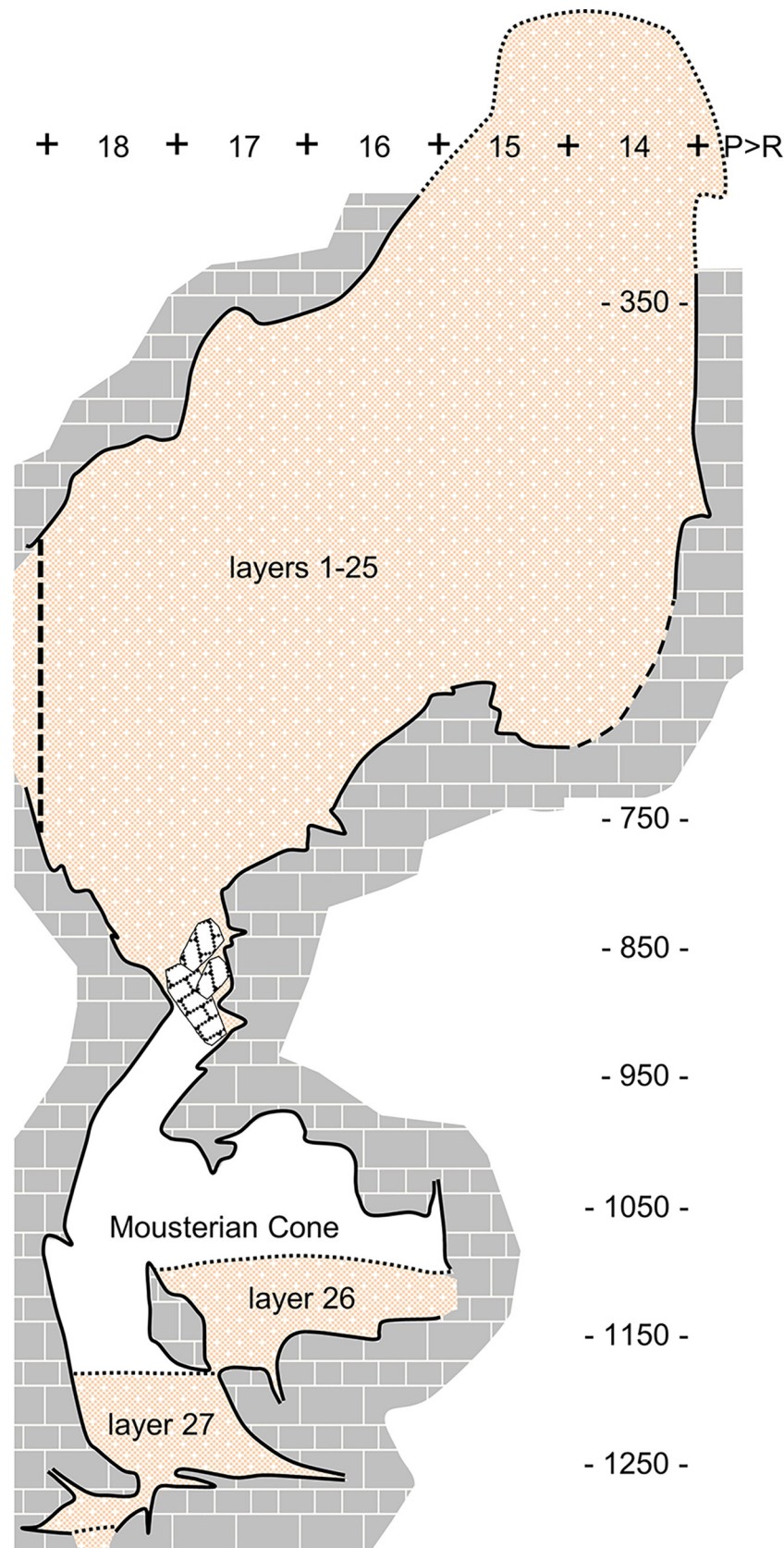

**Fig 12. Topographic profile of the Access Corridor.** Schematic rendering along the separation between the P and R lines of the excavation grid. The dashed line marks the position of the P-R18>19 stratigraphic profile (Fig 8). Elevations are in cm below datum.

and jackdaw). No sampling for micromorphology could be carried out, but these observations suffice to identify the feature as an in situ hearth that, as demonstrated by the scatter of the core and 17 chert flakes making up refit set 2000 [65] (Fig 3, no. 9), provided a focal point for flintknapping activities.

Layer 22 is a sequence of mainly anthropogenic sediment that yielded abundant archaeological remains. Despite the strong cementation affecting this part of the deposit, two accumulations of combusted by-products embedded in very dark grey sediment are apparent in profile view: sub-units 22/2 and 22/4 (Figs 9, 10). Directly under the latter, thin reddened belts could be observed. These two suffosion-deformed, superimposed features were only marginally cut by the excavation trench and so the samples for micromorphology analysis had to be taken directly from the N-P/15>14 profile (but, due to the presence of a large boulder below, to the edge of the visible rubefaction lenses; see Fig 10 for sample positions).

Under the microscope, the thin sections of all sub-units defined in layer 22 (Fig 15) contain common to frequent human inputs (bones, burnt bones, lithic artefacts, charcoal, and phosphate fragments); pedofeatures related to secondary carbonate and phosphate accumulation are also common (Table 4). Randomly arranged anthropogenic components (ABC group, see above) are seen on top of sub-unit 22/1 and lying horizontally at the base of sub-units 22/1 and 22/3. In the latter, thin layers of burnt bones, some large, some much reduced in size, all with intense thermal alteration (Fig 15), are mixed within a reddened matrix featuring aggregates of burnt sediment [78].

**3.3.3. Mousterian Cone and Passage of the Column.** Local topography made for the deposit excavated in this part of the cave system to appear as a cone (layer 26) rising above a flat surface (defined at the time of excavation as a stratigraphic interface, the underlying sediment having correspondingly been designated as layer 27). The sediment from layer 26 is identical to that forming the matrix of the Access Corridor layers above. Layer 27, which fills the Passage of the Column, also contains sediment derived from above, as well as finds that refit as higher-up as layer 15 [65], but, overall, looks quite distinct from the Access Corridor's, both in the field and under the microscope. The thin section reveals a cemented mix of distinct inputs: fragments of carbonate crusts, often phosphatised, clay pellets, fragments of alluvial sediment, well-rounded sand grains, and some fragments of limestone from the wall (Fig 17A and 17B). The arrangement resembles granular aggregation, while continuous coatings around grains are common, as are "rolling pedofeatures" ([86]; Fig 17A and 17B); parallel lamination is weakly visible. The components form a poorly selected mass cemented by sparite (Fig 17C and 17D), which is suggestive of calcite accumulation occurring under phreatic conditions ([89]: 362–363). Fragments of reworked alluvial sediment displaying the same characteristics as the "layer 80" alluvium are also present (Fig 17B).

## 3.4. Spatial distributions

**3.4.1. The ensemble framework.** Through the accumulation of the deposit, several episodes of major collapse involving multi-ton, roof-fallen, or wall-detached boulders and slabs further modified Gruta da Oliveira's original, already complex volumetry. Such occurrences impacted both the nature and rhythms of sediment deposition and the ways that the cave could have been used by people and animals. Additional insight on site formation processes can thus be gained from the spatial patterning of the remains left behind by cave dwellers. To

**Table 3. Main micromorphological characteristics of the Access Corridor succession (I).** Microstructure and components.

| Unit | Aggregation | Porosity | Coarse components | Fine material |
|---|---|---|---|---|
| 11 | microgranular (masked by carbonate precipitation), with crumb and channel areas | few channels and chambers | few SIL; absent LST; few CRB; very few ABC (phosphate frs., some botrioidal; coprolites); very few clay aggregates | reddish brown, speckled |
| 12 | granular, with microgranular areas | few vughs; very few channels and chambers | few SIL and LST; common CRB (weathered calcite common in this class); common ABC (phosphate and organic frs.; bones with varied thermoalteration or weathering) | strongly variable |
| 13 | subangular blocky to crumb, weakly dev. | common chambers; few channels, planes, and vughs | few SIL; common LST; few CRB; common ABC (bones, often small or digested; few phosphate frs.); few opaques (often rounded) | reddish brown, speckled, with dark brown, dotted areas |
| 14 | channel, with microgranular and chamber areas | common channels; few complex pvs. | frequent SIL (quartz dominant, micas absent); common LST (often size > 1 mm); few CRB; common ABC (bones, with varied thermoalteration, often in clusters and digested; phosphate frs.); very few opaques; some ash aggregates | reddish to grayish brown, speckled to dotted |
| 15 | granular to subangular blocky, with microgranular areas | few channels, planes, and chambers | dominant SIL; few LST, CRB and ABC (bone frs. frequent in this class); some frs. of reworked alluvial sediment and phosphate frs. | reddish brown, speckled |
| 16 | channel with microgranular areas | few planes and channels; very few pvs. and vughs | dominant SIL; few LST, CRB (calcite mostly weathered) and ABC (few bone frs.); few frs. of phosphate, among them frs. of reworked phosphate crust, and clay pellets | reddish brown, speckled |
| 17 | granular | common channels, pvs. and vughs | dominant SIL; few CRB (weathered calcite crystals) and ABC (bone frs., only one fr. of shell); few frs. of phosphate, among them frs. of reworked phosphate crust, clay pellets, one "pedorelict"; LST absent | reddish brown (loc. grey), speckled |
| 18 | subangular blocky to crumb, with vughy and microgranular areas | common vughs; few channels, chambers and planes | dominant SIL; rare LST; CRB absent; few ABC (bones, often digested, phosphate frs.); very few opaques | reddish brown (loc. grey), speckled |
| 19 | channel with granular and m/granular parts (masked by secondary carbonate accumulation) | very few fine to very fine channels and vughs | dominant SIL (9/10 of comps), ranging very fine silt to coarse sand, with quartz dominant; among ABC only bones (few, well-preserved); LST and CRB virtually absent; OTHER: gypsum, opaques with varying size and shape (some are reworked Fe-Mn nodules); occasional pedorelics | reddish brown, dotted, not homogeneous; red in XPL, masked by PPL colour |
| 20 | apedal (loc. slight channel) | few planes, fine channels, and vughs | dominant SIL; very few LST (frs. of limestone often affected by secondary phosphatization) and ABC (bones frs., rare roots); CRB absent; frs. of reworked phosphate crust | reddish brown, speckled; XPL colour masked by PPL |
| 22/1 | apedal (loc. slight channel) | few channels and chambers, very few vesicles and vughs | common SIL (characteristics as average); common ABC (bones with distinct size, shape, and thermal alteration; 'charcoal' frs.; lithic artefacts); LST and CRB virtually absent; frs. of reworked phosphate crust | brown, loc. dark brown to reddish brown, many org. puncts in fine mat |
| 22/2 | apedal (loc. slight channel) | few channels and chambers, very few vesicles and vughs | common SIL (characteristics as average); few ABC (bones with distinct size, shape and thermal alteration); few LST; frs. of reworked phosphate crust | reddish brown, dotted |
| 22/3 | apedal, massive | very few channels and chambers | frequent ABC (mostly bones, some charred material); common SIL (but finer grain size, on average); common fine (silt-sized) opaque, with distinct shape; few LST with well-dev. phosphate rims; frs. of reworked phosphate crust | dark reddish brown, speckled |
| 22/4 | apedal, with local spongy areas | very few channels and chambers (loc. common) | common SIL (characteristics as average); common ABC (bones with distinct size, shape, and thermal alteration); few LST; frs. of reworked phosphate crust | reddish brown, dotted |
| 24 | apedal (secondary carbonate accumulation masking former granular aggregation) | few channels, planes, chambers, and vughs | frequent SIL; common LST; rare CRB (calcite crystals are strongly weathered); rare ABC (only charred material); few frs. of reworked phosphate crust | reddish brown, speckled (loc. grey for 2ndary carb accum) |
| 27 | granular (sedimentary), masked by secondary carbonate accumulation | common pvs. and vughs | dominant SIL (mostly sand-sized quartz grains); common CRB (weathered calcite) and ABC (bone frs. and few shell frs.); few LST; few frs. of of reworked phosphate crust; frs. of sediment (alluvial etc.) | reddish brown, dotted |

Key: SIL, LST, CRB, ABC—see text; fr(s).: fragment(s); dev.: developed; pv(s).: packing void(s).

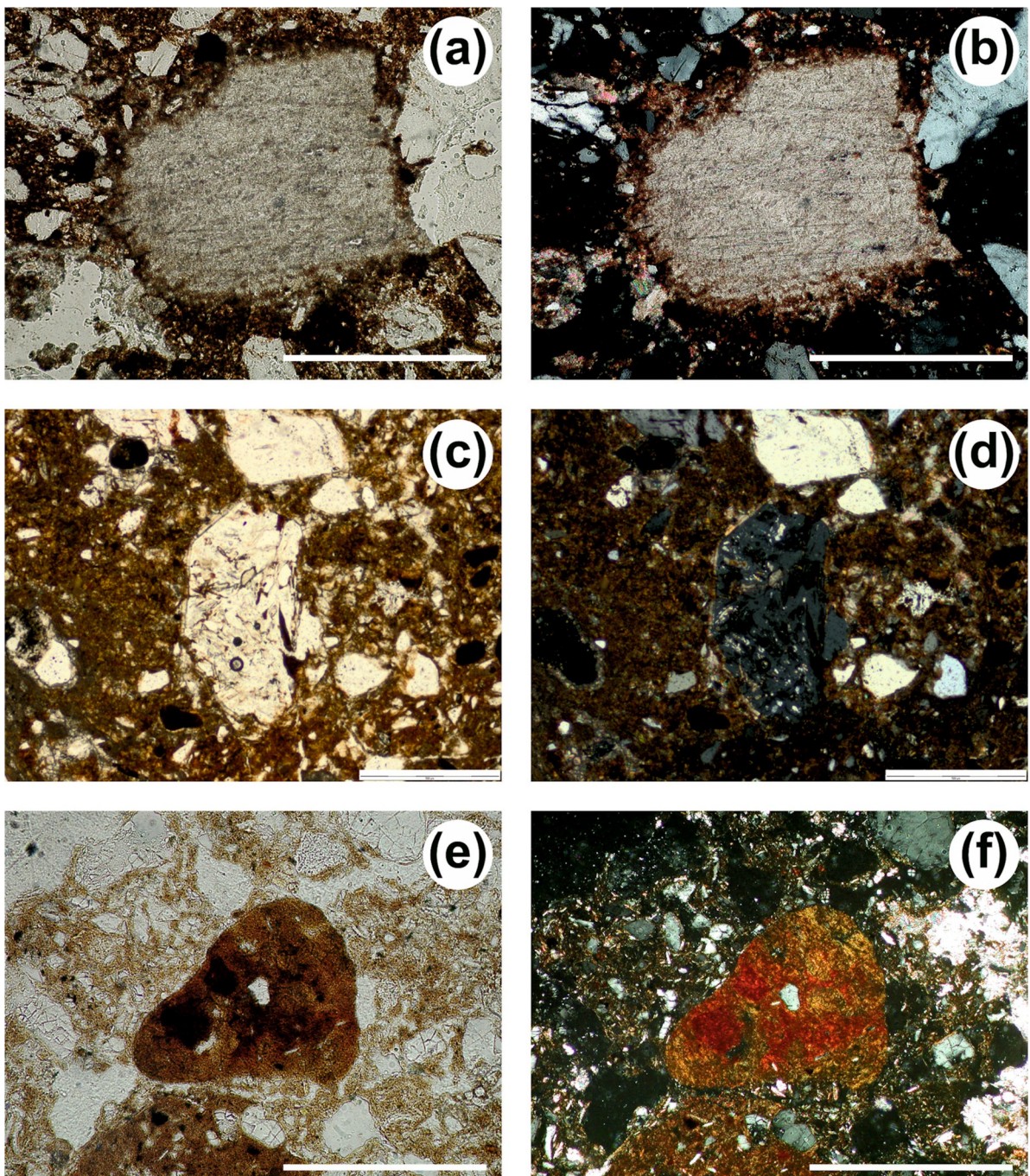

**Fig 13. Micrographs of coarse components.** (a) Calcite crystal (layer 18, PPL, scale 200 μm. (b) Same as (a) but XPL. (c) Gypsum crystal (unit 19, PPL, scale 500 μm). (d) Same as (c) but XPL. (e) Fragments of reworked soil material ("pedorelict"; layer 18, PPL, scale 200 μm. (f) Same as (e) but XPL.

do so, we use the ensemble framework outlined in reference [65], where the information provided by the stratigraphy and the geometry of the deposit is combined with the lithic taphonomy study of layers 15–27 (for which the intra- and inter-layer stone tool refitting work was exhaustive, but concerned the quartzite material only; [65]) and 7–14 (for which the refitting work was less systematic but concerned both chert and quartzite; [72]).

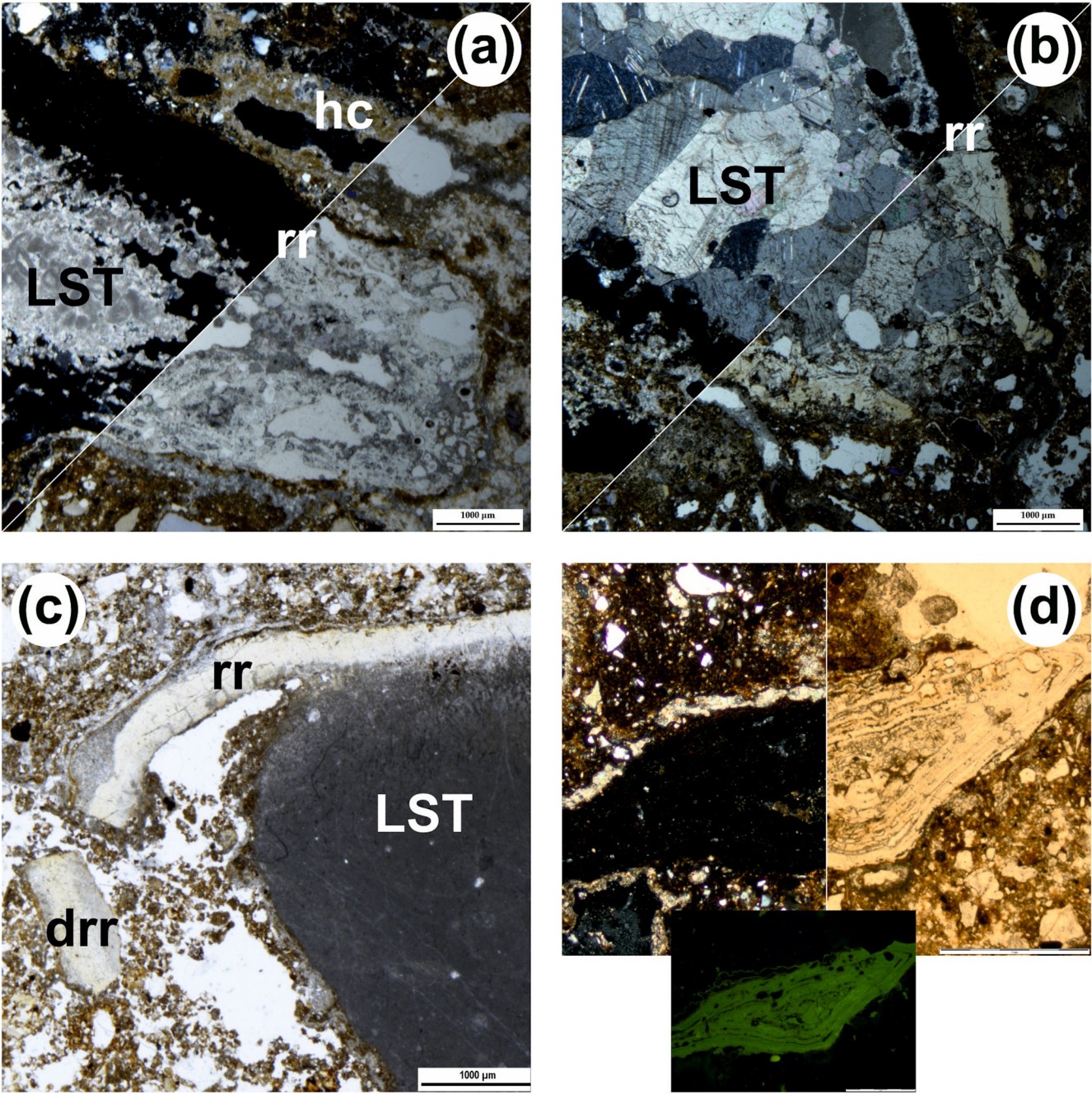

**Fig 14. Micrographs of pedofeatures showing the interaction between calcium carbonate and phosphate.** (a) Phosphate reaction rim (rr) on oncolithic limestone (LST), XPL (top left corner) and PPL (bottom right); notice the micrite hypocoating (hc) above the limestone fragment (unit 24, scale 1 mm). (b) Phosphate reaction rim (rr) on recrystallized limestone (LST), XPL (top left) and PPL (bottom right; unit 22/3, scale 1 mm). (c) Phosphate reaction rim (rr) on micritic limestone (LST), partly detached (drr) and broken, (PPL, unit 22/1, scale 1 mm. (d) Limestone fragments entirely transformed into phosphate, XPL (left), PPL (right) and BL (below; unit 22, scale 1 mm).

From bottom up, the following ensembles have been defined: **layers 23–25 (Access Corridor Lower)**, the infilling that levelled the floor of the area's crevassed bedrock; **layers 20–22 (Access Corridor Middle)**, the archaeologically rich deposit capped by the significant secondary carbonate cementation and flowstone development occurring after the deposition of layer

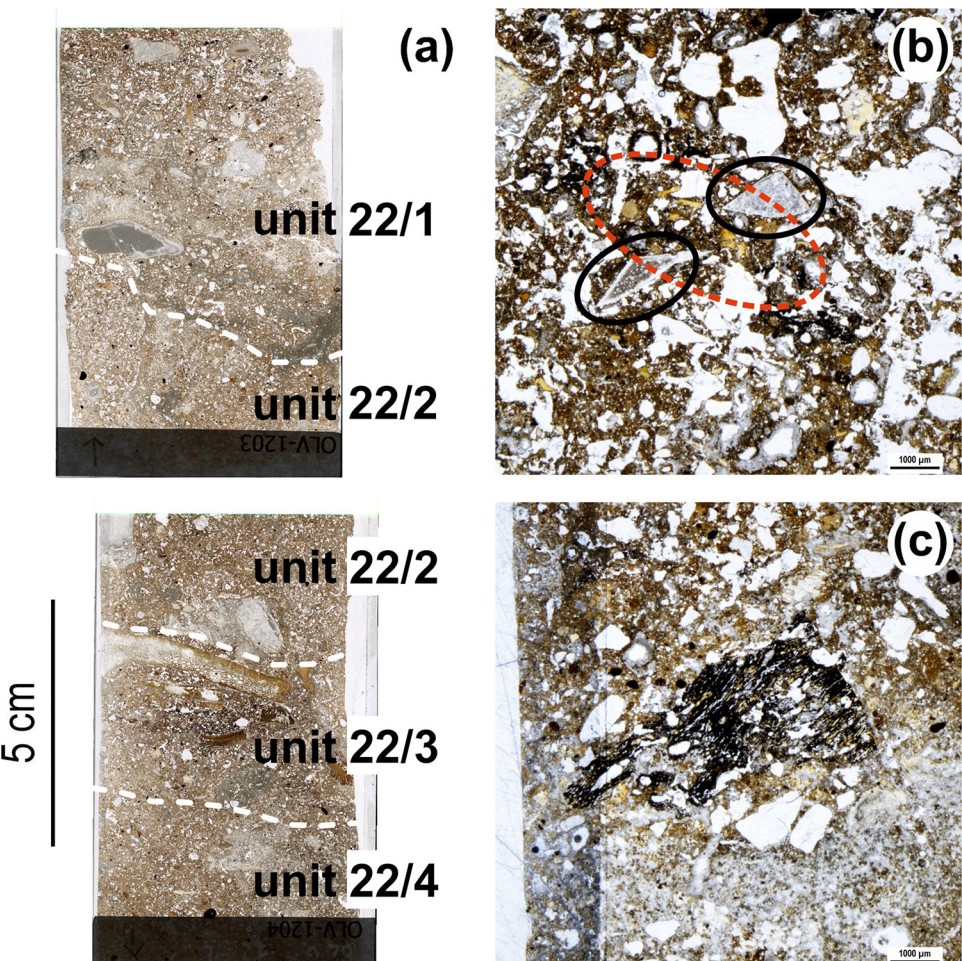

**Fig 15. Thin sections and micrographs from unit 22.** (a) Scan of thin sections OLV1203 (top) and OLV1204 (bottom) in PPL. The thin sections were digitally acquired using a high resolution flat-bed scanner equipped with a polarising nicol [83]) and are placed in the figure according to their original sampling position (see Fig 10) (b) Overall aspect of groundmass of unit 22/2: notice abundance of anthropogenic components such as lithic artefacts (black circles) and burnt bones (red circle; PPL, scale 1 mm). (c) Large charcoal fragment in unit 22/2 (PPL, scale 1 mm).

20; **layers 15–19 (Access Corridor Upper)**, the deposit—capped by the flowstone formed atop of layer 15—infilling the constrained spaces left between the huge boulders fallen before and after the accumulation of layer 19; **layers 13–14 (Basal Cave Interior)**, the archaeological rich infilling accumulated in the Access Corridor and Side Passage behind the major roof collapse fallen on grid units N-O/13-15, which, at ground level, interrupted the passage between the interior of the cave and the then-extant porch (Fig 2B); **layers 9–12 (Middle Cave Interior)**, the talus abutting the N-O/13-15 boulder that eventually infilled the Side Passage and blocked access to the 27-S Chamber; and **layers 7–8 (Upper Cave Interior)**, the deposit—sealed by flowstone and the colmatation éboulis that eventually concealed the entrance to the cave— formed atop of the levelled surface previously formed inwards of the N-R/13-15 boulder.

Within this scheme, the basal units of the 27-S Chamber and the Side Passage pose a particular site formation problem because, topographically, they lie adjacent to but largely disconnected from layers 16 and 13 of the Access Corridor. To account for the lateral discontinuity, those units were labelled separately—as layers 16bis and 13bis, respectively. The problem resides in that human and animal activity may have taken place in the 27-S Chamber and the

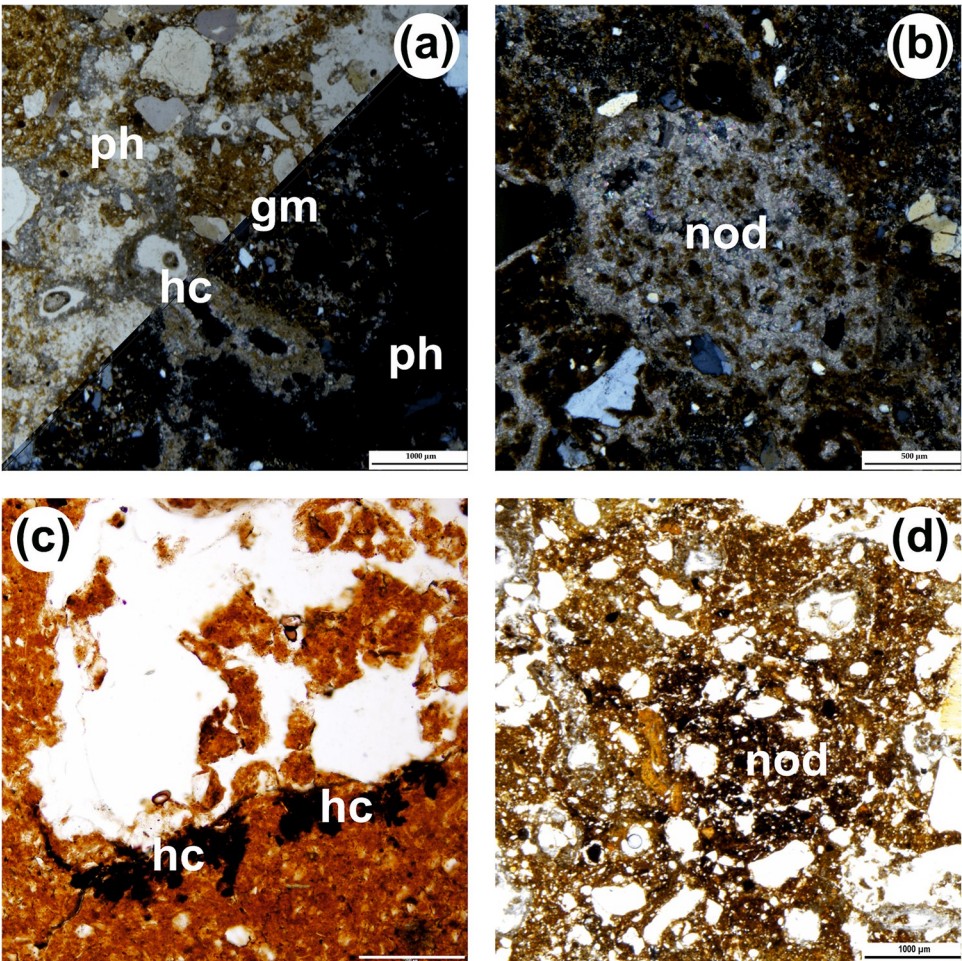

**Fig 16. Micrographs of pedofeatures related to secondary carbonate accumulation or Fe-Mn oxide accumulation.**
(a) Micrite hypocoating (hc) in channel; notice the granular reddish-brown groundmass (gm) and the amorphous phosphate infilling (ph), isotropic in XPL; PPL (top left) and XPL (bottom right; unit 24, scale 1 mm). (b) microsparite nodule (nod; unit 20, XPL, scale 0.5 mm). (c) Fe-Mn oxide hypocoating (hc; boundary between units 14 and 15, PPL with condenser, scale 0.2 mm). (d) Fe-Mn oxide nodule (nod; unit 20, PPL, scale 1 mm).

Side Passage through the long interval during which, at lower elevation, the Access Corridor was being infilled by layers 17–25. Conceivably, the finds retrieved in layers 16bis and 13bis could thus correspond to palimpsests comprising material therein accumulated both prior to and during the formation of the layers that eventually levelled the cave floor, firstly across the Access Corridor and the 27-S Chamber (layer 16), and then across the Access Corridor and the Side Passage (layer 13) too.

In the 27-S Chamber, the upper part of the stratification poses a similar problem. Even tough sediment accumulation and speleothem growth almost completely cluttered-up this area's connection to the outside, the remaining interstitial spaces continued to allow the passage of sediment, artefacts, and bones gravitating inwards from the Access Corridor. Eventually, this process led to the accumulation of a talus with apex in grid unit O18 and extending to rows 22–23. Topographically, that talus lay adjacent to layer 12, but its content was labelled separately as layer 12bis to account for the suspected complexity of its formation process. In addition, through the build-***up*** of the Middle and Upper Cave Interior ensembles (layers 7–12), and more so ever since, layers 13 and 12bis of the 27S-Chamber remained much

**Table 4. Main micromorphological characteristics of the Access Corridor succession (II).** Groundmass and pedofeatures.

| Unit | c/f RlDP | b-fabric | Pedofeatures and sedimentary features |
|---|---|---|---|
| 11 | open porphyric | crystallitic | common sparite / microsparite infs.; few loose disc. biogenic infs.; few dense silty clay cts. around grains |
| 12 | gefuric to enaulic | crystallitic | diffuse cementation by micrite / microsparite; few phosphate nodules; very few loose disc. biogenic infs.; very few irregular impure clay cts. in voids |
| 13 | open to close porphyric | undiff. | common sparite / microsparite infs. and cts. in voids; few loose disc. biogenic infs.; few Fe-Mn cts. and silty clay cts. around limestone frs.; few dense silty clay cts. around grains |
| 14 | open to close porphyric | undiff. with crystallitic parts | common micrite / microsparite hcts. and nodules, and few intercalations; few dense cont. biogenic infs.; few dense silty clay cts. around grains; very few phosphate nodules; very few Fe-Mn cts. around grains |
| 15 | close porphyric with gefuric parts | undiff. | few micrite hcts. (on grains), cts. and infs. (in pores); very few fabric infs. (aka passage features) in channel; very few dense silty clay cts. around grains |
| 16 | close porphyric with enaulic areas | undiff. with crystallitic areas | common micrite hcts. (in pores); very few micrite cts. (in pores); very few Fe-Mn nodules (irregular, orthic); common loose disc. infs.; few clay cts. coarse components; low-angle disc. lamination |
| 17 | close porphyric with enaulic areas | granostriated, poorly dev., with crystallitic areas | common micrite hcts. (in pores); few calcite nodules, irregular; few clay cts. on quartz grains; few NFC calcite cts. in channels; few Fe-Mn nodules, poorly dev. (typic and dendritic); few Fe-Mn cts. (esp. in lower part of TS) on bones and SIL grains; disc. low-angle lamination, poorly visible, especially in upper part of TS |
| 18 | open to close porphyric | granostriated, poorly dev., with crystallitic areas | frequent micrite, microsparite and sparite cts.; common dense incomplete infs., few with calcite crystals and very few with faecal pellets; NFC calcite cts. in channels; few phosphate nodules and impregnations; common microsparite cts. on coarser components (coarse sand fraction such as quartz and clay aggregates, some are rolling pedofeatures); common Fe-Mn typic nodules |
| 19 | single space to close porphyric | undiff. | common carb. (micrite) hcts., nodules and infs. (few intercalations too and few large mm-sized nodules); few carbonate (micrite) cts.; few biogenic loose infs. (incomplete and complete) |
| 20 | close to single space porphyric | undiff. | common impregnative Fe-Mn oxide features: nodules (often anorthic) and intercal; common secondary carb. cts. to infs. (often sparite) and hcts. (micrite), mainly in pores and rarely around SIL components |
| 22/1 | single space to close porphyric | undiff. | common micrite hcts., well-dev; few micrite cts. (some NFC) and calcite nodules; very few loose disc. biogenic infs. in channels and chambers (some made of pellets); few phosphate reaction rims |
| 22/2 | close porphyiric, loc. single space | undiff | common secondary carb. accumulation: micrite hcts., micrite and sparite cts., nodules and intercalation; few phosphate reaction rims and nodules; very few loose cont. biogenic infs. (with pellet, in TS 1204) |
| 22/3 | close porphyric | undiff | common secondary carb. accumulation (as in unit 22/1); common phosphate reaction rims on LST |
| 22/4 | close porphyric | undiff to crystallitic | common secondary carb. accumulation: micrite hcts., micrite and sparite cts., nodules and intercalation; common phosphate pedofeatures: reactions rims and nodules |
| 24 | close porphyric (mainly) | crystallitic to undiff. | common phosphate cts. / rims (especially on LST); common micrite hcts. and cts. (some strongly dev. to infs.); few phosphate nodules and very few quasi-coatings; very few Fe-Mn oxide impregnations and few Fe-Mn nodules (irregular) |
| 27 | enaulic | crystallitic | frequent clay cts. (esp. around SIL components); common calcite cts.; secondary carb. accumulation |

Key: c/f RlDP: coarse / fine related distribution pattern; carb.: calcium carbonate; cont.: continuous; ct(s).: coating(s); dev.: developed; disc.: discontinuous; fr(s).: fragment(s); hct(s).: hypocoating(s); inf(s).: infilling(s); loc.: locally; undiff.: undifferentiated.

exposed to bioturbation by roots, small mammal burrowing, and carnivore denning. Conceivably, the 12bis lithic and bone assemblages could therefore mix material derived syn-depositionally from layers 7–12 of the Exterior and the Access Corridor with upwardly, post-depositionally displaced material from underlying layers 13–15 of the 27-S Chamber itself.

**3.4.2. The integration of the open units.** To guide us on how best to incorporate the material assigned to layers 12bis, 13bis and 16bis in our ensemble-based spatial analysis, we used the available data on refit set-based, inter-layer linkage provided in Table 6 and Fig 18. For 12bis, connections with underlying units of the 27-S Chamber are significant, in accordance with expectations, but most are with layers found adjacent to or above, in both the

Table 5. Synopsis of main micromorphological features from the Access Corridor.

| Unit | THIN SECTION (OLV...) | COARSE COMPONENTS | | | | | | STRUCTURE | | | PEDOFEATURES | | | | | | | | | | | SEDIMENTARY FEATURES | |
| | | SIL | LST | CRB | ABC | phosphate fragments | fragments of soil material | pedality (development) | total porosity | biogenic voids | carbonate | | | cementation | phosphate | | other | | | | rolling features | lamination / bedding |
| | | | | | | | | | | | coatings + infillings | hypocoatings | nodules + intercalations | | coatings + rims | nodules | clay coatings | Fe-Mn coatings | Fe-Mn nodules | biogenic infilling | | |
|---|---|---|---|---|---|---|---|---|---|---|---|---|---|---|---|---|---|---|---|---|---|---|
| 11 | 0605 | 2 | 2 | 2 | 2 | 1 | 1 | p | w | w | 3 | | | p | | | 2 | | | 2 | | |
| 12 | 0604 | 2 | 2 | 3 | 3 | 1 | | p | p | p | | | | w | | 2 | 1 | | | 1 | | |
| 13 | 1, 0607 | 2 | 3 | 2 | 3 | 2 | | w | M | w | 3 | 3 | | p | | | 2 | 2 | | 2 | | |
| 14 | 0602, 0603 | 4 | 3 | 2 | 3 | 1 | | p | M | w | | 3 | 3 | p | | 2 | 2 | 1 | | 2 | | |
| 15 | 0608 | 5 | 2 | 2 | 2 | 1 | 1 | w | w | w | 2 | 2 | | | | | 2 | 1 | | 1 | | |
| 16 | 0802 | 5 | 2 | 2 | 2 | 2 | 1 | | w | w | 1 | 3 | | M | | | 2 | | 1 | 3 | | Y |
| 17 | 0803 | 5 | | 2 | 2 | 2 | 1 | p | w | w | 2 | 3 | 2 | p | | | 2 | 2 | 1 | 1 | | Y |
| 18 | 0606, 0804 | 5 | | | 2 | 1 | | w | w | w | 4 | 1 | 2 | | | 2 | 2 | 2 | | 3 | Y | Y |
| 19 | 0901 | 5 | | | 1 | | 2 | p | p | p | 3 | 3 | 2 | w | | | | | | 2 | | |
| 20 | 1205 | 5 | 1 | | 1 | | | | p | p | 3 | 3 | | S | | | | | 3 | | | |
| 22/1 | 1203 | 3 | | | 3 | 2 | | | p | p | 2 | 3 | 2 | S | 2 | | | | | 2 | | |
| 22/2 | 1203 | 3 | 2 | | 2 | 2 | | | p | p | 3 | 3 | 3 | S | 2 | 2 | | | | 2 | | |
| 22/3 | 1204 | 3 | 2 | | 4 | 2 | | | p | p | 3 | 3 | 3 | S | 3 | 2 | | | | | | |
| 22/4 | 1204 | 3 | 2 | | 3 | 2 | | | p | p | 3 | 3 | 3 | S | 2 | 2 | | | | 2 | | |
| 24 | 1202 | 4 | 3 | 1 | | 2 | | p | w | p | 3 | 3 | | S | 3 | 2 | | | 1 | | Y | |

The quantity and the development of micromorphological characteristics are coded with numbers or letters and colours. Semi-quantitative features (numbers): 5: dominant; 4: frequent; 3: common; 2: scarce or few; 1: occasional or very few; empty: absent or not detected. Descriptive features (letters): S: strong; M: moderate; w: weak; p: very weak; empty: absent. For abbreviations (SIL, LST, CRB, ABC), see text. Presence / absence: Y: present; empty: absent.

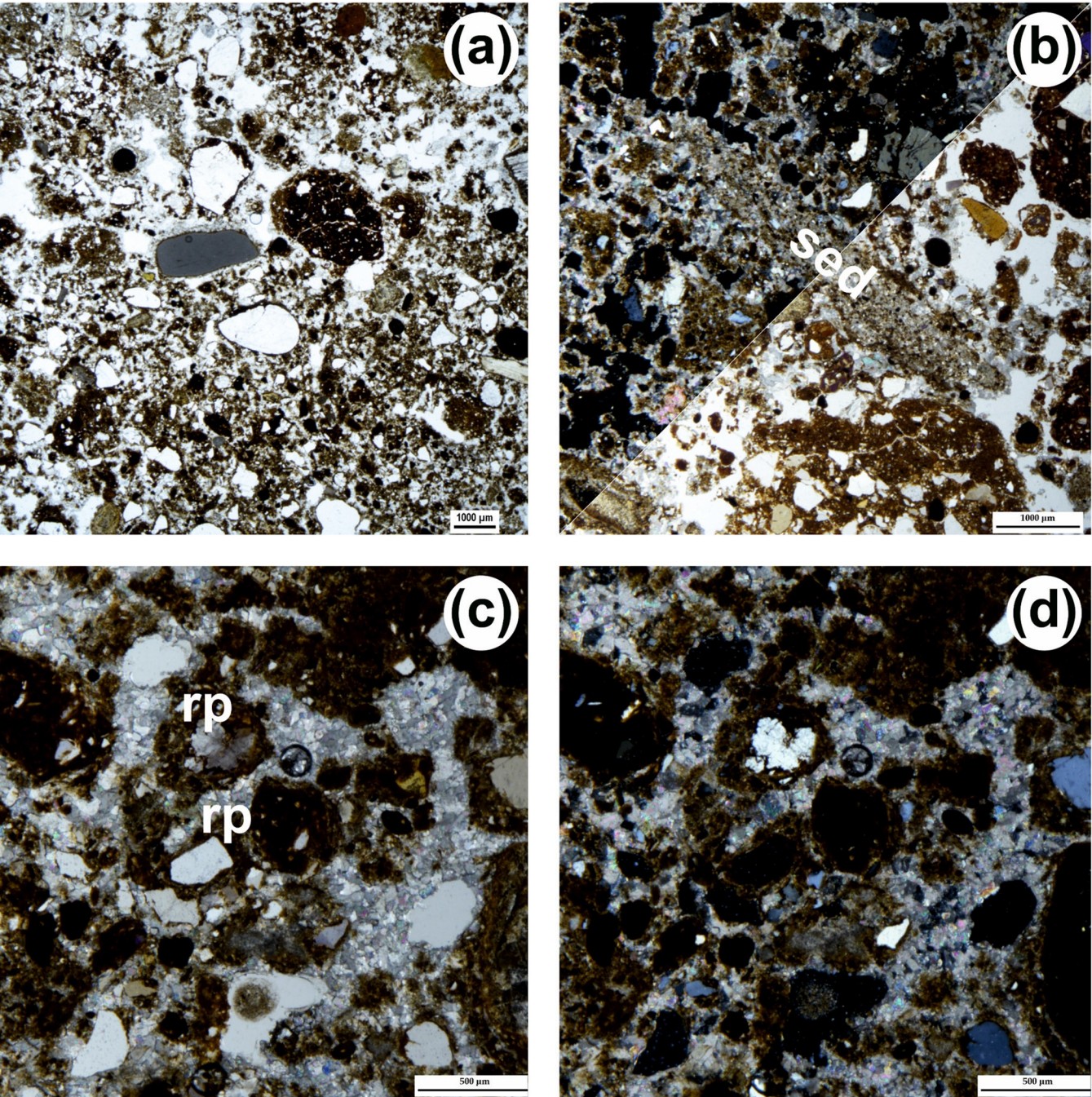

**Fig 17. Micrographs from unit 27 (Passage of the Column).** (a) Overall aspect of groundmass: notice moderate selection and variability of size, shape, and nature of components (PPL, scale 1 mm). (b) Fragment of reworked silty-sand sediment (sed); XPL (top left corner) and PPL (bottom right; scale 1 mm). (c) Sparite cement (PPL, scale 0.5 mm); notice thick continuous coating around grains (*aka* "rolling pedofeatures"–rp). (d) Same as (c) but XPL.

Access Corridor and the Side Passage (Fig 18A). Given these patterns, the 12bis provenience was grouped together with the Middle Cave Interior ensemble (layers 9–12).

With regards to 13bis, almost two-thirds of the connections are with overlying layers in the Side Passage itself or in the 27-S Chamber talus (Fig 18B). The significant number of items in refit sets including material from 13bis that are from lower down in the 27-S Chamber do not

**Table 6. Stone tool refitting data for the "open" stratigraphic units (ᵃ).**

|  | 12bis | 13bis | 16bis |
|---|---|---|---|
| **Number of refit sets that layer is represented in** | 24 | 5 | 6 |
| Sum of inter-layer connections counted by refit set |  |  |  |
| With layers above | 14 | 10 | 2 |
| With layers adjacent | 7 | 2 | 3 |
| With layers below | 17 | 4 | 6 |
| Total | 38 | 16 | 11 |
| **Number of 12bis, 13bis, and 16bis items present in refit sets** | 24 | 6 | 7 |
| Other items in refit sets that contain 12bis, 13bis, and 16bis items |  |  |  |
| From layers above | 24 | 11 | 8 |
| From layers adjacent | 11 | 1 | 5 |
| From layers below | 38 | 25 | 10 |
| Total | 73 | 37 | 23 |

(a) data from references [65, 72]

contradict this pattern. Indeed, almost all such items (20 out of 23) belong in the 25-strong refit set #1133 [65], a block that was reduced during the occupation of layer 15 but, post-depositionally, saw three items slide down along the cave wall in grid unit P18, and two be displaced upwards onto the Side Passage—one of which ended up in layer 13bis. Like 12bis, the 13bis provenience was therefore counted as belonging in the Middle Cave Interior ensemble (layers 9–12).

In the case of 16bis, the connections with units below or above and adjacent are about equally numerous (Fig 18C). The key information here is provided by refit sets #1077 and #1128 [65], which, together, provide horizontally short connections linking eight items from layer 15 with three from 16bis. This observation strongly suggests that post-depositional intrusion from layer 15 (or error in the décapage of the 15/16bis interface) is the main process underpinning the spatial scatter of the 16bis material. Based on these considerations, the 16bis provenience was grouped with layer 15 in the Access Corridor Upper ensemble (layers 15–19).

Note that, of the four other refit sets that include 16bis finds, three show longer connections with layer 16. The other set concerns a flake that, even though retrieved in 16bis, was extracted from a block reduced during the formation of layer 19 that also produced a downwards, along-the-wall scatter reaching as far as the Mousterian Cone (Fig 18C). These four refit sets show that the expected mechanisms of palimpsest formation—activity conducted in the 27-S Chamber during the accumulation of layers 17–25 of the Access Corridor, plus syn-depositional horizontal scattering taking place as the build-up of layer 16 gradually levelled the cave floor across both areas—also played a role, albeit a secondary one, in the constitution of the 16bis assemblage.

**3.4.3. The distribution of activity proxies.** Using the ensemble framework, Table 7 provides the stratigraphic distribution of three proxies for the activity of humans (stone tools and burnt bone fragments) and carnivores (coprolites) in the cave. For stone tools and burnt bone, Fig 19 illustrates their distributions' good correlation. Burnt bone is found throughout but features an anomalous concentration in row 19, at the elevation of the 15–19 ensemble. The anomaly stems from the accumulation against the back wall of the Access Corridor of hundreds of small fragments of tortoise carapace from layers 16–17 that, by themselves, represent 22.2% of the entire burnt bone assemblage; as shown in Fig 20 (which, for better resolution in terms of human occupation patterns, plots layers 15 and 20 separately), that concentration is

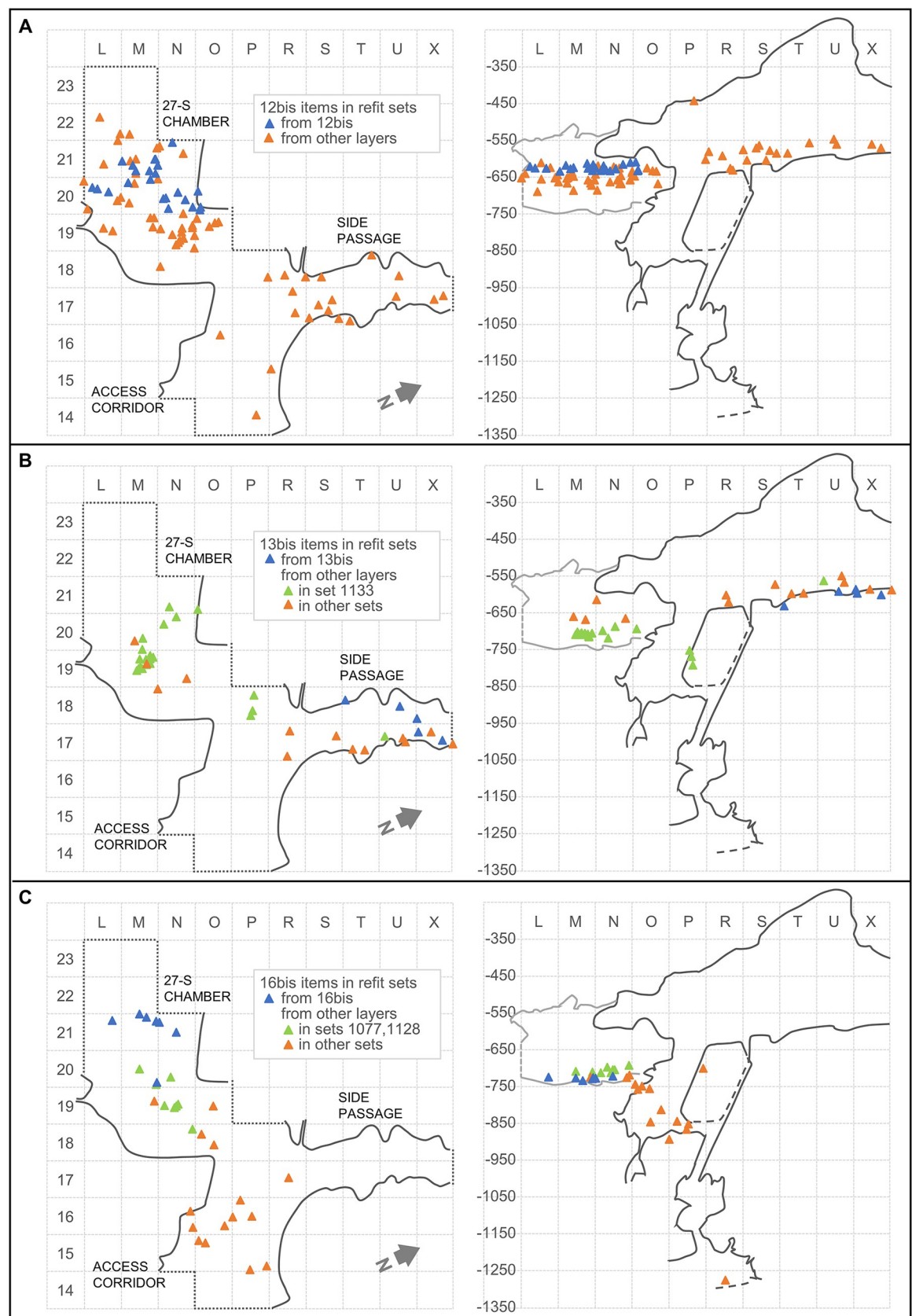

**Fig 18. The "open" units' refitting connections.** Horizontal (x-y plot; left) and vertical (x-z plot; right) distributions. **A.** Layer 12bis **B.** Layer 13bis. **C.** Layer 16bis. Elevations are in cm below datum.

entirely peripheral to the distribution of the stratigraphically associated stone tools. Bear in mind that the content of layers 16–19 reflects syn-depositional displacement from activity areas located outwards. In situ degradation of waste tossed from the entrance of the cave or that naturally rolled down the talus leading to the interior until stopped by an obstacle (the wall) parsimoniously explains the anomaly.

The concentration of burnt bones found in grid unit P15 at the elevation of layers 23–25 is another apparent anomaly. In this case, two factors are involved: downward percolation, along the cave wall, onto layer 23, of burnt bones originally associated with the layer 22 hearths, in connection with the latter's suffosion-induced deformation (as indeed noted at the time of excavation); and the spatial constraints posed by the outline of the cave walls at the elevation of layers 24–25, which largely restricted the accumulation of finds to line P of the excavation grid, where those layers' lithics and faunal remains (including the burnt ones) mostly come from. Otherwise, the spatial association of stone tools and burnt bones is tight, especially in those units for which in situ human activity is implied by stone tool refitting or the preservation of hearth features: layers 14, 15, and 20–22 (Figs 7, 9–11, 20).

**Table 7. Proxies for human and carnivore activity.**

|  | Area at base (m²) | Volume (m³) | Stone tools ([a]) | Burnt bone ([b]) | Coprolites ([c]) |
|---|---|---|---|---|---|
| Upper Cave Interior (layers 7–8) | 17.75 | 11.36 | 146 | 11 | 9 |
| Middle Cave Interior (layers 9–12) | 21.25 | 23.375 | 2942 | 664 | 88 |
| Basal Cave Interior (layers 13–14) |  |  | 7840 (d) | 1975 | 60 |
| (Layer 13) | 18.00 | 4.95 | (2868) | (600) | (55) |
| (Layer 14) | 18.00 | 4.95 | (4963) | (1375) | (5) |
| Access Corridor Upper (layers 15–19) |  |  | 4740 (d) | 3792 | 10 |
| (Layer 15) | 18.00 | 5.4 | (1882) | (1255) | (6) |
| (Layers 16–19) | 6.00 | 7.2 | (2257) | (2534) | (4) |
| Access Corridor Middle (layers 20–22) |  |  | 3435 | 896 | 2 |
| (Layer 20) | 5.20 | 2.34 | (1279) | (268) | (1) |
| (Layers 21–22) | 5.20 | 1.56 | (2156) | (628) | (1) |
| Access Corridor Lower (layers 23–25) | 4.00 | 4.4 | 556 | 292 | – |
| SUBTOTAL |  |  | 19,049 | 7627 | 169 |
| Layers 4–5 |  |  | 13 | – | – |
| Mixed ([d]) |  |  | 67 | 3 | – |
| No layer or attribution lost |  |  | 26 | – | – |
| Layers 26–27 |  |  | 1021 | 479 | 1 |
| SUBTOTAL VARIA |  |  | 1127 | 482 | 1 |
| GRAND TOTAL |  |  | 20,177 | 8109 | 170 |

(a) to account for the differential loss of small items in the brecciated parts of the deposit, chippage and debris were excluded, as also were manuports and hammerstones

(b) includes burnt teeth

(c) after [72], with additional data from subsequent fieldwork; when identifiable, of hyaena in all cases

(d) includes items labelled "layer 13 or 14" or "from the layers 15–18 interval"

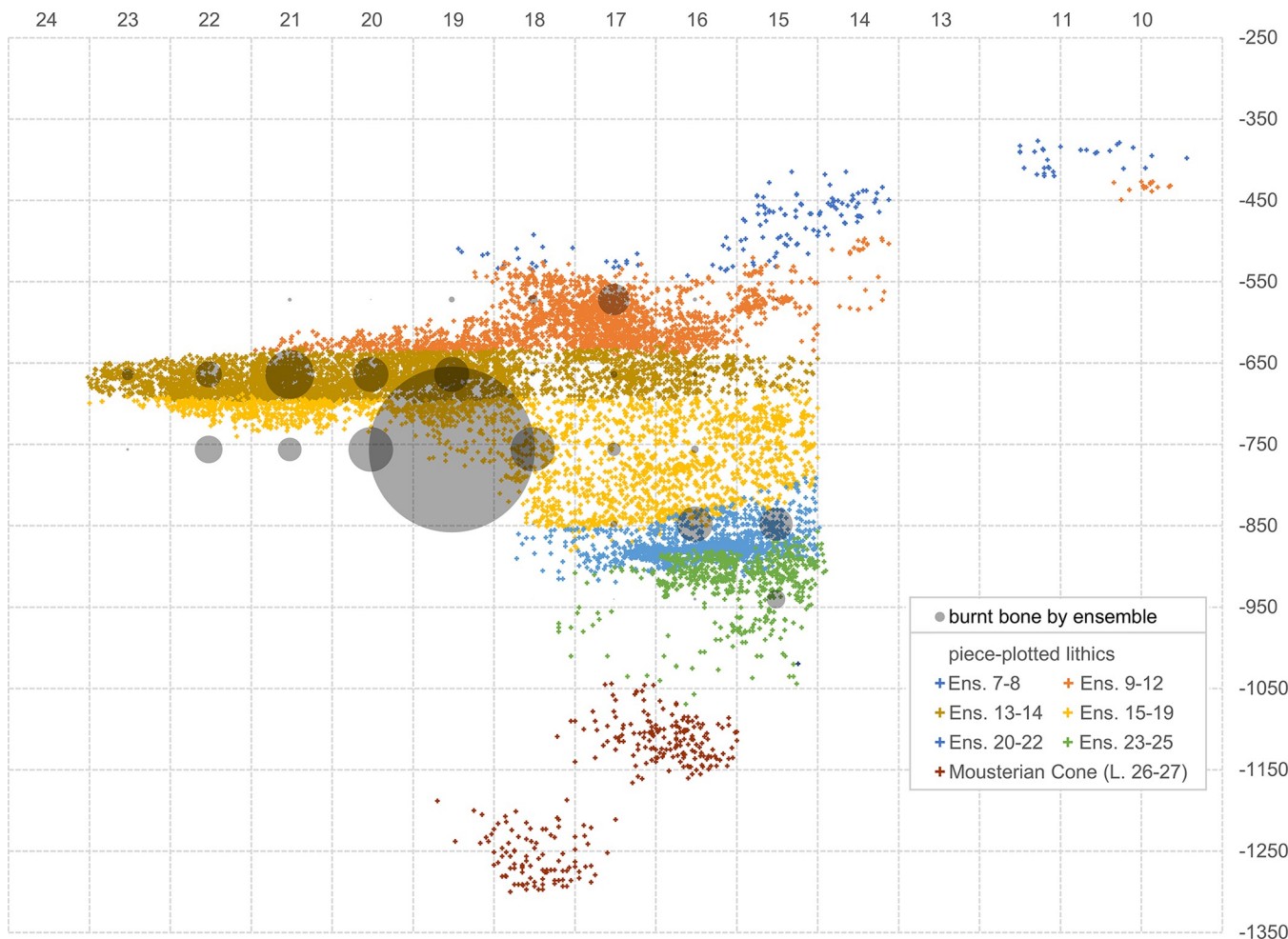

**Fig 19. Vertical distribution, per stratigraphic ensemble, of human occupation proxies.** Bubble-plot of burnt bone fragments, counted by ensemble, superimposed on the vertical projection along the y-axis of the piece-plotted stone tools.

Fig 20 also illustrates that, above layer 13, there is a marked decrease in the abundance and density of the proxies for human activity. The order of magnitude of the change is not easy to assess. In Fig 21, the number of items retrieved per stratigraphic unit is compared with two density indexes that account for the size of each unit's excavated area and volume. These indexes have shortcomings that require discussion. On one hand, using the total size of excavated areas in the denominator implies (a) when distributions are heterogeneous (as in e.g., layers 13–15, where finds were much denser in the 27-S Chamber than in the Access Corridor; Fig 20), that the index will underestimate the true density of the actual concentrations and, hence, the intensity of the human activity that such concentrations reflect, and (b) if rates of sedimentation vary across units, that the intensity of the human activity will be overestimated in those formed when accumulation was slower. On the other hand, considering excavated volumes can mitigate the bias caused by spatially heterogeneous distributions but works against units formed under faster rates of accumulation. An additional complication is that, for the same amount of washed-in matrix and the same number of finds discarded per unit of surface and unit of time, a more voluminous deposit is formed when large rock masses are present; when such is the case, measuring density with a volume denominator will necessarily

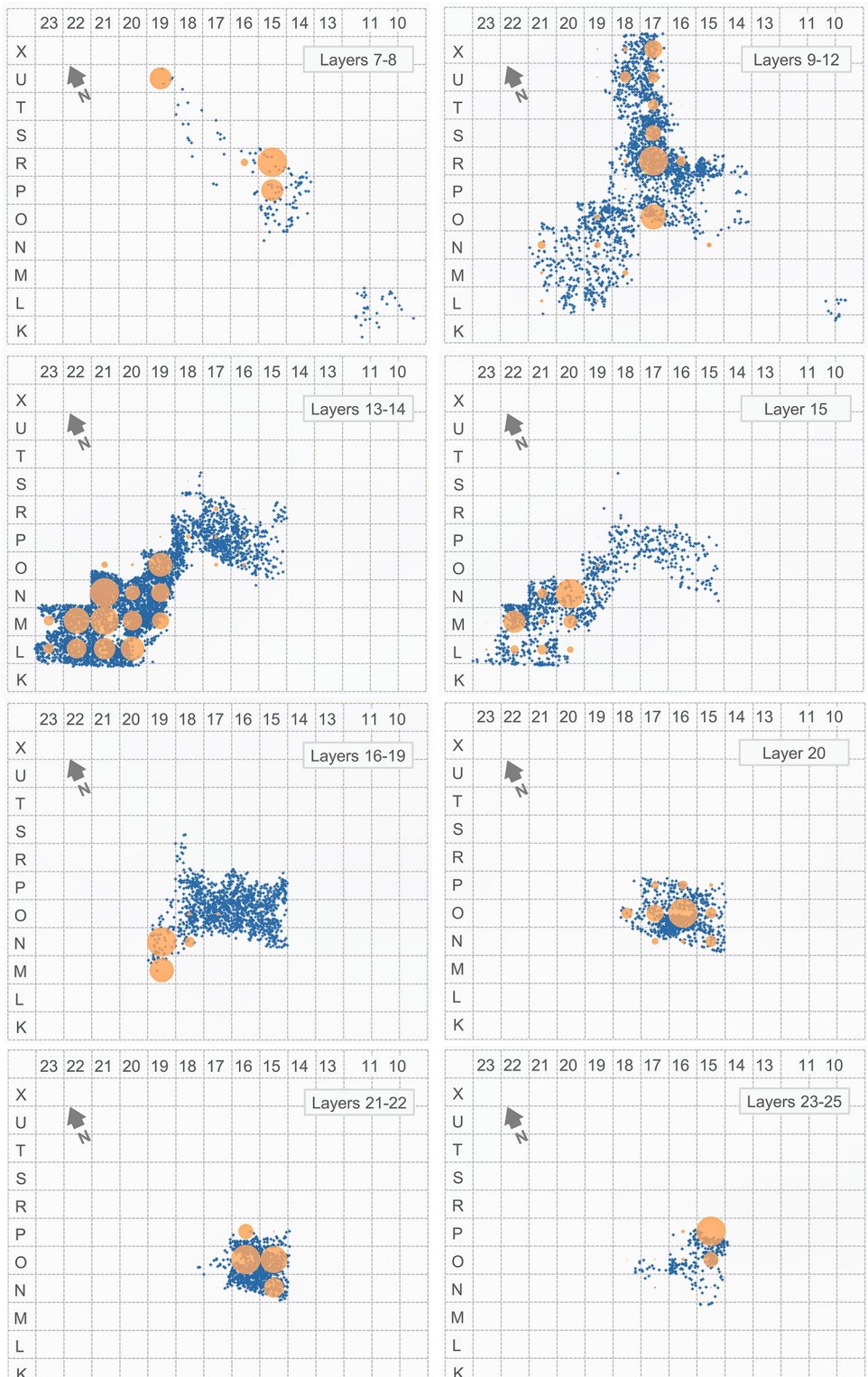

**Fig 20. Human occupation proxies.** Spatial distribution: bubble-plot of burnt bone fragments, counted per the stratigraphic units named in the plots' labels, superimposed on the x-y scatter of each of those units' piece-plotted stone tools.

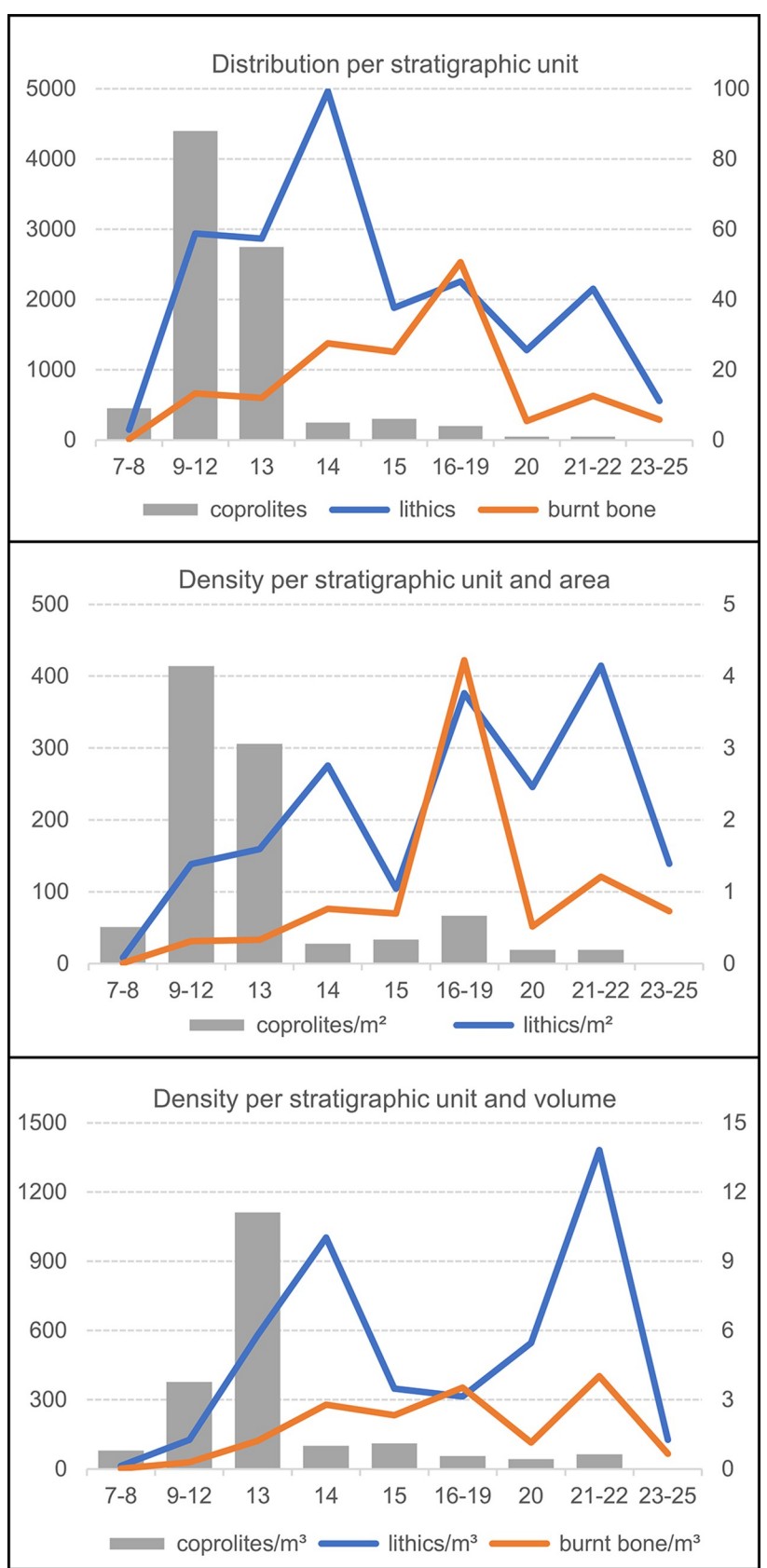

**Fig 21. Human and carnivore occupation proxies.** Change through time, per stratigraphic unit considered, in unstandardised absolute frequency and in absolute frequency standardised by area and by volume. The right axis is for the column plot (coprolites), the left axis is for the line plots (burnt bone and lithics, the latter excluding debris, manuports and hammerstones).

return values that will be lower by comparison. At Gruta da Oliveira, the latter must be borne in mind in the assessment of layers 16–19, as Fig 21 well illustrates for burnt bone: the spike in the category's stratigraphic distribution seen when assessing density based on area disappears when doing it based on volume.

The lack of adequate chronological controls hinders the estimation of rates of accumulation, but the source of the sediment that infilled the cave remained the same throughout, and the animal and plant proxies indicate that the oscillations in climate and environment recorded across the archaeological sequence were rhythmical and of similar magnitude [66, 72, 88]. Taking Fig 21 at face value, i.e., holding rates of accumulation constant, is therefore unlikely to bias the analysis of distributions, except, to some extent, with regards to layers 16–19, where, anyway, the concentration of burnt bone is itself anomalous in terms of occupation history, as seen above. Setting those layers to one side, Fig 21's plot of density by volume pictures a rather clear pattern of change through time: the proxies for human occupation peak in association with the presence of hearths, in layers 22, 21, and 14, and they decrease upwards of the latter, when it is hyaena coprolites, almost entirely absent further down in the sequence, that become quite abundant.

Note that 64% of the coprolites in layer 13 were retrieved along the walls in grid units O/15-17, M-N/19, and L/19-21 (Fig 2), and that the skeletal remains of the hyaenas themselves are exceedingly rare: two right premolars, a lower and an upper, probably from a single old individual, have been identified in layers 11 and 12 [72], and a phalange comes from layer 18. This evidence suggests that hyaenas used the interior area of the cave as a latrine only, and mostly at a time when, because of sedimentary infilling, that space had become too constrained for humans (who, when present, stuck to the porch). Based on the distribution of the remains, that time is after the end of the accumulation of layer 13; the latter's coprolites, therefore, may well reflect the subsurface impact of hyaena activity taking place during the time of formation of layers 9–12, coupled with the along-the-wall, downward migration of items ubiquitously observed across the succession.

## 4. Discussion

### 4.1. Sedimentary sources

This study further details the complexity of the deposit filling the Gruta da Oliveira, which largely results from the cave's configuration and context [64]. The sediments display significant lateral variability, derive from multiple inputs and distinct sedimentary and post-depositional dynamics, both natural and cultural. On average, the characteristics of the succession are those typical of southern European cave deposits formed in fluctuating climatic contexts: crude stratification, occurrence of a coarse angular fraction with variable amounts of fine material, poor textural sorting (see [89, 90]).

The different categories of components detected within the deposit come from distinct sedimentary sources. The SIL fraction washed into the cave through the entrance, the shafts connecting the interior spaces with the surface above, and the passages of the inner karst that these spaces lead to. Ultimately, this fraction derives from a range of sources: the Plio-Pleistocene surface sediments fed by the metamorphic formations of Central Portugal [91], in particular the soil and surface sediments of the Serra d'Aire massif; the fluvial sediment present in

the cave system prior to the beginning of the archaeological succession's accumulation; and, as indicated by the occurrence of reworked clay coatings and soil fragments, terrigenous sediments of Tertiary age and the soils developed from them. SIL components are found in all layers and, along with soil fragments, rolling features, and sedimentary features—namely, laminations—are particularly abundant between layers 14 and 20. This evidence suggests a phase of continuous inputs from the erosion of surface sediments/soils in a context of relative stability of the walls, as otherwise suggested by the scarcity of LST/CRB components.

The size, shape, and characteristics of the LST fragments indicate that they mostly derive from the degradation of the cave walls and that their stratigraphic distribution is largely independent of climate or environmental factors. Coarse and angular, these fragments exhibit no features indicative of rock degradation processes other than stress release or ordinary bedrock disintegration ([64, 90, 92]). However, frost slabs were observed in units 11, 12, and 15, suggesting that they formed under discontinuous frost action conditions. Boulders are common throughout. Roof collapse occurred, minimally, at four points in time: during the formation of layer 25; before and after the accumulation of layer 19; and at the interface between layers 15 and 14. The reason for these massive rockfall events remains unknown; the region is highly seismic, and is still tectonically active today [30], so those events could have been triggered by seismic activity; however, climatic factors and the rapid retreat of the *Arrife* escarpment may also have been involved.

The components of the CRB group (e.g., calcite crystals and speleothem fragments) are ubiquitous through the succession. They relate to phases of dissolution or erosion inside the cave; the calcite crystals, however, may also derive from the bedrock, which includes strongly recrystallized limestone and joints filled with chemically precipitated sparite veins. CRB components appear to be slightly more common in the upper part of the Access Corridor succession, which may simply be due to differential loss (or difficult detection) in the lower part, due to the massive cementation and the intense reaction therein observed between calcium carbonate and secondary phosphate.

All layers feature anthropogenic and biogenic (ABC) components: lithic artefacts, bones (often affected by thermal alteration), excrements, charcoal fragments, fragments of other kinds of burnt/charred material, and fine-grained fragments of amorphous burnt organic matter dispersed in the fine material. The soil-micromorphological evidence is therefore entirely consistent with the presence of an archaeological record all through the timespan covered by the succession and across the entire area of the cave that could be excavated. It corroborates the evidence—charcoal, burnt bone, intact hearth features—for fire to have been systematically and recurrently used through the twenty millennia of Gruta da Oliveira's human occupation history.

## 4.2. Morpho-sedimentary evolution

The base of the Access Corridor succession corresponds to a phase of cave roof collapse: the accumulation of the open-work, clast-supported limestone debris and boulders of layer 25. Subsequently, syn-depositionally with the accumulation of layers 23–24, the interstitial spaces were filled with sediment. Eventually, this process obstructed the communication with the Passage of the Column and provided the necessary support for the build-up of the overlying sedimentary column. The latter was laid down by sedimentary flow mechanisms with intermediate sediment concentration, such as runoff and overland flow, except for layer 22, which is clearly dominated by anthropogenic inputs. Tractive-like dynamics were also occasionally involved, as revealed by the poorly preserved bedding/lamination particularly apparent in layers 16–18. Originating outside, these inputs entered the cave through successive

events of mass movement, usually under wet conditions, as indicated by their field characteristics (massive arrangement, poor iso-orientation of natural components, occasional occurrence of weak lamination dipping inwards at a low inclination angle, stratification) and microfacies (relatively loose fabric, no preferential orientation, no patterning in the distribution of the coarse components, chaotic aspect, undifferentiated b-fabric, occurrence of dense silty clay coatings or external hypocoatings resembling rolling pedofeatures and presence of soil aggregates).

The collapse of the large boulder atop of layer 15 significantly modified the morphology of the cave. Sediment continued to wash in through interstitial spaces between the cave walls and the fallen rock mass, but the presence of the obstacle led to a change in the geometry of the accumulation: firstly, layers 13–14 were laid down in near horizontal manner across the 27-S Chamber and adjacent areas of the Access Corridor; here a debris flow-like accumulation then constituted overlying layers 9–12, which formed a talus that abutted the inner side of the boulder, cluttered the 27-S Chamber, and partially filled-up the Side Passage. At this time, the latter was secondarily sediment-fed by the joints communicating it with the higher levels of the cave system and the surface of the karst. This process formed a SE-dipping talus that merged with the NW-dipping, entrance-related talus in the area of grid units R-S/17-18, at the intersection between Access Corridor and Side Passage (Fig 2). Eventually, these taluses were covered by layers 7–8, which buried the N-R/13-15 boulder and reunited the cave's interior with the porch, forming a single sedimentary prism fed by slope-derived sediment and under the same accumulation dynamics that pertained lower down in the succession. At this time, the Access Corridor became almost completely infilled, explaining the limited number of remains found in layer 8 and the total lack of artefacts in layer 7, which immediately underlies the brecciated éboulis that, eventually, completely closed the cave.

Flowstones and calcareous crusts—some of which could be discontinuously observed across significant extensions of stratigraphic interfaces—were often used as markers and show that the accumulation proceeded by means of regular pulses. The calcite was precipitated by carbonate-saturated waters flowing over the cave floor and dripping, along the walls or from the roof, during the non-depositional hiatuses separating each pulse of gravity-driven accumulation. Thirteen such hiatuses—indicated by the episodes of calcite precipitation recorded atop of layers 24, 21, 20, 19, 18, 16, 15, 14, 13, 12, 11, 10, and 7—have been recognised (Table 2).

## 4.3. Palaeoenvironmental markers

In all likelihood, the recurrent alternation between pulses of accumulation and non-depositional hiatuses is climate-driven, with the hiatuses reflecting the stabilisation of the surrounding slopes during warmer oscillations. Assuming that such is indeed the case carries major implications for a proper understanding of the climatic and environmental proxies found in the deposit's content [66, 72]. Indeed, if so, each of the layers recognised during excavation must be considered as a palimpsest subsuming remains that (a) accumulated coevally with sediment deposition, and (b) were post-depositionally introduced subsurface, via trampling and bioturbation, during the following hiatus. Put another way, each layer is likely to subsume remains that entered the cave over at least one complete stadial/interstadial cycle.

Across the succession, in both its upper (layers 7–14, of MIS 5a age) and lower (layers 15–25, of MIS 5b age) halves, wood charcoal assemblages are exclusively of Scots pine (*Pinus sylvestris*), juniper (*Juniperus* sp.), and heath (*Erica* sp.) [88]. Since these charcoals stand for hearth-burnt fuel, they are unambiguous indicators of the environmental signal associated with human presence, and that signal is therefore for the cave's surroundings to consist of a cold pine-and-heathland landscape, i.e., for human presence to occur exclusively during

**Table 8. Ungulates versus carnivores in the faunal assemblages (NISP counts) (ᵃ).**

|  | Layers 7–14 (MIS 5a) | | Layers 15–25 (MIS 5b) (ᵇ) | |
|---|---|---|---|---|
| UNGULATES |  |  |  |  |
| Bovidae | 6 | 0.2% | 43 | 2.7% |
| Capridae | 814 | 30.4% | 454 | 29.0% |
| Cervidae | 1420 | 53.1% | 745 | 47.5% |
| Equidae | 225 | 8.4% | 173 | 11.0% |
| Rhinocerotidae | 205 | 7.7% | 153 | 9.8% |
| Wild boar | 4 | 0.1% | – | – |
| Indeterminate | 312 | – | 930 | – |
| CARNIVORES |  |  |  |  |
| Canidae | 17 | 11.5% | 39 | 33.6% |
| Hyaena (b) | 4 | 2.7% | 1 | 0.9% |
| Wild cat | 8 | 5.4% | 24 | 20.7% |
| Lynx | 27 | 18.2% | 6 | 5.2% |
| *Panthera* sp. | 5 | 3.4% | 18 | 15.5% |
| Brown bear | 47 | 31.8% | 9 | 7.8% |
| Fox | 34 | 23.0% | 6 | 5.2% |
| Felidae | 5 | 3.4% | 11 | 9.5% |
| Mustelidae | 1 | 0.7% | 2 | 1.7% |
| Indeterminate | 23 | – | 38 | – |
| TOTAL UNGULATES | 2986 | 94.6% | 2498 | 94.2% |
| TOTAL CARNIVORES | 171 | 5.4% | 154 | 5.8% |

(a) after [69, 72, 93]; additional data from J.-Ph. Brugal and M. Sanz (personal communication)

(b) coprolites excluded

stadials, or at least to be most frequent then. The high percentages of open-space taxa (horse, rhino)—16% of the ungulates determined to at least family in layers 7–14, 21% in layers 15–25 (Table 8)—are consistent with these considerations because (a) they must reflect the composition of the herbivore herds that the hunters targeted under the environmental conditions they lived in, and (b) the faunal remains are clearly anthropogenic for the most part [72, 93] (as otherwise implied by the overwhelming predominance of herbivores and the close spatial and stratigraphic association of burnt bone and stone tools) (Figs 20, 21).

With regards to the carnivores, their remains must reflect usage when humans were not present in the area. Assuming the validity of the inferences above, such would have certainly been the case during interstadials, as indeed implied by the pollen retrieved from the site's hyaena coprolites, which is suggestive of climatic conditions favouring the development of a thermophilous woodland landscape with *Corylus*, *Tilia*, and deciduous *Quercus* [72]. However, alternating stadial usage of the site by carnivores must also be contemplated, as the anthropogenic remains retrieved, rather limited if we bear in mind the huge timespan involved, imply that human occupation must have consisted of no more than short, spaced-out events. Indeed, the tibial shaft of a Neandertal retrieved in layer 19 is carnivore-damaged, and it was found at the same elevation—the interface between layers 19 and 20—as a number of cave lion remains, including three metatarsals in anatomical connection (Fig 22). This evidence suggests pene-contemporaneity, if not a direct, causal, predator-prey link. Note also that the carnivore guilds represented by the remains retrieved in the basal (layers 15–25) and upper (layers 7–14) halves of the succession are significantly different: the former are mostly of wolf and lion; the latter, of bear and lynx (Table 8). If we posit that the sediment's phosphate content is at least in part

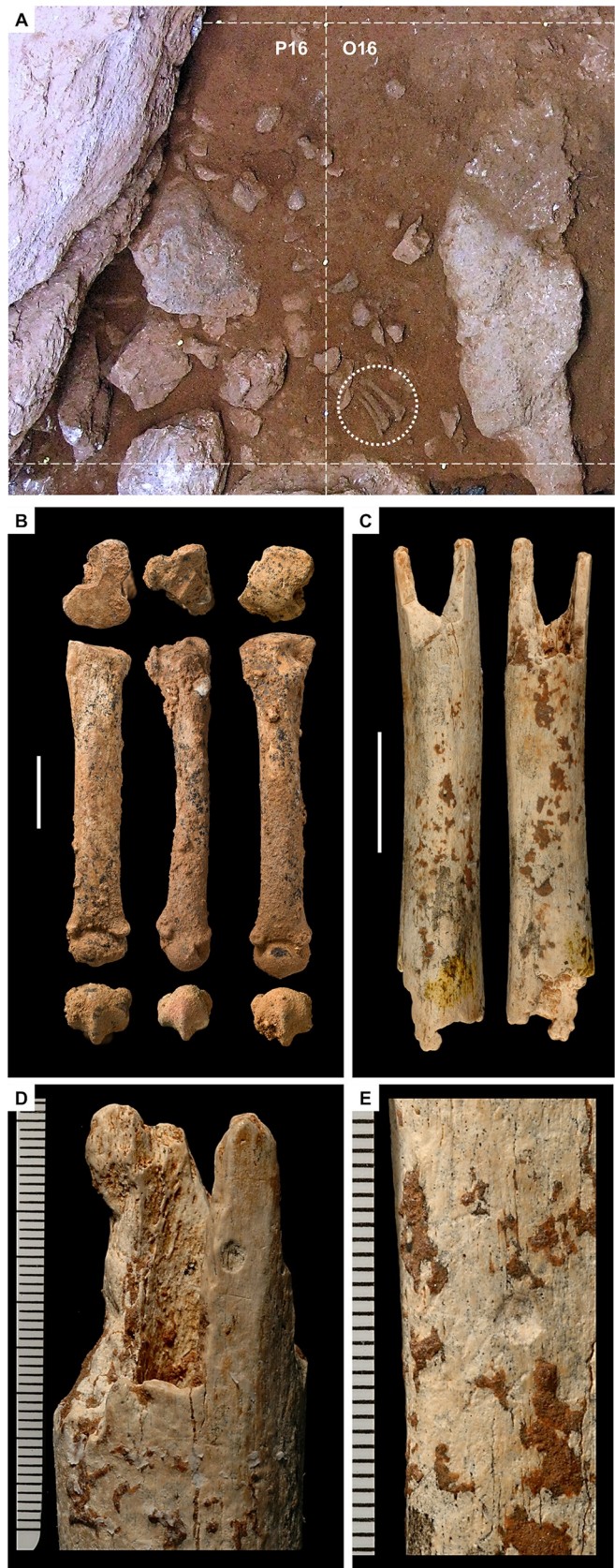

**Fig 22. Carnivores: presence and activity. A.** Detail view of the base of spit A64 (interface between layers 19 and 20); the dotted circle marks the position of three lion metatarsals in anatomical connection (a fourth from the same paw was found a few cm away in grid unit P16). **B.** Dorsal view of the three lion metapodials (scale bar = 3 cm). **C-E.** Tibial shaft of a Neandertal found nearby (grid unit O18) in basal layer 19, which had been chewed, gnawed, and pitted by carnivores (scale bar = 3 cm) (**C**), plus detail views of the carnivore modifications (**D-E**) (after reference [70]).

related to the in situ degradation of the by-products (animal tissue, excrements) of carnivore denning, then the increased phosphatisation observed as one moves down the succession is entirely consistent with the carnivore representation pattern, as the taxa that predominate in the older layers are larger and, being well-known accumulators of bone, may have made a more significant contribution to the faunal assemblages than is the case higher-up.

Bat guano is another potential source of phosphates, and bats are well represented in the micromammal assemblage [66]. In layers 7–14, their percentage of the total MNI varies between 10.1% (in layers 13–14) and 16.4% (in layers 9–12). The higher percentages are those reached in layers 15 (22.0%) and 16–19 (21.1%). If we posit that this variation signals a more significant presence of bats, and, consequently, that larger amounts of phosphate will be incorporated in the sediment when such is the case, then the Gruta da Oliveira data fit expectations.

Interestingly, layers 15 and 16–19 are also those for which available proxies suggest that a more pronounced climatic cooling occurred: the Mean Annual Temperature (MAT) derived from the composition of the microfaunal assemblages would have been, depending on the model used for the reconstruction, some 3 to 6°C below present [66]. This evidence is in good accord with the presence of frost slabs in layer 15 (see above), supporting that the interval during which layers 15–19 formed was the coldest of the succession, and much the same is implied by Bayesian modelling of the site's chronometric data, which assigns the deposition of those units to Greenland Stadial (GS) 22 (85.1–87.6 ka), at the very end of MIS 5b [63]. The composition of the herbivore assemblages is consistent with these inferences. Ibex occur in similar percentages in layers 7–14 and 15–25, reflecting unchanged conditions in the surrounding mountainous terrain, but the assemblage from layers 15–25, which includes the faunal remains from the units assigned to GS 22, features a statistically significant higher representation of the large, open-space taxa (rhino, aurochs, and horse): 23.5% of the identified specimens, contra 16.4% in layers 7–14 (Table 8). These macrofaunal data are consistent with the hunting grounds in the lowlands situated between the *Arrife* and the Tagus being extensively covered by moorland and prairie landscapes during GS 22 (or at least that they were then more so covered than before or after). It would appear, therefore, that the colder-than-average conditions extant during the formation of layers 15–19 would have coincided with (a) less frequent or less intense human visits, as intimated by the marked trough seen at this time in the stratigraphic variation of the stone tools' density-by-volume index (Fig 21), and (b) conversely, a more intense usage of the cave by bats and large carnivores.

## 4.4. Fire use and site occupancy

Putting to one side the burnt bone data for layers 16–19 (as the peak in this proxy's stratigraphic distribution that they represent is an artefact of extreme fragmentation), Fig 21 could also be read as showing that bone burning and, hence, the use of fire at the site, was less frequent during layers 15–19's cooler time of formation. One might therefore be tempted to take Fig 21 as supporting Dibble et al.'s contention that Neandertal fire use would have been largely restricted to warm climatic periods, i.e., that Neandertals would have used fire very infrequently, if at all, during cold periods [23]. However, that contention is based on an analysis that overlooked due consideration of a key point that must be borne in mind when making inferences about the human usage of caves and rock shelters, and this even if the different

units of occupation under comparison come from one and the same archaeological trench: If the morphology of the site changed significantly through time, the position of the trench relative to the inhabited space and, hence, relative to the emplacement of activities, may also have varied significantly across the different rungs of the trench's stratigraphic ladder [94, 95]. If such is the case, that variation can significantly bias the comparison between the samples of human activity that the different archaeological assemblages retrieved by the excavation stand for. In addition, as illustrated by the Portuguese cave site of Gruta da Figueira Brava [96], these kinds of sites are susceptible to erosional losses related to e.g., the receding of overhangs and scarp faces—meaning that, as one moves up the succession, the preserved parts of the deposit that one excavates may well be located in areas that, at the time of occupation, were rather marginal. These caveats are all the more important whenever the palimpsests of multiple occupations that most stratigraphic units of analysis correspond to nonetheless retain to a significant extent the original spatial structuration of the activities.

At Gruta da Oliveira, the case is that (a) the relation of our trench with the changing morphology of the cave does become substantially different as one moves up the stratification and, (b) as illustrated for layers 13–15 by the distributions in Fig 20, the site's formation process did not erase the original structure of the inhabited space. For layers 15–19, for instance, our trench sampled an interior area that the presence of large, incompletely buried boulders may have rendered less amenable to human installation; because of this constraint, humans may at that time have restricted themselves to areas of the cave located outwards. Ditto for layers 9–12, where the impression of sparse, limited presence is biased by the fact that, where excavated, those layers correspond to the back of a sedimentary talus that had already largely filled the cave up. Since there is no reason to think that activity areas were less spatially focused during the formation of layers 9–12, it is entirely conceivable that concentrations of remains akin to those seen further down in the succession may well exist in the unexcavated cave porch and adjacent areas; outwards of the Access Corridor trench, however, no excavation work could be carried out, and it is unlikely that such work will be possible in the future, because of the constraints posed by (a) the instability of the roof near the scarp face, (b) the engineering setup that, in the initial phase of the work, was put in place to secure the site, and (c) the proximity of the extant entrance to the edge of a *c.* 40 m cliff extending down to the spring below.

The assessment of change through time in site occupancy patterns is further complicated by issues relating to the interpretation of the taxonomic composition and degree of fragmentation of the burnt bone assemblage: unidentified fragments (N = 4566) are 60% of the total (N = 7627), and tortoise (N = 2706; 99%, N = 2679, are of shell) makes up 88% of the remainder (N = 3061). As (a) roasting-on-coals of whole tortoises [69] is the culinary preparation that humans used at the site, and (b) tortoise shell is readily identifiable and, when burnt, breaks easily and much, it is to be expected that significant variation in the availability of tortoises, or in the extent to which they were harvested, will carry a no less significant impact on the amount of burnt bone fragments per unit of analysis. Such is indeed the case: in layers 7–12 and 16–25, the variation in the percentage of the burnt bone assemblage that each stratigraphic unit represents closely tracks the corresponding percentages of the site's tortoise total and of tortoise among burnt bone; while the proportion of the total burnt bone assemblage that layers 13–15 represent is three to six times higher than their proportion of the site's tortoise material (Fig 23A). The marked decrease in the values of the burnt bone proxy seen when layers 16–19 are compared to layers 9–12, 13–14 and 15 must therefore be a mere consequence of tortoise becoming much less abundant after layer 16, not of fire being less intensively used at the site.

The data on stone tool density are therefore the less biased, more direct, and best measure of the variation in intensity of site usage. By excavated volume, they peak (a) in precisely those stratigraphic units (layers 14 and 21–22) that yielded well-preserved hearth features and (b) at

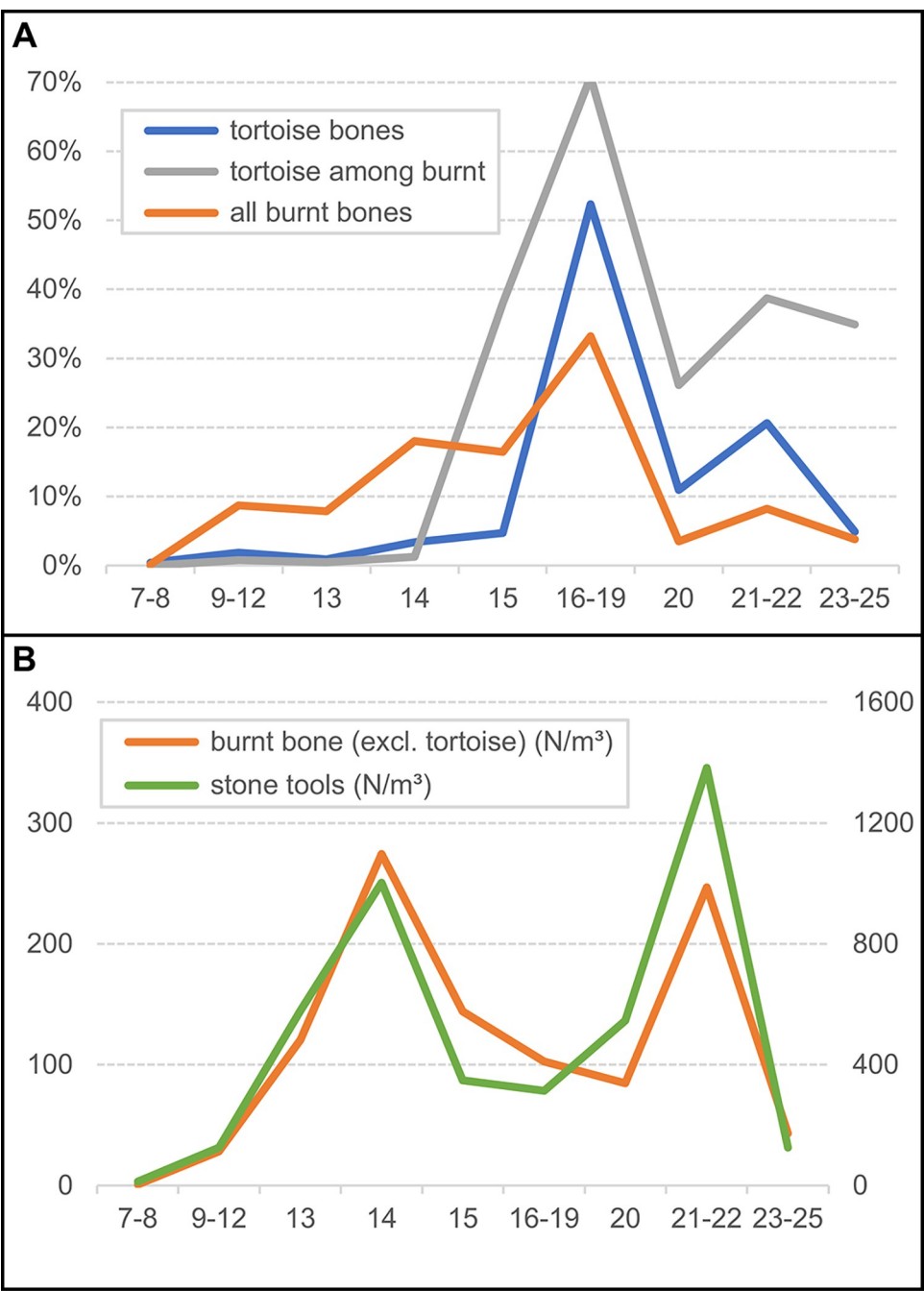

**Fig 23. Stratigraphic variation in the abundance of tortoise. A.** Each stratigraphic unit's percentage of the site's total tortoise remains, and each stratigraphic unit's percentage of tortoise among its burnt bones compared with the vertical distribution of the site's entire burnt bone assemblage. **B.** Change through time in proxy indexes for human occupation (absolute frequency standardised by volume) when tortoise is excluded from the burnt bone counts (the right axis is for the lithics, the left axis is for the burnt bone).

values that are both similar and significantly higher than any seen elsewhere in the succession. Indeed, if tortoise is excluded from the burnt bone counts, the curves for the two proxies become identical (Fig 23B). This is hardly coincidental, as it was during the formation of layers 21–22 and 14 that the configuration of the site was optimal for settlement: those were times of

relative wall stability; the pre-existent, irregular, rocky surfaces (the basal éboulis of the Access Corridor, and the karren-like bedrock of the 27-S Chamber) had been levelled by the accumulation of fine sediments (layers 23 and 15), providing ideal conditions for human installation; and sufficient head room remained for the comfortable circulation of personnel.

It would therefore be inappropriate to assume that the patterns of stratigraphic change seen at Gruta da Oliveira reliably mirror change through time, across the twenty millennia therein represented, in, e.g., the settlement system, or the density of the region's Neandertal population. However, there is one general issue of human behaviour in Europe during late MIS 5 times that the evidence reviewed here does contribute significantly to: the control and use of fire technology by the Neandertals. In Portugal, substantial combustion remains (extensive accumulations of ash and burnt sediment, charcoal, and burnt bone) are a consistent feature of the other two sites that (a) are well dated to the interval, and (b) were both recurrently occupied over many millennia and capacious enough for residential, if short-term and intermittent settlement—Gruta Nova da Columbeira [97, 98], and Gruta da Figueira Brava [96, 99]. Together with the exceptionally well-preserved hearth-focused contexts of MIS 5a and MIS 3 age known in Spain's Mula basin (Cueva Antón and La Boja; [25, 100–102]), these three Portuguese sites add strong support to the notion that Neandertals fully mastered fire-making—as indeed reference [24] argued for the French case, based on micro-wear analysis of chert tools that were occasionally used as "strike-a-light" items. Minimally, the weight of the evidence clearly leans towards placing the burden of proof on the side of those who, like reference [23], take the opposite as their null hypothesis.

## 5. Conclusion

The Gruta da Oliveira infilling forms an archaeological succession with significant lateral variability. The sediment was mostly laid down by gravity-driven processes, through different sedimentary mechanisms. The arrangement of components, the observed groundmass, and the characteristics of the deposit reveal that the mechanisms involved in the accumulation of layers 20–25 of the Access Corridor succession are mostly related to gravitational processes, acting both directly and in synergy with surface waters. This evidence supports the similar conclusions previously reached for the overlying units of the stratification (layers 15–19 of the Access Corridor, and layers 7–14 of the Side Passage and the 27-S Chamber) [64].

Climatic and environmental controls and biogenic, including anthropogenic, factors explain the slight fluctuations in sedimentary inputs and processes and in the diagenetic dynamics observed throughout. Post-depositional dynamics, in part due to human activity, were always in operation; however, their impact on the site's interpretative potential is limited, as demonstrated by stone tool refitting [65, 72].

The combined consideration of the geological, palaeobotanical, zoological, and archaeological evidence paints a coherent picture of alternating usage of the cave by humans and carnivores through the times of global cooling leading from Last Interglacial to Early Glacial conditions. Humans were the main contributors to the deposit's non-sedimentary components; in the case of the layer 22 hearth horizon, the deposit is itself largely anthropogenic. Variation in the emplacement and intensity of human occupation is parsimoniously explained as a by-product of how the morphology of the cave itself changed through time; it bears no necessary link to causes external (e.g., climate/environmental change) or internal (e.g., change in technology, social organisation, or ecological adaptation) to the cultural system.

As demonstrated by the ubiquitous presence of wood charcoal and burnt bone, fire was used throughout, leaving traces whose variation (from microscopic lumps of burnt sediment to in situ hearth features) relates to issues of situation (position of the excavation trench

relative to activity areas that were spatially heterogeneous) and preservation (differential impact of post-depositional dynamics), rather than to issues of availability or production. When compared with the regional Upper Palaeolithic record from 50,000 years later, e.g., the evidence from Gruta do Caldeirão [34, 103], the patterns of cave usage by the Middle Palaeolithic Neandertals who took shelter at Gruta da Oliveira are essentially the same.

## Acknowledgments

Over the years, the archaeological field work carried out in the Almonda karst benefited from logistical support provided by a number of different institutions, primarily the following: Câmara Municipal de Torres Novas, RENOVA–Fábrica de Papel do Almonda, and Sociedade Torrejana de Espeleologia e Arqueologia (STEA). Some of the basic data and pictures from the micromorphological study partly derive from Ms Barbara Maistrelli's 2015 University of Trento M.A. thesis, elaborated under the direction of D.E.A. with the collaboration of Drs. Daniela Anesin and Maurizio Zambaldi; we are indebted to them for allowing us to reinterpret and publish the data. Thin section scans were obtained at the Department of Humanities of the University of Trento in collaboration with Mr. Paolo Chistè and Dr. Maurizio Zambaldi. The micrographs reproduced in this paper were taken from the microscopes of the Laboratorio Bagolini of the University of Trento and the equipment of the AMBI-LAB of the University of La Laguna, Spain, during a short stay by D.E.A. (March 2020), who is indebted to Dr Carolina Mallol and the AMBI-LAB research staff for hosting him.

## Author Contributions

**Conceptualization:** Diego E. Angelucci, João Zilhão.

**Data curation:** João Zilhão.

**Formal analysis:** João Zilhão.

**Funding acquisition:** Diego E. Angelucci, João Zilhão.

**Investigation:** Diego E. Angelucci, Mariana Nabais, João Zilhão.

**Methodology:** Diego E. Angelucci, Mariana Nabais, João Zilhão.

**Project administration:** João Zilhão.

**Resources:** Diego E. Angelucci, Mariana Nabais, João Zilhão.

**Supervision:** João Zilhão.

**Visualization:** Diego E. Angelucci, Mariana Nabais, João Zilhão.

**Writing – original draft:** Diego E. Angelucci, João Zilhão.

**Writing – review & editing:** Diego E. Angelucci, Mariana Nabais, João Zilhão.

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
