## [Decision Letter · Decision Letter 0]

24 May 2023

PONE-D-23-05775Formation processes, fire use, and patterns of human occupation across the Middle Palaeolithic (MIS 5a-5b) of Gruta da Oliveira (Almonda karst system, Torres Novas, Portugal)PLOS ONE

Dear Dr. Zilhão,

Thank you for submitting your manuscript to PLOS ONE. After careful consideration, we feel that it has merit but does not fully meet PLOS ONE’s publication criteria as it currently stands. Therefore, we invite you to submit a revised version of the manuscript that addresses the points raised during the review process.

We look forward to receiving your revised manuscript.

Kind regards,

Enza Elena Spinapolice, Ph.D

Academic Editor

PLOS ONE

Journal Requirements:

2. In your manuscript, please provide additional information regarding the specimens used in your study. Ensure that you have reported specimen numbers and complete repository information, including museum name and geographic location. 

For more information on PLOS ONE's requirements for paleontology and archeology research, see https://journals.plos.org/plosone/s/submission-guidelines#loc-paleontology-and-archaeology-research.

"Over the years, the archaeological field work carried out in the Almonda karst benefited from financial and logistical support provided by a number of different institutions, primarily the following: Câmara Municipal de Torres Novas, Instituto Português de Arqueologia (IPA), RENOVA – Fábrica de Papel do Almonda, Fundação para a Ciência e Tecnologia (projects PTDC/HIS-ARQ/098164/2008 and PTDC/HAR-ARQ/30413/2017), and Sociedade Torrejana de Espeleologia e Arqueologia (STEA). Thin sections from Gruta da Oliveira were paid for by the IPA and the University of Trento (D.E.A.'s personal allowance)... Support for the writing and publication of this paper was provided by FCT grants UIDB/00698/2020 and UIDP/00698/2020."

"Funding for this work was provided by FCT (Fundação para a Ciência e Tecnologia; Portugal) grants UIDB/00698/2020 and UIDP/00698/2020. The funders had no role in study design, data collection and analysis, decision to publish, or preparation of the manuscript."

4. We noted in your submission details that a portion of your manuscript may have been presented or published elsewhere:

"Figure 2 is modified after https://doi.org/10.1016/j.quascirev.2021.106885. Parts of multi-panel Figures 4, 5, 7-12 and 22 were published in previous reports, namely https://doi.org/10.1002/gea.20267, http://hdl.handle.net/10451/31145, and https://doi.org/10.1371/journal.pone.0192423.

The images in those multi-panel figures that are modified or reused here are necessary to illustrate the stratigraphic descriptions and analyses presented in the text."

Please clarify whether this publication was peer-reviewed and formally published. If this work was previously peer-reviewed and published, in the cover letter please provide the reason that this work does not constitute dual publication and should be included in the current manuscript.

5. We note that Figure 1 in your submission contain map/satellite images which may be copyrighted. All PLOS content is published under the Creative Commons Attribution License (CC BY 4.0), which means that the manuscript, images, and Supporting Information files will be freely available online, and any third party is permitted to access, download, copy, distribute, and use these materials in any way, even commercially, with proper attribution. For these reasons, we cannot publish previously copyrighted maps or satellite images created using proprietary data, such as Google software (Google Maps, Street View, and Earth). For more information, see our copyright guidelines: http://journals.plos.org/plosone/s/licenses-and-copyright.

(1) You may seek permission from the original copyright holder of Figure 1 to publish the content specifically under the CC BY 4.0 license.  

6. We note that Figure 4 includes an image of a participant in the study. 

As per the PLOS ONE policy (http://journals.plos.org/plosone/s/submission-guidelines#loc-human-subjects-research) on papers that include identifying, or potentially identifying, information, the individual(s) or parent(s)/guardian(s) must be informed of the terms of the PLOS open-access (CC-BY) license and provide specific permission for publication of these details under the terms of this license. Please download the Consent Form for Publication in a PLOS Journal (http://journals.plos.org/plosone/s/file?id=8ce6/plos-consent-form-english.pdf). The signed consent form should not be submitted with the manuscript, but should be securely filed in the individual's case notes. 

Please amend the methods section and ethics statement of the manuscript to explicitly state that the participant has provided consent for publication: “The individual in this manuscript has given written informed consent (as outlined in PLOS consent form) to publish these case details”. 

7. We note that Figure 1 (C) in your submission contain copyrighted images. All PLOS content is published under the Creative Commons Attribution License (CC BY 4.0), which means that the manuscript, images, and Supporting Information files will be freely available online, and any third party is permitted to access, download, copy, distribute, and use these materials in any way, even commercially, with proper attribution. For more information, see our copyright guidelines: http://journals.plos.org/plosone/s/licenses-and-copyright.

(1) You may seek permission from the original copyright holder of Figure 1 (C) to publish the content specifically under the CC BY 4.0 license. 

Reviewers' comments:

Reviewer's Responses to Questions

**Comments to the Author**

1. Is the manuscript technically sound, and do the data support the conclusions?

Reviewer #1: Yes

Reviewer #2: Yes

2. Has the statistical analysis been performed appropriately and rigorously? 

Reviewer #1: N/A

Reviewer #2: N/A

3. Have the authors made all data underlying the findings in their manuscript fully available?

Reviewer #1: Yes

Reviewer #2: Yes

4. Is the manuscript presented in an intelligible fashion and written in standard English?

Reviewer #1: Yes

Reviewer #2: Yes

5. Review Comments to the Author

Reviewer #1: The paper by Angelucci and colleagues is an important study on the formation dynamics of the complex cave deposit of Gruta da Oliveira, carried with a multi-proxy approach which includes the macroscopical characterization of the stratigraphical units, the micromorphological analysis of soil samples, the analysis of biogenic and anthropogenic features such as lithics, heated remains, hearths, with spatial analysis and refittings. This is an excellent example, both for the methods applied and for the discussed results, of a detailed work aimed at understanding to which extent natural and biological agents affected the accumulation of the cave deposit. This is why I think this paper deserves to be published, since it can be of reference for archaeologists dealing with archaeological excavation of Palaeolithic cave sites. I just have some minor comments which I hope will help to improve the manuscript.

- In the introduction, authors begin straight to describe the geological context and the site. I think that a conceptually broader introduction could help to frame a main question addressed by the study. How is this paper a contribution to research? Is it only addressed to frame the Gruta da Oliveira sequence from a geomorphological and archaeological point of view? Since I believe in the broad scope of such research, I think that introducing the question, e.g., of the anthropic and natural impact on the formation dynamics of cave deposits, especially within the Neandertal sites of the Iberian Peninsula, can better enhance the work and help the reader focus on the issue at a broader level.

- In the lines 303-304, about the deformation of the deposit caused by a localized sinkhole, authors say that “the sinking of the basal parts of the stratification in line P of the grid, along the north wall of the passage, which is well apparent in profile view (Figs. 9-10)”. However, this is not evident to me in such figures. Can it be highlight in a better way?

- Right below, in the lines 304-307, speaking about the “Mousterian cone”, authors say that “The fact that no refits have so far been found linking layers 26-27 with layer 14 or any other layer further up in the “Middle Palaeolithic sequence” complex suggests that, by the time the accumulation of layer 15 came to an end, overall stabilisation of the sedimentary body had been achieved”. Does it mean that the Mousterian cone derives from the Access Corridor’s layers 15 to 25?

- About this, in the chapter 3.3.3 (Mousterian Cone and Passage of the Columns) only layer 27 (Passage of the Columns) is discussed, and nothing is said about the Mousterian Cone. My advice is to better expose the origin of the Mousterian Cone deposit and the relationship with the upper (Access Corridor) deposit.

- When authors talk about the pedogenic features of the deposit, I think that some information on the preservation conditions of the archaeological material could be useful. Are lithics and bones affected by patinas/concretions and other weathering features?

- In the lines 562-566 authors affirm that the concentration of burnt bones in layers 23-25, grid unit P15, is due to percolation from the above layer 22 hearths. However, no burnt bones are documented in layer 22, grid unit P15, but there are in O15, O16 and P16. Is it possible that the “suffusion-induced deformation” cited by authors could have moved all the burnt bones (but not all the lithics) of that specific square?

Reviewer #2: In their article “Formation processes, fire use, and patterns of human occupation across the Middle Palaeolithic (MIS 5a-5b) of Gruta da Oliveira (Almonda karst system, Torres Novas, Portugal)”, Angelucci et al., provide a complete and holistic revision of sedimentary formation processes of one of the most important MIS5 Paleolithic sites of South Western Europe. Combined micromorphological, spatial tools distribution, paleobotanical and archaeozoological analyses allow the authors to discuss: a) the environmental significance of the sedimentary inputs sources into the cave during human occupations, b) the morphological evolution of the Gruta de Oliveira karstic and depositional systems during the MIS5, c) the diachronic fire use and site occupancy patterns evolution by humans and animals.

Because of the extreme complexity of the functioning of the Oliveira karstic and depositional systems through the time and the importance of the archaeological findings excavated during the last decades there, such formation processes investigation study is considered as a necessary in the framework of the Almonda archaeological sites context, gaining a high relevance impact for the scientific community and deserving a publication in PLOS ONE journal. Although the manuscript appears to be in overall good shape for publication, I shall try to suggest here below some lines of action to be considered for a further improvement of the text during the revision process.

General Comments

A – The Scientific Issue

The introduction paragraph appears rather to the reader as a Geological and geochronological context paragraph (lines 42-82), study context (90-98), structure of the manuscript (lines 102-110). I suggest to move all these manuscript parts to a new “Context of the Study” paragraph or sub-paragraph into the introduction section, focusing the Introduction on the problematic and on the scientific broader context of the study.

Indeed, in the introduction paragraph, the authors should emphasise the fundamental scientific issues which have lead the development of the site formation processes/micromorphological study approach, answering to some questions: why Gruta de Oliveira deposits need a geoarcheological approach? (I suggest to highlight in one or two sentences the complexity of the Almonda karstic system and the relevance of the archaeological findings in the Middle Paleolithic research of Oliveira). Why the understanding of the morphological evolution of the karstic and depositional systems could provide human occupancy behaviour patterns inferences? How the geoarchaeological approach can help the understanding of Middle Palaeolithic sites of Spain? Is because the authors discuss about the fire use topic that it would be appreciated also an archaeological context of Fire use in South Western Europe.

The addition of this kind of information, will improve the relevance of the article into the scientific community, highlighting the importance of the geoarcheological approach. In terms of manuscript structure, such introduction will help to put the links between the scientific problematic and the main results of the article presented in the discussion giving a completeness shape of the article.

B- Stratigraphy representations and interpretative drawings

As evident in Fig. 9, layers 20, 22 and 23 are traversed by vertical sedimentary disturbance in the left side, and units 23-24-25 in the right side, following the cave wall morphology. Such feature is described in the text in lines 267-270, however, is not represented in the stratigraphic interpretative drawing (Fig. 10). Similarly, in Fig. 8, the lines delimiting the units didn’t’ follow the morphology of the layers.

Considering the importance of the morphology of the sedimentary units in the formation processes interpretation, I would suggest more precision into the stratigraphy representations.

C- The sedimentary ensembles

One of the fundamental synthesis points of the paper is represented by the definition of the sedimentary ensembles (lines 469-481). They are defined on the basis of the deposits morphogenesis and micromorphological interpretations. Their interpretation should represent one of the main results of the formation processes issue of the article.

With the aim to highlight such result and to simplify the reading of the article, I would suggest to foreground the definition and the interpretation of the ensembles throughout the text and in the illustrations. For example, adding into the Fig. 12 the graphic representation of the ensembles with the related Sedimentary Units. I would also suggest to favour the use of the sedimentary ensemble nomenclature (from line 658) together with the units nomenclature throughout the discussion section. Would also be appreciated the interpretation formation processes ensemble by ensemble in the discussion section.

D – Discussion organisation

With the aim to help the reader and to improve the structuration of the paper I would suggest to organise the discussion section in sub-paragraphs thought the following way:

Lines 618-657: “Sedimentary sources inputs significance”

Lines 658-704: “Morpho-sedimentary evolution of Gruta de Oliveira during the MIS 5”

As suggested above, in this sub-paragraph, please favour the use of the sedimentary ensembles names previously defined, together with the units numbers. This part would need also a chronological indication (ages Ka) of the mentioned layers/ensembles.

Lines 705-772: “Environmental markers”

Lines 773-855: “Fire use and occupancy patterns evolution”

Detailed Comments:

Lines 19-22: These two sentences indicate that there is some sedimentary material coming from the slope, and that the coarse fraction comes from the cave bedrock. However, in the text is indicated that the coarse material is represented by both LST SIL sources. Please clarify the sentences indicating that the filling material is represented by fine and coarse material coming from both sources.

Lines 22-23: How the erosion can induce the tectonic and structural activity? It should be the opposite. Please reformulate the sentence: “The erosional retreat of the scarp face, caused by the tectonic activity, explain the large, roof-collapsed rock masses found through the stratification.”

Line 25: “local factors” is too generic.

Line 26: “hunted ungulates species” would be more appropriate than “composition of the hunted ungulates”.

Line 30: please indicate the age range and the sedimentary ensemble for 15-19 layers

Line 31: the same for 7-14 layers.

Lines 30-31: Why is reported the faunal assemblage of 15-19 and 7-14 and there is no information for units 20-25 in the abstract? Please provide a homogeneous presentation of results in the abstract following the chronological order of the layers.

Lines 17-37: Generally, as announced in the title, we should expect an emphasis of the sedimentary processes results in the abstract.

Lines 42-98: As exposed in the General comment-A, this part should belong to a “Context of the study” sub-paragraph.

Line 114: “excavation unit” would be more appropriate.

Line 130: please refers to the three excavated areas illustration

Line 136: « Twenty undisturbed sedimentary samples….”

Line 137: “simple extraction” is too generic, please add: “ without the use of kubiena boxes or plaster bands”.

Line 226: “carbonate crust”

Lines 267-271: Why such disturbance is not indicated in the interpretative drawing?

Line 284: “randomly” would be more appropriate than “chaotically”

Lines 323: which basal layer?

Line 338: Fig. 15b

Lines 365-367: Be careful, authigenic gypsum is not always derived from weathering. And please define the type of authigenic single crystals/intergrowth formation following the Stoops 2003 guidelines for help the interpretative argumentation.

Line 394: Can the authors say if phosphate accumulation derived from guano? Are there evidences of biocorrosion features on the cave walls?

Line 419: Bone means archaeological finding corresponding to a specific anatomically part of the skeleton. Consequently: are 263 burnt bones or 263 burnt bone fragments?

Line 435: “anthropogenic inputs (ABC group source)”

Line 439: please provide a reference for the “thermal alteration” interpretation.

Line 513 (Fig. 18): With the aim to help the reading and the understanding of this paragraph, please add in Fig. 18 the name of the different excavation areas.

Line 528: the term “décapage” has been introduced here in the text for the first time, without having been previously defined. Maybe did it correspond to the “excavation unit” defined in Line 114? If yes, please add this term in the methodological section.

Fig. 18 and Fig. 20: It would be helpful if in these illustrations a sub-illustration numbering will be provided for each graphic composing them (e.g. Fig. 18a, 18b, 18c…..), with a related referring between the text and the illustrations.

Lines 576-577: This sentence is not clear. Please rephrase.

Line 634: “chemical weathering processes” would be more appropriate than “rock degradation processes”.

Line 636: « discontinuous cryoclastic events » would be more appropriate than “discontinuous frost action conditions”

Line 658: “roof collapse” or “cave wall collapse”

Line 666: “originated outside” is not necessary here.

Lines 696-698: In this sentence the authors indicate that the stratigraphy is characterized by some sedimentary hiatus. However, in lines 593 and 628 they indicate that the sedimentation is “constant” and “continuous” respectively. I suppose that the lines 593 and 628 are referring to the average sedimentation rate through the time. However, with the aim to avoid any contradictory affirmations, please clarify these sentences.

Line 721: I suggest to change the “…when humans were not around.” affirmation with a more appropriate one (e.g. “during temporary periods of cave abandon”).

Lines 882-883: Here the authors indicate how “Humans were the main contributors to the deposit’s content”. However, in the abstract we read “The accumulation primarily consists of sediment washed in from the slope through gravitational…”. Additionally, the results and the interpretation of the data indicate that the accumulation is principally composed by clastic material (coarse components) of natural origin. Please provide a unique interpretation.

All of these comments are intended to stimulate discussion and a clearer explanation of the interesting aspects dealt with in this article. Their aim is to help producing an even more solid scientific contribution of the geoarchaeological an micromorphological approaches applied on Paleolithic cave contexts in Western Europe.

Faithfully yours,

Carlo Mologni

6. PLOS authors have the option to publish the peer review history of their article (what does this mean?). If published, this will include your full peer review and any attached files.

Reviewer #1: No

Reviewer #2: **Yes: **Carlo Mologni

Quaternary African Geology Research Associate

Leverhulme Centre for Human Evolutionary Studies

University of Cambridge

Fitzwilliam street, Cambridge CB2 1QH, UK

E-mail: cm2214@cam.ac.uk

---

## [Author Response · Author response to Decision Letter 0]

12 Aug 2023

Response to Reviewers

To Reviewer #1 

The paper by Angelucci and colleagues is an important study on the formation dynamics of the complex cave deposit of Gruta da Oliveira, carried with a multi-proxy approach which includes the macroscopical characterization of the stratigraphical units, the micromorphological analysis of soil samples, the analysis of biogenic and anthropogenic features such as lithics, heated remains, hearths, with spatial analysis and refittings. This is an excellent example, both for the methods applied and for the discussed results, of a detailed work aimed at understanding to which extent natural and biological agents affected the accumulation of the cave deposit. This is why I think this paper deserves to be published, since it can be of reference for archaeologists dealing with archaeological excavation of Palaeolithic cave sites. I just have some minor comments which I hope will help to improve the manuscript.

- In the introduction, authors begin straight to describe the geological context and the site. I think that a conceptually broader introduction could help to frame a main question addressed by the study. 

We have added a broader-framing, introductory paragraph to the original Introduction section. 

How is this paper a contribution to research? Is it only addressed to frame the Gruta da Oliveira sequence from a geomorphological and archaeological point of view? Since I believe in the broad scope of such research, I think that introducing the question, e.g., of the anthropic and natural impact on the formation dynamics of cave deposits, especially within the Neandertal sites of the Iberian Peninsula, can better enhance the work and help the reader focus on the issue at a broader level.

The added paragraph mentioned above addresses the reviewer’s concerns. 

- In the lines 303-304, about the deformation of the deposit caused by a localized sinkhole, authors say that “the sinking of the basal parts of the stratification in line P of the grid, along the north wall of the passage, which is well apparent in profile view (Figs. 9-10)”. However, this is not evident to me in such figures. Can it be highlight in a better way?

The reviewer is correct, the sinking is not obvious in the picture and drawing. We have deleted from the text the sentence “which is well apparent in profile view”.

- Right below, in the lines 304-307, speaking about the “Mousterian cone”, authors say that “The fact that no refits have so far been found linking layers 26-27 with layer 14 or any other layer further up in the “Middle Palaeolithic sequence” complex suggests that, by the time the accumulation of layer 15 came to an end, overall stabilisation of the sedimentary body had been achieved”. Does it mean that the Mousterian cone derives from the Access Corridor’s layers 15 to 25?

Yes, that’s what it means. We have clarified.

- About this, in the chapter 3.3.3 (Mousterian Cone and Passage of the Columns) only layer 27 (Passage of the Columns) is discussed, and nothing is said about the Mousterian Cone. My advice is to better expose the origin of the Mousterian Cone deposit and the relationship with the upper (Access Corridor) deposit.

Layer 27 corresponds to the fill of the Passage of the Column and itself derives also from layers 15-25 of the Access Corridor. Local topography made for the deposit excavated in this part of the cave system to appear as a cone (layer 26) rising above a flat surface (defined at the time of excavation as the interface with layer 27). In terms of formation process, layer 27 mixes such derived material with other components and that is why it is described separately. We have clarified this point.

- When authors talk about the pedogenic features of the deposit, I think that some information on the preservation conditions of the archaeological material could be useful. Are lithics and bones affected by patinas/concretions and other weathering features?

“Pedofeature” (not “pedogenic feature”) is a descriptive term of soil micromorphology, which doesn’t strictly involve the action of soil dynamics. A short sentence was added concerning the preservation of archaeological remains around line 340 of the original manuscript. 

- In the lines 562-566 authors affirm that the concentration of burnt bones in layers 23-25, grid unit P15, is due to percolation from the above layer 22 hearths. However, no burnt bones are documented in layer 22, grid unit P15, but there are in O15, O16 and P16. Is it possible that the “suffusion-induced deformation” cited by authors could have moved all the burnt bones (but not all the lithics) of that specific square?

Grid unit P15 did yield burnt bones in layers 20-22: 9 in layer 29, 10 in layers 21-22. However, (1) bubble size is in proportion to the total number of burnt bones in each ensemble, and (2) the grid units where the hearth was located represent the overwhelming majority of the burnt bones in ensemble 20-22. That is why that total of 19 corresponds, in the graphical representation, to a tiny dot that is hard to visualize but is there. The number of burnt bones yielded by square P15 in immediately underlying layer 23 is 12. It is this limited number that we had in mind when we wrote the sentence that the reviewer has questions about. We thank the reviewer for pointing out the issue, which we have clarified in the text as follows (which is a rewrite of lines 562-566 of the original manuscript):

The concentration of burnt bones found in grid unit P15 at the elevation of layers 23-25 is another apparent anomaly. In this case, two factors are involved: downward percolation, along the cave wall, onto layer 23, of burnt bones originally associated with the layer 22 hearths, in connection with the latter’s suffosion-induced deformation (as indeed noted at the time of excavation); the spatial constraints posed by the outline of the cave walls at the elevation of layers 24-25, which largely restricted the accumulation of finds to line P of the excavation grid, where those layers’ lithics and faunal remains (including the burnt ones) mostly come from.

To Reviewer #2 

In their article “Formation processes, fire use, and patterns of human occupation across the Middle Palaeolithic (MIS 5a-5b) of Gruta da Oliveira (Almonda karst system, Torres Novas, Portugal)”, Angelucci et al., provide a complete and holistic revision of sedimentary formation processes of one of the most important MIS5 Paleolithic sites of South Western Europe. Combined micromorphological, spatial tools distribution, paleobotanical and archaeozoological analyses allow the authors to discuss: a) the environmental significance of the sedimentary inputs sources into the cave during human occupations, b) the morphological evolution of the Gruta de Oliveira karstic and depositional systems during the MIS5, c) the diachronic fire use and site occupancy patterns evolution by humans and animals.

Because of the extreme complexity of the functioning of the Oliveira karstic and depositional systems through the time and the importance of the archaeological findings excavated during the last decades there, such formation processes investigation study is considered as a necessary in the framework of the Almonda archaeological sites context, gaining a high relevance impact for the scientific community and deserving a publication in PLOS ONE journal. Although the manuscript appears to be in overall good shape for publication, I shall try to suggest here below some lines of action to be considered for a further improvement of the text during the revision process.

General Comments

A – The Scientific Issue

The introduction paragraph appears rather to the reader as a Geological and geochronological context paragraph (lines 42-82), study context (90-98), structure of the manuscript (lines 102-110). I suggest to move all these manuscript parts to a new “Context of the Study” paragraph or sub-paragraph into the introduction section, focusing the Introduction on the problematic and on the scientific broader context of the study.

Indeed, in the introduction paragraph, the authors should emphasise the fundamental scientific issues which have lead the development of the site formation processes/micromorphological study approach, answering to some questions: why Gruta de Oliveira deposits need a geoarcheological approach? (I suggest to highlight in one or two sentences the complexity of the Almonda karstic system and the relevance of the archaeological findings in the Middle Paleolithic research of Oliveira). Why the understanding of the morphological evolution of the karstic and depositional systems could provide human occupancy behaviour patterns inferences? How the geoarchaeological approach can help the understanding of Middle Palaeolithic sites of Spain? Is because the authors discuss about the fire use topic that it would be appreciated also an archaeological context of Fire use in South Western Europe.

The addition of this kind of information, will improve the relevance of the article into the scientific community, highlighting the importance of the geoarcheological approach. In terms of manuscript structure, such introduction will help to put the links between the scientific problematic and the main results of the article presented in the discussion giving a completeness shape of the article.

We believe that the paragraph added at the beginning of the Introduction to address a similar concern of Reviewer 1 also addresses the above points. 

B- Stratigraphy representations and interpretative drawings

As evident in Fig. 9, layers 20, 22 and 23 are traversed by vertical sedimentary disturbance in the left side, and units 23-24-25 in the right side, following the cave wall morphology. Such feature is described in the text in lines 267-270, however, is not represented in the stratigraphic interpretative drawing (Fig. 10). Similarly, in Fig. 8, the lines delimiting the units didn’t’ follow the morphology of the layers.

Considering the importance of the morphology of the sedimentary units in the formation processes interpretation, I would suggest more precision into the stratigraphy representations.

With regards to Figures 9-10, the disturbance mentioned by the reviewer is indeed represented in the Figure 10 drawing — in the form of shaded areas superimposed to the colours, as identified by letter “g” and “h” of the key). We have enhanced the contrast to make that shading more readily apparent.

With regards to Figure 8, the lines were intended to indicate the general course of the boundaries, not their exact outline. To avoid confusion, we have removed the lines. For that level of detail, the caption referred the reader to the drawing published in the 2009 paper, and we believe that is sufficient.

C- The sedimentary ensembles

One of the fundamental synthesis points of the paper is represented by the definition of the sedimentary ensembles (lines 469-481). They are defined on the basis of the deposits morphogenesis and micromorphological interpretations. Their interpretation should represent one of the main results of the formation processes issue of the article.

With the aim to highlight such result and to simplify the reading of the article, I would suggest to foreground the definition and the interpretation of the ensembles throughout the text and in the illustrations. For example, adding into the Fig. 12 the graphic representation of the ensembles with the related Sedimentary Units. I would also suggest to favour the use of the sedimentary ensemble nomenclature (from line 658) together with the units nomenclature throughout the discussion section. Would also be appreciated the interpretation formation processes ensemble by ensemble in the discussion section.

We disagree with the reviewer’s suggestion, for a number of reasons. The ensemble framework is an analytical construct, one that we believe most adequately serves the discussion of the issues of site occupancy addressed in the manuscript. In our opinion, it would be inappropriate for such a construct to obscure the actual stratigraphic succession as observed in the field and analysed under the microscope. The layers remain the basic units into which the succession can be subdivided and, therefore, reference to them ought to retain precedence in the description of any groupings that the layers are combined into. Contrary to the reviewer’s opinion, we believe that our mode of presentation actually facilitates the reading of the paper. In addition, one must bear in mind that, depending on the nature of the question being asked, it may be necessary, or advisable, to vary the composition of the layer groups, i.e., of the ensembles, and that is why, even in the present manuscript, we sometimes need to present the data in ways that correspond to variations around the proposed ensemble framework. For instance, while it makes sense to consider layers 15-19 together as the “Access Corridor Upper” ensemble for the purposes of chronostratigraphy and the identification of a cultural horizon of “Mousterian with cleavers”, it turns out that the higher integrity of layer 15 makes it useful to separate it from layers 16-19 when it comes to interpret spatial distributions or the paleoclimatic signal provided by the microfaunal remains. Ditto for layers 20-22 and the “Access Corridor Middle” ensemble, which, because of the impact that boulder collapse had on the integrity of the interface with layer 19, makes it advisable to consider layers 21-22 separately when dealing with a discussion of the extent to which Access Corridor Middle and Access Corridor Upper differ in terms of stone tool technology. Finally, we note that Figure 12 would be an inappropriate place to indicate the different ensembles, as it illustrates a topographic cross-section along a line of the grid chosen to highlight the physical separation between the sedimentary column filling the Access Corridor and the Mousterian Cone, and the latter’s topographic relationship with the Passage of the Column. In fact, the ensemble framework is illustrated in Figure 2B, where the layer groupings indicated for the Access Corridor follow that exact framework. To make this point clearer, the ensemble designations have been added to Figure 2C as acronyms whose meaning is explained in the caption.

D – Discussion organisation

With the aim to help the reader and to improve the structuration of the paper I would suggest to organise the discussion section in sub-paragraphs thought the following way:

Lines 618-657: “Sedimentary sources inputs significance”

Lines 658-704: “Morpho-sedimentary evolution of Gruta de Oliveira during the MIS 5”

As suggested above, in this sub-paragraph, please favour the use of the sedimentary ensembles names previously defined, together with the units numbers. This part would need also a chronological indication (ages Ka) of the mentioned layers/ensembles.

Lines 705-772: “Environmental markers”

Lines 773-855: “Fire use and occupancy patterns evolution”

We have followed the reviewer’s suggestion.

Detailed Comments:

Lines 19-22: These two sentences indicate that there is some sedimentary material coming from the slope, and that the coarse fraction comes from the cave bedrock. However, in the text is indicated that the coarse material is represented by both LST SIL sources. Please clarify the sentences indicating that the filling material is represented by fine and coarse material coming from both sources.

Here, the reviewer confuses “coarse material” as used in the context of the macroscopic description of the sediment, which means stones, cobble, pebbles, boulders, etc., with “coarse components” as considered from the point of view of archaeological micromorphology: they are not the same. We changed the wording at those points in the text where the different meanings of “coarse” could be misleading (e.g., in the abstract).

Lines 22-23: How the erosion can induce the tectonic and structural activity? It should be the opposite. Please reformulate the sentence: “The erosional retreat of the scarp face, caused by the tectonic activity, explain the large, roof-collapsed rock masses found through the stratification.”

Thanks for noting the typo. The word “by” was erased. The text now reads “Tectonic activity and structural instability caused the erosional retreat of the scarp face, explaining the large, roof-collapsed rock masses found through the stratification.”

Line 25: “local factors” is too generic.

It is generic because it refers to several factors (geological, geomorphological, environmental etc.) that locally control site formation and are explained in the text, and it would be too long to list them in the abstract. We see no need to change the current wording. 

Line 26: “hunted ungulates species” would be more appropriate than “composition of the hunted ungulates”. 

Rephrased to “the assemblages of hunted ungulates”

Line 30: please indicate the age range and the sedimentary ensemble for 15-19 layers.

Done

Line 31: the same for 7-14 layers.

Done.

Lines 30-31: Why is reported the faunal assemblage of 15-19 and 7-14 and there is no information for units 20-25 in the abstract? Please provide a homogeneous presentation of results in the abstract following the chronological order of the layers. 

This being the abstract, we provide a generic assessment of the faunal assemblage, which is valid for all layers — “the composition of the assemblages of hunted ungulates, mostly open-country and rocky terrain taxa (rhino, horse, ibex) throughout” (throughout now added to make that clearer) — and note the differences in the composition of the carnivore guilds between layers 7-14 and 15-25 (the original manuscript mentioned layers 15-19 only, which we have now corrected).

Lines 17-37: Generally, as announced in the title, we should expect an emphasis of the sedimentary processes results in the abstract.

Emphasis is already placed on sedimentary processes, which is what >50% of the abstract deals with (e.g., lines 17-29 of the original submission).

Lines 42-98: As exposed in the General comment-A, this part should belong to a “Context of the study” sub-paragraph. 

See above. 

Line 114: “excavation unit” would be more appropriate.

We disagree, and left the text unchanged, except for the elimination of a redundant clause.

Line 130: please refers to the three excavated areas illustration.

Reference to fig. 2A was added.

Line 136: « Twenty undisturbed sedimentary samples….”

The same sentence reports “thanks to good sediment cohesion”, adding “sedimentary” here is pleonastic.

Line 137: “simple extraction” is too generic, please add: “ without the use of kubiena boxes or plaster bands”.

Done. Please notice that they are called “tins”, not “boxes”.

Line 226: “carbonate crust”.

This refers to the process “incrustation … formation” not to the object. We haven’t changed it.

Lines 267-271: Why such disturbance is not indicated in the interpretative drawing?

It is indicated indeed; see above.

Line 284: “randomly” would be more appropriate than “chaotically”.

Done.

Lines 323: which basal layer?

Changed.

Line 338: Fig. 15b

OK

Lines 365-367: Be careful, authigenic gypsum is not always derived from weathering. And please define the type of authigenic single crystals/intergrowth formation following the Stoops 2003 guidelines for help the interpretative argumentation.

We didn’t write that gypsum is authigenic, we wrote “All samples contain sand-sized crystals of gypsum showing traces of weathering”. The gypsum grains are components, not part of pedofeatures.

Line 394: Can the authors say if phosphate accumulation derived from guano? Are there evidences of biocorrosion features on the cave walls?

This is the “Results” chapter: it is not appropriate to discuss the origin of phosphate here. The point is considered in the “Discussion” chapter.

Line 419: Bone means archaeological finding corresponding to a specific anatomically part of the skeleton. Consequently: are 263 burnt bones or 263 burnt bone fragments?

Fragments. Clarified.

Line 435: “anthropogenic inputs (ABC group source)”.

Changed.

Line 439: please provide a reference for the “thermal alteration” interpretation.

Done.

Line 513 (Fig. 18): With the aim to help the reading and the understanding of this paragraph, please add in Fig. 18 the name of the different excavation areas.

Done.

Line 528: the term “décapage” has been introduced here in the text for the first time, without having been previously defined. Maybe did it correspond to the “excavation unit” defined in Line 114? If yes, please add this term in the methodological section.

Although one sometimes sees the term “décapage” used in the sense of “excavation unit”, and such tends indeed to be the practice of francophone researchers, it is clear from the context that, here, the word is used in the sense of “the action of exposing the surface of a new stratigraphic unit as it is reached during the excavation of the overlying one”. There is no need to introduce the term in the Methodological section.

Fig. 18 and Fig. 20: It would be helpful if in these illustrations a sub-illustration numbering will be provided for each graphic composing them (e.g. Fig. 18a, 18b, 18c…..), with a related referring between the text and the illustrations. 

Done for Figure 18. There’s no need to do that for Figure 20, as each of the latter’s panels plots a different unit, which is appropriately and unambiguously identified. Moreover, when reference to this figure is made in the text, it is the comparative information provided by the visualisation of all panels together that is of relevance.

Lines 576-577: This sentence is not clear. Please rephrase.

Done. The text now reads as follows: “These indexes have shortcomings that require discussion. Using the total size of excavated areas in the denominator implies (a) when distributions are heterogeneous (as in e.g., layers 13-15, where finds were much denser in the 27-S Chamber than in the Access Corridor; Fig. 20), that the index will underestimate the true density of the actual concentrations and, hence, the intensity of the human activity that such concentrations reflect, and (b) if rates of sedimentation vary across units, that the intensity of the human activity will be overestimated in those formed when accumulation was slower. Considering excavated volumes can mitigate the bias caused by spatially heterogeneous distributions but works against units formed under faster rates of accumulation.”

Line 634: “chemical weathering processes” would be more appropriate than “rock degradation processes”.

The process of wall disintegration is mostly physical, not only chemical. We prefer to keep it like that.

Line 636: « discontinuous cryoclastic events » would be more appropriate than “discontinuous frost action conditions”.

No, frost action is broader than cryoclastic. Moreover, the latter word is poorly understandable to English readers.

Line 658: “roof collapse” or “cave wall collapse”.

Clarified.

Line 666: “originated outside” is not necessary here.

We’d like to keep it for clarity.

Lines 696-698: In this sentence the authors indicate that the stratigraphy is characterized by some sedimentary hiatus. However, in lines 593 and 628 they indicate that the sedimentation is “constant” and “continuous” respectively. I suppose that the lines 593 and 628 are referring to the average sedimentation rate through the time. However, with the aim to avoid any contradictory affirmations, please clarify these sentences. 

Rates of accumulation in lines 593-628 do refer to the overall build-up of the succession through time. We see no contradiction between our framing of the issue in those lines and our diagnosis that the succession features hiatuses and that the different units may have formed at different rates. Our argument is that, even though we know that rates of accumulation varied, holding them constant is unlikely to bias the distributions for the reasons given (“the source of the sediment that infilled the cave remained the same throughout, and the animal and plant proxies indicate that the oscillations in climate and environment recorded across the archaeological sequence were rhythmical and of similar magnitude”). We believe that our text is unambiguous and see no need to change it.

Line 721: I suggest to change the “…when humans were not around.” affirmation with a more appropriate one (e.g. “during temporary periods of cave abandon”). 

“Around” was changed to “present in the area”.

Lines 882-883: Here the authors indicate how “Humans were the main contributors to the deposit’s content”. However, in the abstract we read “The accumulation primarily consists of sediment washed in from the slope through gravitational…”. Additionally, the results and the interpretation of the data indicate that the accumulation is principally composed by clastic material (coarse components) of natural origin. Please provide a unique interpretation. 

We apologize for the imprecision in language. Here, by “content” we meant the non-sedimentary components of the deposit (charcoal, animal bone, stone tools). This has been clarified. The sentence now reads “Humans were the main contributors to the deposit’s non-sedimentary components”.

All of these comments are intended to stimulate discussion and a clearer explanation of the interesting aspects dealt with in this article. Their aim is to help producing an even more solid scientific contribution of the geoarchaeological an micromorphological approaches applied on Paleolithic cave contexts in Western Europe.

---

## [Decision Letter · Decision Letter 1]

12 Sep 2023

Formation processes, fire use, and patterns of human occupation across the Middle Palaeolithic (MIS 5a-5b) of Gruta da Oliveira (Almonda karst system, Torres Novas, Portugal)

PONE-D-23-05775R1

Dear Dr. Zilão,

We’re pleased to inform you that your manuscript has been judged scientifically suitable for publication and will be formally accepted for publication once it meets all outstanding technical requirements.

Kind regards,

Enza Elena Spinapolice, Ph.D

Academic Editor

PLOS ONE

Additional Editor Comments (optional):

Reviewers' comments:

Reviewer's Responses to Questions

**Comments to the Author**

1. If the authors have adequately addressed your comments raised in a previous round of review and you feel that this manuscript is now acceptable for publication, you may indicate that here to bypass the “Comments to the Author” section, enter your conflict of interest statement in the “Confidential to Editor” section, and submit your "Accept" recommendation.

Reviewer #1: All comments have been addressed

Reviewer #2: All comments have been addressed

2. Is the manuscript technically sound, and do the data support the conclusions?

Reviewer #1: Yes

Reviewer #2: Yes

3. Has the statistical analysis been performed appropriately and rigorously? 

Reviewer #1: N/A

Reviewer #2: N/A

4. Have the authors made all data underlying the findings in their manuscript fully available?

Reviewer #1: Yes

Reviewer #2: Yes

5. Is the manuscript presented in an intelligible fashion and written in standard English?

Reviewer #1: Yes

Reviewer #2: Yes

6. Review Comments to the Author

Reviewer #1: (No Response)

Reviewer #2: After the revision process, Angelucci and colleagues provided the manuscript in a correct shape for a publication in PLOSONE journal.

The manuscript has been modified and improved following the review #2 comments A, B, C and D.

In details: A) the introduction section has been substantially improved by the addition of a boarder scientific context of the study, which gives a strengthen relevance of the article into the scientific community, highlighting also the importance of the geoarchaeological approach into archaeological studies; B) the stratigraphic representations and interpretative drawings has been slightly modified avoiding any stratigraphic misinterpretations of the described layers; C) the choice to circumvent the review suggestion (comment C) has been correctly augmented by the authors Additionally, the ensemble designations have been added to Figure 2C, which clarify the reading of the manuscript (however, the illustration still need the addition of the capital letter “c”); D) the discussion organisation has been structured following the review suggestions. Minor comments have been correctly addressed providing minor revisions or intelligible argumentations.

Finally, I recommend the current revised manuscript version for a publication in PLOSONE journal.

7. PLOS authors have the option to publish the peer review history of their article (what does this mean?). If published, this will include your full peer review and any attached files.

Reviewer #1: No

Reviewer #2: No

---

## [Editor Report · Acceptance letter]

18 Sep 2023

PONE-D-23-05775R1 

Formation processes, fire use, and patterns of human occupation across the Middle Palaeolithic (MIS 5a-5b) of Gruta da Oliveira (Almonda karst system, Torres Novas, Portugal) 

Dear Dr. Zilhão:

I'm pleased to inform you that your manuscript has been deemed suitable for publication in PLOS ONE. Congratulations! Your manuscript is now with our production department. 

Kind regards, 

on behalf of

Dr. Enza Elena Spinapolice 

Academic Editor

PLOS ONE